# Towards Straggler-Resilient Split Federated Learning: An Unbalanced Update Approach

**Dandan Liang**
Rochester Institute of Technology
Rochester, New York
dl5974@rit.edu

**Jianing Zhang**
Purdue University
West Lafayette, Indiana
zhan4670@purdue.edu

**Evan Chen**
Purdue University
West Lafayette, Indiana
chen4388@purdue.edu

**Zhe Li**
Rochester Institute of Technology
Rochester, New York
zl4063@rit.edu

**Rui Li**
Rochester Institute of Technology
Rochester, New York
rxlics@rit.edu

**Haibo Yang**
Rochester Institute of Technology
Rochester, New York
hbycis@rit.edu

## Abstract

Split Federated Learning (SFL) enables scalable training on edge devices by combining the parallelism of Federated Learning (FL) with the computational offloading of Split Learning (SL). Despite its great success, SFL suffers significantly from the well-known straggler issue in distributed learning systems. This problem is exacerbated by the dependency between Split Server and clients: the Split Server side model update relies on receiving activations from clients. Such synchronization requirement introduces significant time latency, making straggler a critical bottleneck to the scalability and efficiency of the system. To mitigate this problem, we propose `MU-SplitFed`, a straggler-resilient SFL algorithm in zeroth-order optimization that decouples training progress from straggler delays via a simple yet effective unbalanced update mechanism. By enabling the server to perform $\tau$ local updates per client round, `MU-SplitFed` achieves a convergence rate of $\mathcal{O}(\sqrt{d/(\tau T)})$ for non-convex objectives, demonstrating a linear speedup of $\tau$ in communication rounds. Experiments demonstrate that `MU-SplitFed` consistently outperforms baseline methods with the presence of stragglers and effectively mitigates their impact through adaptive tuning of $\tau$. The code for this project is available at `https://github.com/Johnny-Zip/MU-SplitFed`.

## 1 Introduction

Split Federated Learning (SFL) [1–3] integrates the strengths of Federated Learning (FL) [4] and Split Learning (SL) [5, 6], enabling efficient training on resource-constrained devices. FL offers parallel client updates but imposes heavy computation on edge devices [7], while SL reduces client load by offloading computation to the server but suffers from high latency due to its sequential nature. SFL balances these trade-offs, making it a promising framework for scalable training, especially as model sizes grow. However, the relay-based training mechanism in SFL introduces synchronization bottlenecks due to stragglers: clients with the slowest computation or communication speeds delay the overall process [8, 9]. Both global aggregation and client-side updates must wait for the slowest participant, limiting scalability [10]. This issue is a well-known bottleneck in distributed learning that

39th Conference on Neural Information Processing Systems (NeurIPS 2025).

can severely degrade training efficiency [11–14]. This issue is further exacerbated by the increasing size of ML models and the limited computational capacity of edge devices [15].

To address this issue, existing works draw inspiration from straggler mitigation strategies in FL. Adaptive splitting techniques [9, 16] dynamically adjust the client-side cut layer based on network conditions to enforce synchronization. However, this strategy requires the model architecture to expose layers with varying activation dimensions. In modern transformer-based models, where activation sizes are nearly uniform across layers, such flexibility is absent, and shifting the cut therefore provides little benefit: the amount of data transmitted remains essentially constant, leading to persistent communication delays regardless of the split location. Another approach enables asynchronous updates by allowing the server to proceed with stale information [8]. While this reduces idle time, it exacerbates client drift under high data heterogeneity, harming model performance. Although these methods focus on reducing the straggler-induced latency, they often overlook a more dominant factor contributing to training overhead: the total global communication round. As a result, we investigate the following question: *Can we **efficiently** alleviate the impact of stragglers in SFL by strategically **reducing** communication round?*

We provide an affirmative answer in this paper, aiming to accelerate convergence under practical system heterogeneity and thereby reduce training time overhead. We propose `MU-SplitFed`, a SFL framework that leverages unbalanced server-client updates to improve training efficiency by controlling communication frequency. Our approach exploits the computational advantage of powerful servers: instead of idly waiting for slow edge devices, the Split Server performs $\tau$ optimization steps for each client-server communication round, effectively accelerating the training process. To further ease memory and computation burdens on edge devices, we incorporate Zeroth-Order (ZO) optimization on the client side, enabling training without backpropagation [17, 18]. Beyond empirical performance, we provide a rigorous theoretical analysis showing that our method achieves *linear speedup with respect to the server iteration $\tau$*, without relying on strong assumptions. As a result, the total training time is no longer affected by the speed of the slowest client. Our analysis also rigorously accounts for the variance introduced by ZO methods. Due to model splitting, obtaining tight convergence bounds for SFL is more challenging than for standard FL: existing theoretical results in parallel SFL [9] use stronger assumptions (e.g., bounded gradients) while failing to capture the acceleration from clients or local updates. In contrast, our theoretical results not only reflect the acceleration from $\tau$ but also account for other factors such as the number of clients.

We summarize our main contributions as follows:

- **A novel SFL framework:** We propose `MU-SplitFed`, a straggler-resilient SFL framework that effectively reduces the communication round by leveraging unbalanced server-client updates. While other SFL methods suffer from server idleness due to stragglers, `MU-SplitFed` enables the server to perform $\tau$ local updates during each client-server communication round. This effectively utilizes server-side computation and decouples total training time from the slowest client. By incorporating ZO optimization, our method further reduces resource usage on low-capacity edge devices (Sec. 3).

- **Theoretical convergence with linear speedup:** We provide a rigorous convergence analysis of `MU-SplitFed`. The convergence rate is $\mathcal{O}(\sqrt{d/(\tau T)})$ for non-convex setting with the standard assumptions, showing the linear speedup w.r.t the server-side update $\tau$. Furthermore, our theory supports that the reduction in the communication round allows the total training time to become independent of the straggler's speed, directly addressing a major bottleneck in SFL (Sec. 4.2).

- **Insights into model partitioning and update alignment:** We uncover a critical connection between the model splitting strategy and the unbalanced update ratio. Both our theoretical and empirical results demonstrate that aligning the server-side model depth with the value of $\tau$ is essential for optimal convergence. A larger $\tau$ would benefit from more layers on the server side, thus accelerating convergence through more effective server-side computation (Sec. 4.1).

- **Empirical validation:** We validate the effectiveness of `MU-SplitFed` through experiments on benchmark datasets. Beyond its advantage in reducing communication round, our method consistently outperforms baselines under high client heterogeneity, highlighting its practical feasibility for straggler mitigation in SFL (Sec. 5).

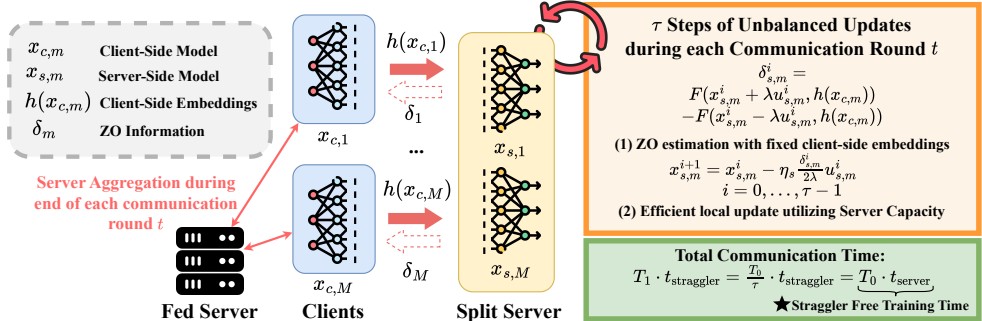

Figure 1: Overview of `MU-SplitFed`. The global model $\mathbf{x}$ is split at the cutting layers into two parts: client-side model $x_c$ and server-side model $x_s$. Each client $m$ trains its local copy $x_{c,m}$ while the Split Server performs $\tau$ local updates on $x_{s,m}$ using the latest embedding, without waiting for the client to finish. At the end of each global round, the Fed Server aggregates all client-side models, and the Split Server averages all the server-side models to form the updated global model.

## 2 Background and Motivation

**SFL Setup.** We consider the parallel SFL framework [1] which combines the model-splitting strategy of SL [5] with the parallel client updates of FL [4]. In SFL, a neural network is partitioned at layer $L_c$, assigning the first $L_c$ layers to the $M$ clients as "client-side model", parameterized by $\{x_{c,1}, \ldots, x_{c,M}\}$, and the remaining layers to the *Split Server*, which maintains $M$ corresponding "server-side model" $\{x_{s,1}, \ldots, x_{s,M}\}$. The combined parameters for client $m$ are denoted as $x_m = \{x_{c,m}, x_{s,m}\}$. Client $m$ computes the embedding $h_m = h(x_{c,m}; \xi_m)$ at the cut layer and sends it to the *Split Server*, which holds the label $y_m$ and computes the loss:

$$F(x_m; \xi_m) = F(x_{s,m}, h(x_{c,m}; \xi_m); y_m), \tag{1}$$

where $\xi_m \sim \mathcal{D}_m$ is the data sample as client input. The server computes a gradient estimate and returns it to the client, which uses it to update both client-side and server-side parameters. The $M$ client-server pairs collaboratively train a global model. After each round of local training, the client-side models are aggregated by the *Fed Server*, while the server-side models are aggregated by the *Split Server*. The overall objective of the SFL framework can be formulated as:

$$\min_x f(x) := \sum_{m=1}^M w_m f_m(x), \tag{2}$$

where $f_m(x) = \frac{1}{|\mathcal{D}_m|} \sum_{\xi \in \mathcal{D}_m} F(x; \xi)$ is the local loss function, and $w_m$ is the weight of client $m$, with $w_m \in [0, 1]$ satisfying $\sum_{m=1}^M w_m = 1$.

**ZO Optimization.** Zeroth-Order Optimization (ZOO) is a gradient-free method, offering an alternative solution for scenarios where explicit gradient computation is impractical, expensive, or unreliable [19–21]. ZOO has shown significant advantages in memory saving because it requires only forward passes [17, 18]. Since our goal is to improve training efficiency for edge devices with limited memory resources, we adopt ZOO to reduce even more memory consumption for our resource-constrained devices. In specific, we adopt Simultaneous Perturbation Stochastic Approximation (SPSA) [22] as our ZO gradient estimator. Let $u$ be uniformly sampled from the Euclidean sphere $\sqrt{d}\mathbb{S}^{d-1}$, for any function $f(x) : \mathbb{R}^d \to \mathbb{R}$ and $\lambda > 0$, we define its ZO gradient estimator as:

$$g(x) = \frac{f(x+\lambda u) - f(x-\lambda u)}{2\lambda} u \tag{3}$$

**Challenges in Mitigating Stragglers in SFL.** The straggler problem is a persistent bottleneck in distributed learning systems, where synchronous training requires coordinated updates across multiple agents [9, 10, 23]. In SFL, this issue is further exacerbated by the interdependence between clients and server. There are two factors that contribute to this severity: 1) the server must wait for all clients to transmit embeddings or gradients before continuing, making the system highly sensitive to the slowest participant; 2) the model is split across client and server, requiring frequent communication during both forward and backward passes. This tight coupling amplifies the impact of stragglers compared to traditional FL, where delays are typically limited to full model updates.

In FL, asynchronous updates have been proposed to mitigate such issues by decoupling client updates from global synchronization [23–27]. However, these approaches are insufficient for SFL,

as they only address global aggregation. In SFL, the straggler problem also arises from split-layer communication, a fundamental difference that makes asynchronous techniques in FL less effective when directly applied to SFL. Recent efforts in SFL have explored adaptive model partitioning to balance computation and communication delays [8, 9, 16]. These methods are constrained by the network architecture and fail to address the core issue: the high communication frequency between clients and the server. As a result, none of the existing straggler solutions explicitly aim to reduce the number of communication between client and Split Server, which is the key problem to SFL's straggler-induced inefficiency. *These limitations point to the need for a new framework that explicitly exploits SFL's structural properties to reduce communication frequency, thereby mitigating stragglers without sacrificing model performance.*

## 3 Methodology

Building upon the aforementioned challenges, we propose `MU-SplitFed` to mitigate the straggler issue by jointly addressing memory inefficiency, computation imbalance, and communication overhead. By combining unbalanced update scheduling and zeroth-order optimization, our algorithm achieves robust and scalable performance tailored for resource-constrained edge devices.

---

**Algorithm 1:** `MU-SplitFed`

---

**Input:** Unbalanced update steps $\tau$, global communication rounds $T$, local learning rate on server side $\eta_s$, learning rate on client side $\eta_c$
**Output:** Global model $x^T = \{x_c^T, x_s^T\}$

1 **each global round** $t = 0, \ldots, T-1$ **do**
2      **each client** $m \in \{1, 2, \ldots, M\}$ **in parallel do**
3         Pull global model for initialization: $x_{c,m}^t \leftarrow x_c^t; x_{s,m}^{t,0} \leftarrow x_s^t$;

     /* Phase 1:  Unbalanced Update on Split Server                            */
4      **each client** $m \in \{1, 2, \ldots, M\}$ **in parallel do**
5         Send embeddings $h_{c,m}^{t+}, h_{c,m}^{t-}$ to the Split Server;
6         **each local iteration** $i = 0, \ldots \tau - 1$ **do**
7             Compute zeroth-order gradient $g_{s,m}^{t,i}$ according to (5);
8             Update Split Server model: $x_{s,m}^{t,i+1} \leftarrow x_{s,m}^{t,i} - \eta_s G_{s,m}^{t,i}$;
9         Compute zeroth-order info $\delta_{c,m}^t$ according to (6) and send it back to the client;
10      Update client model: $x_{c,m}^{t+1} \leftarrow x_{c,m}^t - \eta_c G_{c,m}^t(\mathbf{x}^t)$;

     /* Phase 2:  Model Aggregation on Fed Server                              */
11      Fed Server and Split Server updates according to (7), Fed Server broadcasts $x_c^{t+1}$ to all clients.

---

**Training Procedures.** `MU-SplitFed` integrates an unbalanced update strategy and ZO optimization into the SFL framework. The overall training process consists of two main phases: *1) Unbalanced ZO updates between clients and Split Server:* A subset of clients communicates with their corresponding server-side models on the Split Server and performs local training using ZO optimization in an unbalanced update manner. *2) Federated Aggregation across $M$ models:* The Fed Server collects the updated model weights $x_m$ for $m \in [M]$ and applies the FedAvg strategy to compute a new global model. We detail both phases below and provide the full procedure in Algorithm 1.

**Client Model Perturbation and Forwarding.** At global round $t$, each activated client $m$ samples a data point $\xi_m^t \in \mathcal{D}_m$. To perform ZO updates, the client perturbs its model parameters and computes the corresponding embeddings multiple times. First, the client computes the unperturbed embedding $h_m^t = h(x_{c,m}^t; \xi_m^t)$, and the perturbed embeddings:

$$h_m^{t+} = h(x_{c,m}^t + \lambda u_{c,m}^t; \xi_m^t), \quad \text{and} \quad h_m^{t-} = h(x_{c,m}^t - \lambda u_{c,m}^t; \xi_m^t), \tag{4}$$

where $u_{c,m}^t$ is the perturbation direction sampled according to Equation (3), $\lambda$ is a smooth parameter, $x_{c,m}^t$ is the client-side model at round $t$. We define $H_m^t = \{h_m^t, h_m^{t+}, h_m^{t-}\}$. The client then transmits $H_m^t$ to the server for computing the ZO gradient required for model updates.

**Unbalanced Split Server Update.** The transmission of embeddings follows an on-the-fly manner: each embedding is sent immediately after it is computed. Unlike the client, which requires feedback from the server to proceed updates, the Split Server can compute ZO gradients independently. To fully utilize the server's computational capacity, we introduce an unbalanced update mechanism, allowing the server to perform multiple updates using the unperturbed embedding $h_m^t$. Specifically,

instead of remaining idle, the server initiates multiple local updates using $h_m^t$, while waiting for the full set of perturbed embeddings $h_m^{t+}$ and $h_m^{t-}$. We denote $i = 0, 1, \ldots, \tau - 1$ as the server update round. At global round $t$ and server round $i$, the server perturbs its model parameters and computes the corresponding ZO gradient differences:[1]

$$\delta_{s,m}^{t,i} = F(x_{s,m}^{t,i} + \lambda u_{s,m}^{t,i}, h_m^t) - F(x_{s,m}^{t,i} - \lambda u_{s,m}^{t,i}, h_m^t), \tag{5}$$

where $u_{s,m}^{t,i}$ is sampled according to (3), and $x_{s,m}^{t,i}$ denotes the server-side model parameters for client $m$ at global round $t$ and server update step $i$. The corresponding ZO gradient estimator is computed as: $G_{s,m}^{t,i} = \frac{\delta_{s,m}^{t,i}}{2\lambda} u_{s,m}^{t,i}$, where $\delta_{s,m}^{t,i}$ denotes the loss difference obtained from the perturbed embeddings. The server-side model is updated iteratively over $\tau$ local steps using the ZO oracle: $x_{s,m}^{t,i+1} = x_{s,m}^{t,i} - \eta_s^t G_{s,m}^{t,i}, i \in [0, \tau)$.

**Zeroth-order Back Propagation and Client Update.** After completing server-side local updates, it then computes the ZO loss differences required for client-side model updates:

$$\delta_{c,m}^t = F(x_{s,m}^{t,\tau}, h_m^{t+}) - F(x_{s,m}^{t,\tau}, h_m^{t-}), \tag{6}$$

where each $\delta_{c,m}^t$ is a scalar and incurs minimal communication overhead. These ZO differences are sent back to the client. Clients compute their ZO estimates as $G_{c,m}^t = \frac{\delta_{c,m}^t}{2\lambda} u_{c,m}^t$, and update their models via $x_{c,m}^{t+1} = x_{c,m}^t - \eta_c^t G_{c,m}^t$.

**Global Aggregation.** At the end of the global communication round $t$, once all activated local models $x_m = \{x_{c,m}, x_{s,m}\}$ has completed their update, the Fed Server collects the updated parameters $x_{c,m}$ and performs model aggregation, while the Split Server also locally aggregates $x_{s,m}$ and performs an update on $x_s$:

$$x_c^{t+1} = x_c^t - \eta_g \sum_m w_m (x_{c,m}^{t+1} - x_{c,m}^t), \quad \text{and} \quad x_s^{t+1} = x_s^t - \eta_g \sum_m w_m (x_{s,m}^{t,\tau} - x_{s,m}^t), \tag{7}$$

where $w_m$ denotes the aggregation weight for client $m$, in our algorithm we choose to set $w_m = \frac{1}{M}$. $\eta_g$ is the learning rate for global update. Then, the Fed Server broadcasts $x_c^{t+1}$ to all clients.

## 4 Convergence Analysis

In this section, we present a rigorous convergence analysis of `MU-SplitFed`. Specifically, we want to quantify the effect of our unbalanced update mechanism on convergence. However, in FL, this effect may be intertwined with other factors such as data and system heterogeneity. To isolate the influence of the unbalanced updates, we first analyze the single-client setting, which simplifies to a standard SL framework (Sec. 4.1). Then, we propose our general result under SFL settings (Sec. 4.2). The complete proofs are deferred to Appendix C and D. Here, we first make some standard assumptions that will facilitate our analysis.[2]

**Assumption 4.1** (*L*-Smooth)**.** The loss function $f$ is bounded from below, and is $L$-smooth, i.e. $\forall x, y, \| \nabla f(x) - \nabla f(y) \| \leq L \|x - y\|$.

**Assumption 4.2** (Bounded Variance)**.** The variance of the stochastic gradient w.r.t. the client and the server is upper-bounded by $\sigma_c^2$ and $\sigma_s^2$. Specifically, for $\forall \xi \in \mathcal{D}_m, \|\nabla_{x_c} f(x; \xi) - \nabla_{x_c} f(x)\|^2 \leq \sigma_c^2$ and $\|\nabla_{x_s} f(x; \xi) - \nabla_{x_s} f(x)\|^2 \leq \sigma_s^2$.

### 4.1 Convergence Analysis for `MU-Split`

To analyze the impact of multiple server updates alone, we consider the special case where $M = 1$, denoted as `MU-Split`, which reduces to the SL setting. The convergence of `MU-Split` is established in the following theorem:

**Theorem 4.1.** *Under Assumption 4.1 and 4.2, and let the server iteration number be $\tau$. If the learning rates on client and server satisfy $\eta_c / \tau = \eta_s = \eta \leq \min\{\frac{1}{64L(\tau + 2d_s)}, \frac{1}{16L\tau d_c}\}$, the sequence of iterates generated by our `MU-Split` satisfies:*

$$\frac{1}{T} \sum_{t=0}^T \mathbb{E}[\|\nabla_{\mathbf{x}} f(\mathbf{x}^t)\|^2] \leq \frac{4\mathcal{F}}{\eta\tau T} + 16\eta L(\eta\tau L + 1)d_s\sigma_s^2 + 8\eta\tau L d_c\sigma_c^2$$
$$+ 4L^2(\eta^2\tau^2L^2 + 1/4)\lambda^2 d_s^3 + L^2\lambda^2 d_c^3, \tag{8}$$

---

[1] Here we slightly abuse the notation and denote $F(x_{s,m}^{t,i}) = F(x_{s,m}^{t,i}, h(x_{c,m}^t, \xi_m^t); y_m^t)$, where $y_m^t$ is the label corresponding to data $\xi_m^t$.

[2] The assumptions adopted in our analysis are standard and consistent with those commonly used in the distributed optimization literature [28–30]. We focus on the non-convex setting.

where $\mathcal{F} = \mathbb{E}[f(\mathbf{x}^0) - f(\mathbf{x}^T)]$; $d_c$ and $d_s$ represent the dimensions of the parameters on the client and server side, respectively; $d = d_c + d_s$ is the total number of parameters. $\lambda$ is the smooth parameters for ZO Oracle defined in (3), and $\sigma_c^2, \sigma_s^2$ are the upper bound of the gradient variance on client and server, respectively. $\eta = \eta_c/\tau = \eta_s$ is the unified learning rate.

To establish the theorem, the learning rate on server needs to shrink linearly with multiple update steps $\tau$, i.e. $\eta_c/\tau = \eta_s$. This requirement stems from the need to balance client and server progress: since the server performs $\tau$ updates for each client update, a proportionally smaller server learning rate ensures synchronized convergence. The convergence bound in equation (8) contains five distinct terms, each capturing different aspects of the algorithm's behavior.

The first term, $\frac{4\mathcal{F}}{\eta\tau T}$, represents the optimization error and decays as either the total number of communication rounds $T$ or the server-side update frequency $\tau$ increases. This rate matches the same rate as typical ZO-SGD methods when $\tau = 1$, which generalizes the classical convergence rate without unbalanced update. It also highlights the benefit of unbalanced updates: increasing the number of server iterations per round leads to a faster reduction of this term. *This demonstrates the improved convergence behavior enabled by unbalanced server updates.*

The second and third terms quantify the error introduced by the variance of the stochastic gradient estimates on the server and client, respectively. Notably, those two terms scales up with the parameter $\tau$. This means that a larger $\tau$ exacerbates the stochastic error, thus leading to high variance in the estimated gradient that hinders convergence performance. To keep these terms small, an inverse relationship between the Split training learning rate and server-side local steps should be satisfied, i.e., $\eta_s = \eta = \mathcal{O}(1/\sqrt{\tau})$. Specifically, note that both the server-side and client-side variances are linearly amplified by $\tau$. This requires a sufficiently small $\eta$ to offset the variance between two successive communication rounds to make the those error term in small. The intuitive explanation behind this is that when the server applies multiple consecutive updates using outdated client information, it introduces client drift and allows stochastic errors to accumulate progressively. Consequently, smaller step sizes are required to balance the impact of these accumulated error terms.

The last two terms, $4L^2(\eta^2\tau^2 L^2 + 1/4)\lambda^2 d_s^3 + L^2\lambda^2 d_c^3$ capture errors introduced by the zeroth-order gradient estimation. These terms are independent of the learning rate choice and decrease as the smoothing parameter $\lambda$ decreases, indicating that more accurate ZO gradient estimation improves overall convergence.

We can further derive a convergence rate for all terms if certain conditions are met.
**Corollary 4.2.** *Based on Theorem 4.1, let the model split satisfies $d_c = \sqrt{d/\tau}, d_s = d - \sqrt{d/\tau}$; let $\tau \leq d$, the smoothing parameter satisfies $\lambda^2 \leq \frac{1}{\sqrt{\tau T}d^{5/2}L}$, and choose the unified learning rate as $\eta \leq \min\{\frac{1}{64L(\tau+2d_s)}, \frac{1}{16L\tau d_c}, \frac{1}{\sqrt{d\tau T}}\}$. Then we have the following convergence rate:*

$$\frac{1}{T}\sum_{t=0}^{T}\mathbb{E}[\|\nabla_{\mathbf{x}}f(\mathbf{x}^t)\|^2] \leq \frac{4\sqrt{d}\mathcal{F}}{\sqrt{\tau T}} + \frac{48L\sqrt{d}\sigma_s^2}{\sqrt{\tau T}} + \frac{9\sqrt{d}}{\sqrt{\tau T}} + \frac{8L\sigma_c^2}{\sqrt{T}} \tag{9}$$

**Discussion.** All dominant terms in equation (9) converge at the rate of $\mathcal{O}(\sqrt{d/\tau T})$, when we choose $d_c = \sqrt{d/\tau}$ and $d_s = d - \sqrt{d/\tau}$, where $d = d_c + d_s$ is the total number of parameters. This rate highlights a linear speedup in term of $\tau$.[3] *The linear speedup is achieved when the client-side parameter dimension $d_c$ scales as $\mathcal{O}(d/\sqrt{\tau})$.* This has direct implications for network architecture design in split learning systems. In particular, when the server has higher computational capacity, it is beneficial to allocate fewer parameters to the client side, thereby placing the split closer to the input layer. That's being said, ZOO provides a natural mechanism for controlling stochastic variance through the cutting layer strategy. By connecting the cutting layer choice with multiple server updates steps $\tau$, the variance impact on the client side is effectively reduced. This variance reduction occurs because fewer layers are processed on the client side, which inherently limits the accumulation of gradient estimation errors. This theoretical finding aligns with our empirical observations in the ablation study presented in Section 5.

---

[3]To attain $\varepsilon$ accuracy for an algorithm, it needs $\mathcal{O}(\frac{1}{\varepsilon^2})$ communication rounds with a convergence rate $\mathcal{O}(\frac{1}{\sqrt{T}})$, while needing $\mathcal{O}(\frac{1}{\tau\varepsilon^2})$ rounds if the convergence rate is $\mathcal{O}(\frac{1}{\sqrt{\tau T}})$. In this sense, one achieves a linear speedup with respect to $\tau$.

## 4.2 Convergence Analysis for `MU-SplitFed`

We further derive the following convergence result for `MU-SplitFed` under SFL with $M$ clients. For the convergence analysis of `MU-SplitFed` under SFL, we further assume that the above two assumptions apply to $f_m$ for $\forall m \in [M]$. To quantify the data heterogeneity across clients, we make the following assumption on data distribution:

**Assumption 4.3** (Bounded Heterogeneity). *For $\forall m \in [M]$, the global variability of the local gradient is upper bounded: $\|\nabla f_m(x) - \nabla f(x)\|^2 \leq \epsilon^2$.*

**Theorem 4.3.** *Under Assumption 4.1 to 4.3, consider a SFL framework with $M$ clients, and let the server iteration number be $\tau$. If the learning rates on client and server satisfy $\eta_c/\tau = \eta_s = \eta \leq \min\{\frac{1}{120L\tau(1+2d_s/\tau)}, \frac{M}{12\tau Ld_c}\}$, the sequence of iterates generated by MU-Split satisfies:*

$$\frac{1}{T}\sum_{t=0}^{T}\mathbb{E}[\|\nabla_{\mathbf{x}}f(\mathbf{x}^t)\|^2] \leq \frac{4\mathcal{F}}{T\eta_g\eta\tau} + 16\eta(2\eta\tau L + \eta_g/M)Ld_s\sigma_s^2 + \frac{4\eta_g\eta\tau Ld_c\sigma_c^2}{M}$$
$$+ 24\eta(4\eta\tau L + \eta_g/M)L(\tau + 2d_s)\epsilon^2 + \frac{12\eta_g\eta\tau Ld_c\epsilon^2}{M}$$
$$+ (1/\tau + 8\eta^2\tau L^2 + 2\eta_g\eta/M)\tau L^2\lambda^2 d_s^3 + L^2\lambda^2 d_c^3 \qquad (10)$$

*where $\mathcal{F} = \mathbb{E}[f(\mathbf{x}^0) - f(\mathbf{x}^T)]$; $d_c$ and $d_s$ represent the dimensions of the parameters on the client-side and server-side, respectively; $\lambda$ is the smooth parameters for ZO Oracle defined in (3), and $\sigma_c^2, \sigma_s^2$ are the upper bound of the gradient variance on client and server. Additionally, $\eta_g$ is the global learning rate for model aggregation, and $\epsilon^2$ quantifies data heterogeneity.*

The first term and the last two terms are similar to `MU-Split`, which are attributed to model initialization and ZO optimization. Compared to traditional SFL, the presence of server iteration $\tau$ is again observed on the denominator, which corresponds to our observation in `MU-Split`: convergence is accelerated by multiple server updates. The second and third terms correspond to the variance of the stochastic gradient estimator on the server and client, respectively. Again, both terms scales with the increase of $\tau$, which is consistent with `MU-Split`. In contrast to the analysis in `MU-Split`, the fourth and fifth terms are newly introduced to account for data heterogeneity, and they are also observed in other Federated Learning literature. Notably, those two terms scales with the parameter $\tau$. This means that a larger $\tau$ exacerbates the heterogeneity error thus leading to increases client drift consequently. So, similar to SL, to offset the variance introduced by data heterogeneity and stochastic gradient estimation, a sufficiently small $\eta$ should be selected and decay linearly with $\tau$.

**Corollary 4.4.** *Based on Theorem 4.3, if we further ensure that the neural network is cut such that $d_c = \sqrt{d/\tau}, d_s = d - \sqrt{d/\tau}$; let $\tau \leq d$, let the smoothing parameter $\lambda^2 \leq \frac{1}{\sqrt{\tau T}d^{5/2}L}$, and choose learning rate as $\eta \leq \min\{\frac{1}{120L\tau(1+2d_s/\tau)}, \frac{M}{12\tau Ld_c}, \frac{1}{L\tau\sqrt{dT}}\}$, $\eta_g = \sqrt{\tau M}$. Define $\mathcal{F} = \mathbb{E}[f(\mathbf{x}^0) - f(\mathbf{x}^T)]$, and we have the following bound:*

$$\frac{1}{T}\sum_{t=0}^{T}\mathbb{E}[\|\nabla_{\mathbf{x}}f(\mathbf{x}^t)\|^2] \leq \frac{4L\sqrt{d}\mathcal{F}}{\sqrt{\tau TM}} + \frac{8\sqrt{d}(3\epsilon^2 + 2\sigma_s^2)}{\sqrt{\tau TM}} + \frac{32\sqrt{d}(3\epsilon^2 + \sigma_s^2)}{\tau T} + \frac{12\epsilon^2 + 4\sigma_c^2}{\sqrt{TM}} + \frac{6\sqrt{d}}{\tau T} \qquad (11)$$

**Discussion.** The first and second term converge at the rate $\mathcal{O}(\sqrt{d/(\tau TM)})$. Compared with `MU-Split`, the involvement of multiple clients $M$ accelerates convergence through the increased number of participating clients. This property is particularly desirable in the federated setting, where large-scale parallelism can be leveraged to speed up training. In contrast, the third and final terms do not benefit from parallelism across clients. Nevertheless, their impact is mitigated by the faster convergence rate with respect to $T$, which decays faster than the dominant terms. The fourth term, which captures client heterogeneity and gradient variance at the client side, does not contain the $\tau$ acceleration factor. This further confirms that multiple local updates contribute to the acceleration of initial error and variance introduced by the server, while the client side does not benefit from it. More importantly, while the server-side learning rate decrease with $\tau$, the global learning rate amplifies by $\tau$. The intuition behind this is as follows: as the server side uses stale information to update, a smaller learning rate ensures that each server update remains close to the original model, preventing large deviations. However, smaller learning rates reduce the cumulative gradient step at the server. To ensure a globally faster convergence rate, the global aggregation compensates for this by applying a slightly larger learning rate. Finally, the overall convergence rate is $\mathcal{O}(\sqrt{d/(\tau TM)})$, demonstrating that multiple local updates $\tau$ and multiple clients $M$ jointly accelerate convergence in SFL.

**Straggler resilient communication time.** The total communication time in SFL is largely determined by the straggler, as all other parties must wait for the slowest client to complete its computation before

Table 1: Test accuracy on four datasets. We run each method for 100 epochs on Fashion-MNIST and 500 epochs on the others, and report the resulting test accuracy at the final epoch.

| Dataset | GAS | Vanilla SplitFed/($\tau = 1$) | Ours($\tau = 2$) | Ours($\tau = 3$) | Ours($\tau = 4$) |
|---------|-----|------------------------------|-------------------|-------------------|-------------------|
| CIFAR-10 | 75.28 | 69.73 | **77.86** | 73.20 | 69.40 |
| Fashion-MNIST | 83.70 | 77.50 | **85.45** | 85.28 | 84.47 |
| CINIC-10 | 57.80 | 51.96 | **59.50** | 55.75 | 52.43 |
| CIFAR-100 | 25.33 | 16.58 | **32.16** | 24.64 | 22.38 |

proceeding to the next communication round. We first define three terms for further explanation: 1) $t_{\text{straggler}}$ denotes the time delay of the straggler, 2) $T_0$ represents the number of communication rounds required for convergence, and 3) $t_{\text{server}}$ as the server-side computation time for one local update. In parallel SFL settings, the required total delay caused by straggler can be represented as $T_0 \cdot t_{\text{straggler}}$, which mainly depends on the straggler and results in slow and unstable convergence.

In contrast, with unbalanced updates, if we let the server perform $\tau = t_{\text{straggler}}/t_{\text{server}}$ local iterations during each round. According to Corollary 4.4, this reduces the total number of communication rounds from $T_0$ to $T_1 = T_0/\tau$. Consequently, the total communication time becomes:

$$T_1 \cdot t_{\text{straggler}} = T_0 \cdot t_{\text{straggler}}/\tau = T_0 \cdot t_{\text{server}}, \tag{12}$$

which is now *independent of the straggler time*. This result highlights a key advantage of `MU-SplitFed`: by appropriately choosing $\tau$, the system can effectively decouple overall training time from the performance of the slowest client.

## 5 Experiments

**Experimental Setup.** To evaluate the effectiveness of `MU-SplitFed`, we conduct experiments on four image classification benchmarks: Fashion-MNIST [31], CINIC-10 [32], CIFAR-10, and CIFAR-100 [33]. All experiments are carried out on a node with 3 NVIDIA A100 40GB GPUs. The model cut layer is denoted as $L_c$, where $L_c = n$ means the model is split after the $n$-th block. For these tasks, we adopt the AlexNet architecture, assessing the framework's ability to mitigate the impact of stragglers. As AlexNet contains only 8 layers, it offers limited flexibility in exploring different splitting configurations. To further analyze the role of the unbalanced update ratio $\tau$ in controlling communication round, we extend our study to a large language model (LLM), OPT-1.3B [34], which has 24 transformer blocks and enables a broader range of splitting strategies. We evaluate its performance on the SST-2 dataset [35], a binary sentiment classification task, to examine the applicability of `MU-SplitFed` in the LLM domain.

We compare `MU-SplitFed` with vanilla SplitFed and GAS [8], a recent SFL method that addresses stragglers via asynchronous updates. Vanilla SplitFed serves as a baseline without straggler mitigation strategy. To simulate the device heterogeneity, we follow the simulation design of [8, 12]. In particular, we sample the computation time from an exponential distribution to represent different computation capacities across different clients. In our experiment, we train 10 clients in total with 50% partial partitioning for each global aggregation. For a fairness comparison, we modify both vanilla SplitFed and GAS to use ZO optimization, aligning them with `MU-SplitFed`'s gradient-free design. Additionally, we evaluate the convergence performance w.r.t to time unit of our simulation, providing a direct measure of each method's performance to straggler-induced delays.

**Impact of $\tau$ Selection.** First, we investigate how the choice of $\tau$ impacts the performance of our proposed `MU-SplitFed`. We compare the accuracy from the same global communication round across different methods: we pull the result of the 100th epoch for Fashion-MNIST, and choose the 500th epoch result for the rest three datasets. As shown in Table 1, we compare the training accuracy with different values of the server iterations $\tau \in \{2, 3, 4\}$. Our method achieves the highest accuracy when $\tau = 2$, demonstrating its effectiveness in reducing communication round. However, increasing $\tau$ over 2 leads to a noticeable drop in accuracy. This observation aligns with our theoretical insights. Specifically, Corollary 4.2 suggests that the choice of $\tau$ is related to the parameter size of the client-side submodel $d_c = \sqrt{\frac{d}{\tau}}$, which is governed by the cut layer $L_c$. Given the structure of AlexNet, $L_c = 2$ is the only split type satisfied this setting without violating the constraint $L_c \geq 1$. Thereby, $\tau = 2$ corresponds to the optimal choice of server steps given fixed cutting strategy. Consequently, as $\tau$ exceeds this value, the mismatch between $\tau$ and splitting strategy contributed to the observed accuracy drop. Based on this insight, we use $\tau = 2$ for our method in the next experiment.

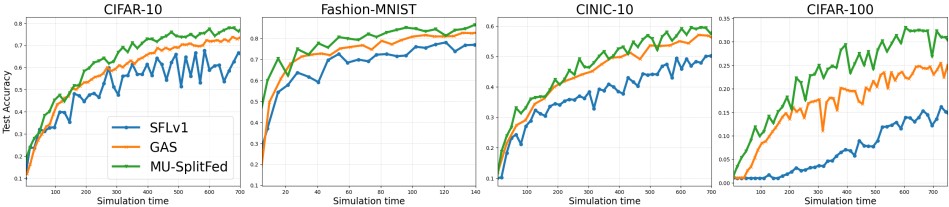

Figure 2: Performance Under Stragglers, where we set $\tau = 2$ for `MU-SplitFed`.

**Performance under Straggler.** In this subsection, we evaluate the resilience of `MU-SplitFed` to straggler effects by comparing its convergence performance against baseline methods on four datasets. Here, we introduce random delays following an exponential distribution to emulate straggler-induced latency. Figure 2 presents the accuracy over wall-clock time for all methods. Across all tasks, `MU-SplitFed` consistently achieves higher accuracy and in less time compared to both `vanilla SplitFed` and `GAS`, highlighting its efficiency in mitigating straggler-induced delays. Notably, on both CIFAR-10 and more complex task CIFAR-100, `MU-SplitFed` *maintains a fast and stable convergence trend, while* `GAS` *exhibits slower convergence and less consistency*. One possible reason for these scenarios is that `GAS` supports asynchronous updates, its activation generation step scales poorly with the increasing size of the label, introducing significant computational overhead that limits its efficiency in straggler-prone settings. In contrast, `MU-SplitFed` *maintains lightweight computation on both server and client sides, which allows efficient parallelization and better utilization of system resources during straggler delays.*

**Interaction Between Cut Layer and Server Iterations.** To fully explore the potential of our proposed unbalanced update in reducing the communication round, we fine-tune the OPT-1.3B that enables more types of model splitting. This allows us to more thoroughly explore how to jointly select $\tau$ and cut layer $L_c$ to optimize communication efficiency. Figure 3 shows the total communication round required to attain 85% accuracy across different cut layers and values of $\tau$. For a fixed cut layer (e.g. $L_c = 4$), increasing $\tau$ reduces communication round by up to 33%, confirming the benefit of unbalanced updates.

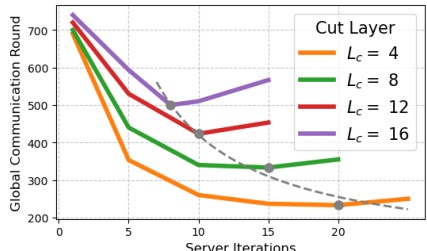

Figure 3: Interaction between cut layer $L_c$ and server iteration $\tau$.

More interestingly, there a clear trade-off emerging between $\tau$ and $L_c$. When $L_c$ is fixed, increasing $\tau$ initially improves convergence, but excessive server updates eventually lead to diminishing or adverse effects. Conversely, when fixing $\tau$ and tuning the cut layer, convergence consistently improves as $L_c$ decreases, *indicating a deeper server-side model is beneficial for model performance.* Moreover, the optimal value of $\tau$ shifts higher as $L_c$ moves earlier in the model. These trends confirm our theoretical insight in Remark 4.1: *to fully exploit server-side acceleration, the model partition must scale with the number of server iterations.* The dashed gray curve

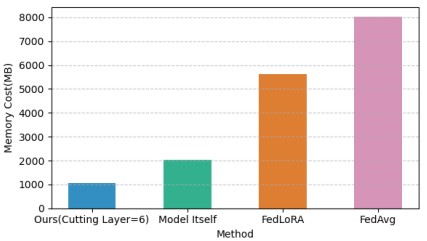

Figure 4: Comparison of peak memory cost for different methods for fine-tuning LLM.

illustrates this joint optimization trajectory, highlighting that coordinated tuning of $L_c$ and $\tau$ yields the most communication-efficient convergence.

**Memory Efficiency.** To evaluate the memory efficiency of our ZO-based framework in the context of LLM fine-tuning, we compare the peak memory usage on the client side. Specifically, we compare our proposed `MU-SplitFed` with FedAvg [4] and FedAvg with LoRA [36] (FedLoRA) for fine-tuning the OPT-1.3B model on the SST-2 dataset. As illustrated in Figure. 4, FedAvg incurs a peak memory cost of 8.02 GB on the client. FedLoRA, which reduces memory usage by updating only low-rank adapter matrices, reduces this to 5.64 GB. Despite these improvements, both FedAvg and FedLoRA still require substantial memory to store gradients and maintain the full model locally. In contrast,

`MU-SplitFed` reduces the peak client-side memory footprint to just 1.05 GB. This is achieved by storing only a partial model on the client and leveraging ZO optimization, which eliminates the need to store gradient information during training, further contributing to its memory efficiency.

## 6    Related Work

**Split Federated Learning.** SFL [1] is a powerful distributed learning framework that enables scalable training across resource-constrained edge devices. By model partitioning on the client side without sharing raw data with the server, SFL provides a memory-efficient and privacy-preserving solution for resource-constrained devices. Recent advances in SFL have addressed key challenges from different perspectives. To mitigate the communication bottleneck, Chen et al. [37] reduces communication frequency by proposing a loss threshold that determines when to exchange information between client and Split Server. Han et al. [3] employ different local loss functions on the client and server sides, thus reducing the gradient information transmission rounds. Other approaches apply quantization or sparsification techniques to reduce communication costs in each transaction round. For instance, [38] leverages Top-S sparsification for both forward embedding and backward gradient transmissions, while [39] introduces randomness for further enhancement. FedLite applies Top-K quantization to compress intermediate features [40]. For privacy purposes, several methods tackle model inversion attacks. ResSFL [41] and NoPeek [42] achieve attacker-aware training by integrating inversion score regularization term. Moreover, other strategies apply differential privacy on intermediate embedding features to provide privacy guarantees against label leakage [43]. In heterogeneous settings, methods like SCALA [44] and GAS [8] introduce activation concatenation and centralized training to enhance robustness and accommodate for varying client capabilities. However, theoretical research for SFL is still insufficient. [45] provides the first convergence analysis for sequential SFL, while [2] proposes an efficient update mechanism using different synchronization frequencies on client and server with rigorous convergence analysis for both sequential and parallel SFL.

**Existing Straggler Solutions.** The straggler issue in FL has been well explored, with asynchronous updates emerging as one of the most promising directions [10]. Yet, asynchronous methods rely on stale information to update, which can lead to performance degradation due to outdated or inconsistent model information. To address this, ASO-Fed [23] proposed a dynamic learning rate adjustment mechanism tailored to each client's training progress to reduce the staleness effect from straggler. FedBuff [26] enables efficient training by using a buffer to store information from faster clients. Based on that, CA2FL [27] enhances the performance on heterogeneous data by adaptively adjusting model updates based on data property. Similarly, FedCompass [46] adopts a resource-aware scheduling policy that prioritizes clients with high computation capacity, thus mitigating the impact of stragglers. FedASMU [47] employs dynamic model aggregation with adaptive model adjustment to mitigate the impact of stragglers. Yet, existing strategies regarding the straggler in SFL remain limited. [9, 16] reduce the time delay by employing adaptive splitting strategies to balance the arrival times of activations. GAS [8] propose an asynchronous SFL framework that utilizes an activation buffer to generate activations based on the degree of bias, thereby enhancing the robustness of the algorithm.

## 7    Conclusion and Limitations

We propose `MU-SplitFed`, a simple and effective framework for mitigating the straggler problem in Split Federated Learning by introducing unbalanced updates on server-side. The simple yet efficient unbalanced update strategy enables faster training by reducing communication complexity, thereby mitigating delays caused by stragglers. Notably, both our theory and experiments show that increasing the unbalanced update ratio $\tau$ yields a linear reduction in communication frequency. When $\tau = t_{straggler}/t_{server}$, the total training time becomes independent of the straggler delay. Moreover, our analysis uncovers a key connection between the choice of the splitting layer and the optimal $\tau$, offering practical guidance for further system design. These findings suggest that `MU-SplitFed` is a promising solution for enabling scalable and efficient training on resource-constrained edge devices.

Our work also highlights the potential of applying SFL for fine-tuning task of LLM, where memory efficiency is an impetus need. In LLM setting, SFL offers a natural fit: edge or local servers can serve as client-side device, while high-performance cloud servers act as the central server. Although our framework demonstrates initial promise in this direction by solving the bottleneck in this realm, how to fully realize the benefits of SFL for scalable LLM fine-tuning remains an open challenge and needs further investigation.

## Acknowledgement

Research reported in this publication was supported by the National Institute Of General Medical Sciences of the National Institutes of Health under Award Numbers R16GM159671 and 1R35GM156653, and the National Science Foundation under Award Number 2045804. The content is solely the responsibility of the authors and does not necessarily represent the official views of the National Institutes of Health.

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

# Appendix

# A  Communication Benefits of Unbalanced Update

## A.1  Dimension-Free ZOO achieved by Unbalanced Updates

As shown in Table 2, the proposed `MU-SplitFed` can further achieve dimension-free ZOO with convergence rate $\mathcal{O}(1/\sqrt{T})$, when $\tau \to d$. By appropriately scaling the unbalanced update factor $\tau$ to match the model dimension $d$, the convergence rate becomes independent of $d$. This is particularly significant for ZOO, where the parameter dimension $d$ often dominates the denominator of the convergence rate and thus slows down training as the model size grows. Large models exacerbate this issue because the increased $d$ in the denominator hinders convergence and adds communication overhead. `MU-SplitFed` mitigates this by exploiting unbalanced updates, which not only accelerates ZOO training but also reduces communication costs. Specifically, the convergence rate improves from $\mathcal{O}(\sqrt{d/T})$ to $\mathcal{O}(1/\sqrt{T})$, meanwhile, the communication complexity reduces from $\mathcal{O}(d/\epsilon^2)$ to $\mathcal{O}(1/\epsilon^2)$. Compared with other dimension-free methods [17, 18], which often rely on strong assumptions, e.g. low-rank assumption, that are impractical in real-world scenarios, `MU-SplitFed` provides a more flexible way towards this end. By introducing unbalanced updates into ZOO, we effectively remove the dependency on $d$ without imposing additional assumptions, making the method significantly more feasible in practice.

## A.2  Comparable Analysis

We analyze the communication costs of `MU-SplitFed` under different choices of $\tau$ and compare against two existing theoretical baselines for SFL frameworks. SFL-V1, introduced in [2], serves as the fundamental baseline for parallel SFL architectures using first-order optimization. Reference [45] provides rigorous convergence analysis with the perspective of SFL in a sequential update manner. To systematically validate the benefits of unbalanced updates, we present results across different $\tau$ configurations: $\tau = 1$ represents the balanced update scenario where client and server updates with equal frequency, providing insight into combining ZOO with traditional SFL; $\tau > 1$ corresponds to our proposed unbalanced update strategy; and $\tau \to d$ is the optimal case that $\tau$ scales to the same order of dimensionality $d$. That being said, the convergence rate is no longer dependent on $d$, thus achieving the dimension-free convergence rate.

**Communication Advantage of Unbalanced Updates.** Compared to balanced SFL with ZOO ($\tau = 1$), our unbalanced update strategy ($\tau > 1$) demonstrates linear convergence acceleration with respect to $\tau$. This improvement translates directly to communication complexity, where $\tau$ provides linear communication cost reduction from $\mathcal{O}(d/M\epsilon^2)$ to $\mathcal{O}(d/\tau M\epsilon^2)$. Specifically, unbalanced updates reduce total communication overhead by decreasing the number of communication rounds required for convergence. When $\tau \to d$, we achieve a convergence rate of $\mathcal{O}(\sqrt{1/TM})$ that eliminates dependence on dimensionality $d$, resulting in dimension-free communication complexity of $\mathcal{O}(1/M\epsilon^2)$.

**Comparison with SFL-V1.** To the best of our knowledge, SFL-V1 [2] provides the first theoretical analysis for parallel SFL under bounded gradient, non-convex, and non-iid assumptions. However, their theoretical results exhibit no acceleration with respect to either the number of clients $M$ or local update steps. In contrast, our convergence rate demonstrates faster convergence as both the number of clients $M$ increase under the more loose assumption, e.g. bounded variance, consequently requiring fewer communication rounds to reach an $\epsilon$-approximation solution.

**Comparison with SFL-V2.** Our method achieves comparable convergence rates to SFL-V2 [45], where $K$ is the number of local updates. While multiple local updates $K$ accelerate convergence

| Method | Convergence Rate | SplitServer Comm. Cost | Assumptions |
|---|---|---|---|
| SFL-V1 [2] | $\mathcal{O}(1/\sqrt{T})$ | $\mathcal{O}(K/\epsilon^2)$ | b.g./N.C./non-iid |
| SFL-V2 [45] | $\mathcal{O}(1/\sqrt{TMK}))$ | $\mathcal{O}(K/M\epsilon^2)$ | b.v./N.C/non-iid |
| MU-SplitFed ($\tau = 1$) | $\mathcal{O}(\sqrt{d/TM})$ | $\mathcal{O}(d/M\epsilon^2)$ | |
| MU-SplitFed ($\tau > 1$) | $\mathcal{O}(\sqrt{d/\tau TM})$ | $\mathcal{O}(d/\tau M\epsilon^2)$ | b.v./N.C/non-iid |
| MU-SplitFed ($\tau \to d$) | $\mathcal{O}(\sqrt{1/TM})$ | $\mathcal{O}(1/M\epsilon^2)$ | |

Table 2: Comparison of Communication Complexity

in FL settings by reducing communication frequency, they impose additional communication costs when applied to SFL architectures. As demonstrated in Table 2, increasing local updates $K$ actually increases the total communication cost for convergence in the SFL setting. This counterintuitive result stems from the relay-based update mechanism inherent in SFL, where local updates exacerbate communication overhead between clients and servers rather than reducing it. Conversely, our unbalanced update parameter $\tau$ facilitates convergence without requiring additional communication rounds, achieving linear communication cost reduction with respect to $\tau$. This fundamental architectural advantage establishes the superior communication efficiency of our unbalanced update strategy over existing theoretical result.

## B  Preliminaries

### B.1  Notations

Table 3: Notations in this paper

| Notation | Meaning |
|----------|---------|
| $d$ | Total model parameter dimension |
| $m, M$ | Index, total number of clients |
| $t, T$ | Index, total number of communication round |
| $p, P$ | Index, total number of perturbations |
| $i, \tau$ | Index, total number of server iterations |
| $\mathbf{x}^t$ | Global model parameters in the $t$-th round |
| $x_c^t$ | Client-side model parameters in the $t$-th round |
| $x_s^{t,i}$ | Server-side model parameters in the $i$-th iteration |
| $\xi_m^t$ | Data sample in the $t$-th round for $m$-th client |
| $g_{c,p}^{t,i}$ | Stochastic Zeroth-order gradient for $t$-th round |
| $g_{s,p}^{t,i}$ | Stochastic Zeroth-order gradient for $i$-th iteration |
| $G_{c,m}^t$ | Zeroth-order gradient estimator for client |
| $G_{s,m}^{t,i}$ | Zeroth-order gradient estimator for server |
| $f_m(\cdot)$ | Local loss function for client $m$ |
| $f(\cdot)$ | Global loss function for SL or SFL |

### B.2  Assumptions

**Assumption B.1** ($L$-Smooth). *For $\forall m \in [M]$, the loss function $f_m$ is bounded from below, and is $L$-smooth, i.e. $\forall x, y, \| \nabla f_m(x) - \nabla f_m(y)\| \leq L\|x - y\|$.*

**Assumption B.2** (Bounded variance). *For $\forall m \in [M]$, the variance of the stochastic gradient w.r.t. the client and the server is upper-bounded by $\sigma_c^2$ and $\sigma_s^2$. Specifically, for $\forall \xi \in \mathcal{D}_m$,*

$$\|\nabla_{x_c} F_m(x; \xi) - \nabla f_m(x)\|^2 \leq \sigma_c^2$$

$$\|\nabla_{x_s} F_m(x; \xi) - \nabla f_m(x)\|^2 \leq \sigma_s^2$$

**Assumption B.3** (Bounded Heterogeneity). *For $\forall m \in [M]$, the global variability of the local gradient is upper bounded:*

$$\|\nabla f_m(x) - \nabla f(x)\|^2 \leq \epsilon^2$$

### B.3  Technical Lemmas

**Lemma B.1.** *Let $g(x)$ be defined as in* (3). *We define the smoothed function $f_\lambda(x) = \mathbb{E}_v[f(x + \lambda v)]$, where $v$ is uniformly sampled from the Euclidean ball $\sqrt{d}\mathbb{B}^d = \{x \in \mathbb{R}^d \mid \|x\| \leq \sqrt{d}\}$. The following properties hold:*

   *(i)  $f_\lambda(x)$ is differentiable and $\mathbb{E}_u[g_\lambda(x)] = \nabla f_\lambda(x)$.*

   *(ii)  If $f(x)$ is $L$-smooth, then we have that*

$$\|\nabla f(x) - \nabla f_\lambda(x)\| \leq \frac{L}{2}\lambda d^{3/2}, \tag{13}$$

*and*

$$\mathbb{E}_u[\|g_\lambda(x)\|^2] \le 2d \cdot \|\nabla f(x)\|^2 + \frac{L^2}{2}\lambda^2 d^3. \tag{14}$$

**Remark B.1.** By (13) we immediately have

$$\|\nabla f_\lambda(x)\|^2 \le 2\|\nabla f(x)\|^2 + \frac{L^2}{2}\lambda^2 d^3 \tag{15}$$

$$\|\nabla f(x)\|^2 \le 2\|\nabla f_\lambda(x)\|^2 + \frac{L^2}{2}\lambda^2 d^3 \tag{16}$$

The dual-paced model aggregation and model update in SFL presents more challenge in convergence analysis compared to the analysis in traditional FL setting. To address this problem, we decompose the convergence analysis into client-side and server-side, respectively. The following lemma reveals this relationship.

**Lemma B.2** (Decomposition). *Let $\mathbf{x}^t \equiv [x_c^t; x_s^t]$ denote the global model at the $t$th training rounds. By applying Assumption B.1, we have:*

$$\mathbb{E}[f(\mathbf{x}^{t+1}) - f(\mathbf{x}^t)]$$

$$\le \mathbb{E}[\langle \nabla_{\mathbf{x}} f(\mathbf{x}^t), \mathbf{x}^{t+1} - \mathbf{x}^t \rangle] + \frac{L}{2}\|\mathbf{x}^{t+1} - \mathbf{x}^t\|^2$$

$$\le \underbrace{\mathbb{E}[\langle \nabla_{x_s} f(\mathbf{x}^t), x_s^{t+1} - x_s^t \rangle]}_{\mathcal{K}_1} + \underbrace{\frac{L}{2}\mathbb{E}[\|x_s^{t+1} - x_s^t\|^2]}_{\mathcal{K}_2} + \underbrace{\mathbb{E}[\langle \nabla_{x_c} f(\mathbf{x}^t), x_c^{t+1} - x_c^t \rangle]}_{\mathcal{K}_3} + \underbrace{\frac{L}{2}\mathbb{E}[\|x_c^{t+1} - x_c^t\|^2]}_{\mathcal{K}_4}$$

$$\tag{17}$$

## C  Proof For MU-Split

### C.1  Proof of main theorem

We now prove the main theorem of MU-Split, and defer all important lemmas to Appendix C.2. We first restate the main theorem below.

**Theorem C.1.** *Under Assumption B.1 and B.2, and let the server iteration number be $\tau$. If the learning rates satisfy $\eta_c/\tau = \eta_s = \eta \le \min\{\frac{1}{64L(\tau+2d_s)}, \frac{1}{16L\tau d_c}\}$, the sequence of iterates generated by MU-Split satisfies:*

$$\frac{1}{T}\sum_{t=0}^T \mathbb{E}[\|\nabla_{\mathbf{x}} f(\mathbf{x}^t)\|^2] \le \frac{4}{\eta\tau T}\mathbb{E}[f(\mathbf{x}^0) - f(\mathbf{x}^T)] + 16\eta L(\eta\tau L + 1)d_s\sigma_s^2$$

$$+ 8\eta\tau L d_c\sigma_c^2 + 4L^2(\eta^2\tau^2 L^2 + 1/4)\lambda^2 d_s^3 + L^2\lambda^2 d_c^3, \tag{18}$$

#### C.1.1  One-Round Update on Server Side

For $\mathcal{K}_1$:

$$\mathbb{E}[\langle \nabla_{x_s} f(\mathbf{x}^t), x_s^{t+1} - x_s^t \rangle]$$

$$= \mathbb{E}[\langle \nabla_{x_s} f(\mathbf{x}^t), -\sum_{i=0}^{\tau-1} \eta_s G_s(\mathbf{x}^{t,i}; \xi^t) \rangle]$$

$$= \mathbb{E}[\langle \nabla_{x_s} f(\mathbf{x}^t), -\sum_{i=0}^{\tau-1} \eta_s \left( \nabla_{x_s} f_\lambda^{t,i} - \nabla_{x_s} f(\mathbf{x}^t) + \nabla_{x_s} f(\mathbf{x}^t) \right) \rangle]$$

$$= \mathbb{E}[\langle \sqrt{\eta_s\tau}\nabla_{x_s} f(\mathbf{x}^t), -\frac{\sqrt{\eta_s}}{\sqrt{\tau}}\sum_{i=0}^{\tau-1} \left( \nabla_{x_s} f_\lambda^{t,i} - \nabla_{x_s} f^t \right) \rangle] - \eta_s\tau\mathbb{E}[\|\nabla_{x_s} f(\mathbf{x}^t)\|^2]$$

$$= \frac{\eta_s\tau}{2}\mathbb{E}[\|\nabla_{x_s} f(\mathbf{x}^t)\|^2] + \frac{\eta_s}{2\tau}\mathbb{E}\left\|\sum_{i=0}^{\tau-1} \left( \nabla_{x_s} f_\lambda^{t,i} - \nabla_{x_s} f^t \right)\right\|^2$$

$$- \frac{\eta_s}{2\tau}\mathbb{E}\left\|\sum_{i=0}^{\tau-1} \nabla_{x_s} f_\lambda^{t,i}\right\|^2 - \eta_s\tau\mathbb{E}[\|\nabla_{x_s} f(\mathbf{x}^t)\|^2]$$

$$= -\frac{\eta_s \tau}{2}\mathbb{E}[\|\nabla_{x_s} f(\mathbf{x}^t)\|^2] + \frac{\eta_s}{2\tau}\mathbb{E}\left\|\sum_{i=0}^{\tau-1}\left(\nabla_{x_s} f_\lambda^{t,i} - \nabla_{x_s} f^{t,i} + \nabla_{x_s} f^{t,i} - \nabla_{x_s} f^t\right)\right\|^2$$

$$-\frac{\eta_s}{2\tau}\mathbb{E}\left\|\sum_{i=0}^{\tau-1}\nabla_{x_s} f_\lambda^{t,i}\right\|^2$$

$$\leq -\frac{\eta_s \tau}{2}\mathbb{E}[\|\nabla_{x_s} f(\mathbf{x}^t)\|^2] + \frac{\eta_s}{\tau}\mathbb{E}\left\|\sum_{i=0}^{\tau-1}(\nabla_{x_s} f_\lambda^{t,i} - \nabla_{x_s} f^{t,i})\right\|^2 + \frac{\eta_s}{\tau}\mathbb{E}\left\|\sum_{i=0}^{\tau-1}\left(\nabla_{x_s} f_m^{t,i} - \nabla_{x_s} f_m^t\right)\right\|^2$$

$$-\frac{\eta_s}{2\tau}\mathbb{E}\left\|\sum_{i=0}^{\tau-1}\nabla_{x_s} f_\lambda^{t,i}\right\|^2$$

$$\leq -\frac{\eta_s \tau}{2}\mathbb{E}[\|\nabla_{x_s} f(\mathbf{x}^t)\|^2] + \eta_s\sum_{i=0}^{\tau-1}\mathbb{E}\left\|\nabla_{x_s} f_\lambda^{t,i} - \nabla_{x_s} f^{t,i}\right\|^2$$

$$+ \eta_s\sum_{i=0}^{\tau-1}\mathbb{E}\left\|\nabla_{x_s} f^{t,i} - \nabla_{x_s} f^t\right\|^2 - \frac{\eta_s}{2\tau}\mathbb{E}\left\|\sum_{i=0}^{\tau-1}\nabla_{x_s} f_\lambda^{t,i}\right\|^2$$

$$\leq -\frac{\eta_s \tau}{2}\mathbb{E}[\|\nabla_{x_s} f(\mathbf{x}^t)\|^2] + \frac{\eta_s}{4}\tau L^2\lambda^2 d_s^3 + \eta_s L^2\underbrace{\sum_{i=0}^{\tau-1}\mathbb{E}\left\|x_s^{t,i} - x_s^t\right\|^2}_{\mathcal{A}_1} - \frac{\eta_s}{2\tau}\mathbb{E}\left\|\sum_{i=0}^{\tau-1}\nabla_{x_s} f_\lambda^{t,i}\right\|^2$$

$$\leq -\frac{\eta_s \tau}{2}\mathbb{E}[\|\nabla_{x_s} f(\mathbf{x}^t)\|^2] + \frac{\eta_s}{4}\tau L^2\lambda^2 d_s^3 - \frac{\eta_s}{2\tau}\mathbb{E}\left\|\sum_{i=0}^{\tau-1}\nabla_{x_s} f_\lambda^{t,i}\right\|^2$$

$$2\eta_s L^2\left(8\eta_s^2(\tau^3 + \tau^2 d_s/P)\mathbb{E}[\|\nabla_{x_s} f(\mathbf{x}^t)\|^2] + 4\eta_s^2\tau^2 d_s\sigma_s^2/P + \eta_s^2\tau^3 L^2\lambda^2 d_s^3\right)$$

$$= \left(16\eta_s^3 L^2(\tau^3 + \tau^2 d_s/P) - \frac{\eta_s \tau}{2}\right)\mathbb{E}[\|\nabla_{x_s} f(\mathbf{x}^t)\|^2] + \frac{\eta_s}{4}\tau L^2\lambda^2 d_s^3$$

$$+ \frac{8\eta_s^3\tau^2 L^2 d_s\sigma_s^2}{P} + 2\eta_s^3\tau^3 L^4\lambda^2 d_s^3 - \frac{\eta_s}{2\tau}\mathbb{E}\left\|\sum_{i=0}^{\tau-1}\nabla_{x_s} f_\lambda^{t,i}\right\|^2 \tag{19}$$

where we apply $L$-smooth, Lemma B.1, $\langle a, b\rangle \leq \frac{\|a\|^2 + \|b\|^2}{2}$ and $\|a+b\|^2 \leq 2(\|a\|^2 + \|b\|^2)$, and substitute Lemma C.4 into $\mathcal{A}_1$.

For $\mathcal{K}_2$:

$$\mathbb{E}[\|x_s^{t+1} - x_s^t\|^2] = \eta_s^2\mathbb{E}[\|\sum_{i=0}^{\tau-1} G_s(\mathbf{x}^{t,i};\xi^t)\|^2]$$

By (31):

$$\mathbb{E}[\|\sum_{i=0}^{\tau-1} G_s(\mathbf{x}^{t,i};\xi^t)\|^2]$$

$$\leq 2\mathbb{E}[\|\sum_{i=0}^{\tau-1}\nabla_{x_s} f_\lambda(\mathbf{x}^{t,i})\|^2] + 2\sum_{i=0}^{\tau-1}\mathbb{E}[\|G_s(\mathbf{x}^{t,i};\xi^t) - \nabla_{x_s} f_\lambda(\mathbf{x}^{t,i})\|^2$$

Similar to the proof in C.4, we substitute in (25) and (30) in order:

$$\sum_{i=0}^{\tau-1}\mathbb{E}[\|G_s(\mathbf{x}^{t,i};\xi^t) - \nabla_{x_s} f_\lambda(\mathbf{x}^{t,i})\|^2$$

$$\leq \frac{-1}{P}\sum_{i=0}^{\tau-1}\mathbb{E}[\|\nabla_{x_s} f_\lambda^{t,i}\|^2]$$

$$+ \sum_{i=0}^{\tau-1} \left( \frac{4d_s}{P} \mathbb{E}[\|\nabla_{x_s} f(\mathbf{x}^t)\|^2] + \frac{4L^2 d_s}{P} \mathbb{E}[\|x_s^{t,i} - x_s^t\|^2] + \frac{2d_s \sigma_s^2}{P} + \frac{L^2 \lambda^2 d_s^3}{2P} \right)$$

$$\leq \frac{-1}{P} \sum_{i=0}^{\tau-1} \left( 4\mathbb{E}[\|\nabla_{x_s} f(\mathbf{x}^t)\|^2] + 4L^2 \mathbb{E}[\|x_s^{t,i} - x_s^t\|^2] + \frac{L^2}{2} \lambda^2 d_s^3 \right)$$

$$+ \sum_{i=0}^{\tau-1} \left( \frac{4d_s}{P} \mathbb{E}[\|\nabla_{x_s} f(\mathbf{x}^t)\|^2] + \frac{4L^2 d_s}{P} \mathbb{E}[\|x_s^{t,i} - x_s^t\|^2] + \frac{2d_s \sigma_s^2}{P} + \frac{L^2 \lambda^2 d_s^3}{2P} \right)$$

$$\leq \frac{4\tau d_s}{P} \mathbb{E}[\|\nabla_{x_s} f(\mathbf{x}^t)\|^2] + \frac{4\tau L^2 d_s}{P} \sum_{i=0}^{\tau-1} \mathbb{E}[\|x_s^{t,i} - x_s^t\|^2] + \frac{2\tau d_s \sigma_s^2}{P}$$

So

$$\mathbb{E}[\|x_s^{t+1} - x_s^t\|^2] \leq \frac{8\eta_s^2 \tau d_s}{P} \mathbb{E}[\|\nabla_{x_s} f(\mathbf{x}^t)\|^2] + \frac{8\eta_s^2 \tau L^2 d_s}{P} \sum_{i=0}^{\tau-1} \mathbb{E}[\|x_s^{t,i} - x_s^t\|^2]$$

$$+ \frac{4\eta_s^2 \tau d_s \sigma_s^2}{P} + 2\eta_s^2 \mathbb{E}[\|\sum_{i=0}^{\tau-1} \nabla_{x_s} f_\lambda(\mathbf{x}^{t,i})\|^2]$$

$$(1 - \frac{8\eta_s^2 \tau L^2 d_s}{P}) \mathbb{E}[\|x_s^{t+1} - x_s^t\|^2] \leq \frac{8\eta_s^2 \tau d_s}{P} \mathbb{E}[\|\nabla_{x_s} f(\mathbf{x}^t)\|^2] + \frac{4\eta_s^2 \tau d_s \sigma_s^2}{P}$$

$$+ 2\eta_s^2 \mathbb{E}[\|\sum_{i=0}^{\tau-1} \nabla_{x_s} f_\lambda(\mathbf{x}^{t,i})\|^2]$$

Further assume that $\eta_s \leq \frac{1}{4L\sqrt{\tau d_s / P}}$, we have

$$\mathbb{E}[\|x_s^{t+1} - x_s^t\|^2] \leq \frac{16\eta_s^2 \tau d_s}{P} \mathbb{E}[\|\nabla_{x_s} f(\mathbf{x}^t)\|^2] + \frac{8\eta_s^2 \tau d_s \sigma_s^2}{P} + 4\eta_s^2 \mathbb{E}[\|\sum_{i=0}^{\tau-1} \nabla_{x_s} f_\lambda(\mathbf{x}^{t,i})\|^2] \quad (20)$$

### C.1.2  One-Round Update on Client Side

For $\mathcal{K}_3$, we have

$$\mathbb{E}\langle \nabla_{x_c} f(\mathbf{x}^t), x_c^{t+1} - x_c^t \rangle$$
$$= \mathbb{E}\langle \nabla_{x_c} f(\mathbf{x}^t), \eta_c G_c(\mathbf{x}^t; \xi^t) \rangle$$
$$= \mathbb{E}\langle \nabla_{x_c} f(\mathbf{x}^t), -\eta_c \left( \nabla_{x_c} f_\lambda^t - \nabla_{x_c} f(\mathbf{x}^t) + \nabla_{x_c} f(\mathbf{x}^t) \right) \rangle$$
$$= \mathbb{E}\langle \sqrt{\eta_c} \nabla_{x_c} f(\mathbf{x}^t), -\sqrt{\eta_c} \left( \nabla_{x_c} f_\lambda^t - \nabla_{x_c} f^t \right) \rangle - \eta_c \mathbb{E}\|\nabla_{x_c} f(\mathbf{x}^t)\|^2$$
$$= \frac{\eta_c}{2} \mathbb{E}[\|\nabla_{x_c} f(\mathbf{x}^t)\|^2] + \frac{\eta_c}{2} \mathbb{E}[\| \left( \nabla_{x_c} f_\lambda^t - \nabla_{x_c} f^t \right) \|^2] - \frac{\eta_c}{2} \mathbb{E}\left\|\nabla_{x_c} f_\lambda^t\right\|^2 - \eta_c \mathbb{E}[\|\nabla_{x_c} f(\mathbf{x}^t)\|^2]$$
$$\leq -\frac{\eta_c}{2} \mathbb{E}[\|\nabla_{x_c} f(\mathbf{x}^t)\|^2] + \frac{\eta_c}{4} L^2 \lambda^2 d_c^3 - \frac{\eta_c}{2} \mathbb{E}\left\|\nabla_{x_c} f_\lambda^t\right\|^2 \quad (21)$$

For $\mathcal{K}_4$:

$$\mathbb{E}[\|x_c^{t+1} - x_c^t\|^2] = \eta_c^2 \mathbb{E}\left\|G_c^t(\mathbf{x}^t; \xi^t)\right\|^2$$
$$= \eta_c^2 \mathbb{E}\left\|\nabla_{x_c} f_\lambda^t\right\|^2 + \eta_c^2 \mathbb{E}\left\|G_c^t - \nabla_{x_c} f_\lambda^t\right\|^2$$

Substituting (25) and (30) in order, we have

$$\eta_c^2 \mathbb{E}\left\|G_c^t - \nabla_{x_c} f_\lambda^t\right\|^2$$
$$\leq \eta_c^2 \left( \frac{4d_c}{P} \mathbb{E}[\|\nabla_{x_c} f(\mathbf{x}^t)\|^2] + \frac{4L^2 d_c}{P} \mathbb{E}[\|x_c^{t+1} - x_c^t\|^2] + \frac{2d_c \sigma_c^2}{P} + \frac{L^2 \lambda^2 d_c^3}{2P} - \frac{1}{P} \mathbb{E}\|\nabla_{x_s} f_\lambda^t\|^2 \right)$$
$$\leq \eta_c^2 \left( \frac{4d_c}{P} \mathbb{E}[\|\nabla_{x_c} f(\mathbf{x}^t)\|^2] + \frac{4L^2 d_c}{P} \mathbb{E}[\|x_c^{t+1} - x_c^t\|^2] + \frac{2d_c \sigma_c^2}{P} + \frac{L^2 \lambda^2 d_c^3}{2P} \right)$$
$$- \frac{\eta_c^2}{P} \left( 4\mathbb{E}[\|\nabla_{x_c} f(\mathbf{x}^t)\|^2] + 4L^2 \mathbb{E}[\|x_c^{t,i} - x_c^t\|^2] + \frac{L^2}{2} \lambda^2 d_c^3 \right)$$

$$\leq \frac{4\eta_c^2 d_c}{P}\mathbb{E}[\|\nabla_{x_c} f(\mathbf{x}^t)\|^2] + \frac{4\eta_c^2 L^2 d_c}{P}\mathbb{E}[\|x_c^{t+1} - x_c^t\|^2] + \frac{2\eta_c^2 d_c \sigma_c^2}{P}$$

So

$$\mathbb{E}[\|x_c^{t+1} - x_c^t\|^2] \leq \eta_c^2 \mathbb{E}\left\|\nabla_{x_c} f_\lambda^t\right\|^2 + \frac{4\eta_c^2 d_c}{P}\mathbb{E}[\|\nabla_{x_c} f(\mathbf{x}^t)\|^2]$$
$$+ \frac{4\eta_c^2 L^2 d_c}{P}\mathbb{E}[\|x_c^{t+1} - x_c^t\|^2] + \frac{2\eta_c^2 d_c \sigma_c^2}{P}$$

$$(1 - \frac{4\eta_c^2 L^2 d_c}{P})\mathbb{E}[\|x_c^{t+1} - x_c^t\|^2] \leq \eta_c^2 \mathbb{E}\left\|\nabla_{x_c} f_\lambda^t\right\|^2 + \frac{4\eta_c^2 d_c}{P}\mathbb{E}[\|\nabla_{x_c} f(\mathbf{x}^t)\|^2] + \frac{2\eta_c^2 d_c \sigma_c^2}{P}$$

Further assume $\eta_c \leq \frac{1}{L\sqrt{8d_c/P}}$, and we have

$$\mathbb{E}[\|x_c^{t+1} - x_c^t\|^2] \leq 2\eta_c^2 \mathbb{E}\left\|\nabla_{x_c} f_\lambda^t\right\|^2 + \frac{8\eta_c^2 d_c}{P}\mathbb{E}[\|\nabla_{x_c} f(\mathbf{x}^t)\|^2] + \frac{4\eta_c^2 d_c \sigma_c^2}{P} \qquad (22)$$

### C.1.3 Server-Client Combination

We now substitute (19), (20), (21), (22) into (17):

$$\mathbb{E}[f(\mathbf{x}^{t+1}) - f(\mathbf{x}^t)]$$

$$\leq \underbrace{\mathbb{E}[\langle \nabla_{x_s} f(\mathbf{x}^t), x_s^{t+1} - x_s^t \rangle]}_{\mathcal{K}_1} + \underbrace{\frac{L}{2}\mathbb{E}[\|x_s^{t+1} - x_s^t\|^2]}_{\mathcal{K}_2} + \underbrace{\mathbb{E}[\langle \nabla_{x_c} f(\mathbf{x}^t), x_c^{t+1} - x_c^t \rangle]}_{\mathcal{K}_3} + \underbrace{\frac{L}{2}\mathbb{E}[\|x_c^{t+1} - x_c^t\|^2]}_{\mathcal{K}_4}$$

$$\overset{(i)}{\leq} \underbrace{\left(16\eta_s^3 L^2(\tau^3 + \tau^2 d_s/P) - \frac{\eta_s \tau}{2}\right)\mathbb{E}[\|\nabla_{x_s} f(\mathbf{x}^t)\|^2] + \frac{\eta_s}{4}\tau L^2 \lambda^2 d_s^3 + \frac{8\eta_s^3 \tau^2 L^2 d_s \sigma_s^2}{P} + 2\eta_s^3 \tau^3 L^4 \lambda^2 d_s^3}_{\mathcal{K}_1}$$

$$\underbrace{- \frac{\eta_s}{2\tau}\mathbb{E}\left\|\sum_{i=0}^{\tau-1} \nabla_{x_s} f_\lambda^{t,i}\right\|^2}_{\mathcal{K}_1} + \underbrace{\frac{8\eta_s^2 L\tau d_s}{P}\mathbb{E}[\|\nabla_{x_s} f(\mathbf{x}^t)\|^2] + \frac{4\eta_s^2 L\tau d_s \sigma_s^2}{P} + 2\eta_s^2 L\mathbb{E}[\|\sum_{i=0}^{\tau-1} \nabla_{x_s} f_\lambda(\mathbf{x}^{t,i})\|^2]}_{\mathcal{K}_2}$$

$$\underbrace{- \frac{\eta_c}{2}\mathbb{E}[\|\nabla_{x_c} f(\mathbf{x}^t)\|^2] + \frac{\eta_c}{4}L^2 \lambda^2 d_c^3 - \frac{\eta_c}{2}\mathbb{E}\left\|\nabla_{x_c} f_\lambda^t\right\|^2}_{\mathcal{K}_3}$$

$$+ \underbrace{\eta_c^2 L\mathbb{E}\left\|\nabla_{x_c} f_\lambda^t\right\|^2 + \frac{4\eta_c^2 L d_c}{P}\mathbb{E}[\|\nabla_{x_c} f(\mathbf{x}^t)\|^2] + \frac{2\eta_c^2 L d_c \sigma_c^2}{P}}_{\mathcal{K}_4}$$

$$\overset{(ii)}{\leq} \left(16\eta_s^2 \tau L(\tau + d_s/P) - \frac{\eta_s \tau}{2}\right)\mathbb{E}[\|\nabla_{x_s} f(\mathbf{x}^t)\|^2] + \frac{4\eta_s^2 \tau L(2\eta_s \tau L + 1)d_s \sigma_s^2}{P}$$

$$+ \eta_s \tau L^2(2\eta_s^2 \tau^2 L^2 + 1/4)\lambda^2 d_s^3 + (\frac{4\eta_c^2 L d_c}{P} - \frac{\eta_c}{2})\mathbb{E}[\|\nabla_{x_c} f(\mathbf{x}^t)\|^2] + \frac{\eta_c}{4}L^2 \lambda^2 d_c^3 + \frac{2\eta_c^2 L d_c \sigma_c^2}{P}$$

$$\overset{(iii)}{\leq} - \frac{\eta_s \tau}{4}\mathbb{E}[\|\nabla_{x_s} f(\mathbf{x}^t)\|^2] + \frac{4\eta_s^2 \tau L(2\eta_s \tau L + 1)d_s \sigma_s^2}{P} + \eta_s \tau L^2(2\eta_s^2 \tau^2 L^2 + 1/4)\lambda^2 d_s^3$$

$$- \frac{\eta_c}{4}\mathbb{E}[\|\nabla_{x_c} f(\mathbf{x}^t)\|^2] + \frac{\eta_c}{4}L^2 \lambda^2 d_c^3 + \frac{2\eta_c^2 L d_c \sigma_c^2}{P} \qquad (23)$$

where in $(i)$ we applied (19), (20), (21), (22); in $(ii)$ we assume $\eta_s \leq \frac{1}{\tau L}$ to index on terms of $\eta_s$, assume $\eta_s \leq \frac{1}{4\tau L}, \eta_c \leq \frac{1}{2L}$ to remove the term $\mathbb{E}\left\|\sum_{i=0}^{\tau-1} \nabla_{x_s} f_\lambda^{t,i}\right\|^2$, and combine the terms of $\|\nabla_{x_c} f(\mathbf{x}^t)\|^2$ and $\|\nabla_{x_c} f(\mathbf{x}^t)\|^2$. In $(iii)$, we let

$$\eta_s \leq \frac{P}{64L(\tau P + 2d_s)}$$

And

$$\eta_c \leq \frac{P}{16L d_c}.$$

To combine the squared norm of the server gradient $\mathbb{E}[\|\nabla_{x_s} f\|^2]$ and client gradient $\mathbb{E}[\|\nabla_{x_c} f\|^2]$, we define the universal step size $\eta := \eta_s$, and let $\eta_c = \eta\tau$. Rearranging the terms in (23), we have

$$\frac{\eta\tau}{4}\left(\mathbb{E}[\|\nabla_{x_s} f(\mathbf{x}^t)\|^2] + \mathbb{E}[\|\nabla_{x_c} f(\mathbf{x}^t)\|^2]\right) \leq \mathbb{E}[f(\mathbf{x}^t) - f(\mathbf{x}^{t+1})] + \frac{4\eta^2\tau L(2\eta\tau L + 1)d_s\sigma_s^2}{P}$$
$$+ \eta\tau L^2(2\eta^2\tau^2 L^2 + 1/4)\lambda^2 d_s^3 + \frac{\eta}{4}\tau L^2\lambda^2 d_c^3$$
$$+ \frac{2\eta^2\tau^2 L d_c\sigma_c^2}{P}$$

$$\frac{\eta\tau}{4}\mathbb{E}[\|\nabla_{\mathbf{x}} f(\mathbf{x}^t)\|^2] \leq \mathbb{E}[f(\mathbf{x}^t) - f(\mathbf{x}^{t+1})] + \frac{4\eta^2\tau L(2\eta\tau L + 1)d_s\sigma_s^2}{P}$$
$$+ \eta\tau L^2(2\eta^2\tau^2 L^2 + 1/4)\lambda^2 d_s^3 + \frac{\eta}{4}\tau L^2\lambda^2 d_c^3 + \frac{2\eta^2\tau^2 L d_c\sigma_c^2}{P} \qquad (24)$$

Take the average from $t = 0$ to $T - 1$ at both sides:

$$\frac{1}{T}\sum_{t=0}^{T}\frac{\eta\tau}{4}\mathbb{E}[\|\nabla_{\mathbf{x}} f(\mathbf{x}^t)\|^2] \leq \frac{1}{T}\mathbb{E}[f(\mathbf{x}^0) - f(\mathbf{x}^T)] + \frac{4\eta^2\tau L(2\eta\tau L + 1)d_s\sigma_s^2}{P}$$
$$+ \eta\tau L^2(2\eta^2\tau^2 L^2 + 1/4)\lambda^2 d_s^3 + \frac{\eta}{4}\tau L^2\lambda^2 d_c^3 + \frac{2\eta^2\tau^2 L d_c\sigma_c^2}{P}$$

$$\frac{1}{T}\sum_{t=0}^{T}\mathbb{E}[\|\nabla_{\mathbf{x}} f(\mathbf{x}^t)\|^2] \leq \frac{4}{\eta\tau T}\mathbb{E}[f(\mathbf{x}^0) - f(\mathbf{x}^T)] + \frac{16\eta L(2\eta\tau L + 1)d_s\sigma_s^2}{P}$$
$$+ 4L^2(2\eta^2\tau^2 L^2 + 1/4)\lambda^2 d_s^3 + L^2\lambda^2 d_c^3 + \frac{8\eta\tau L d_c\sigma_c^2}{P},$$

where in the last step we divided both sides by $\frac{\eta\tau}{4}$. Let $P = 1$, and we complete the proof.

### C.1.4 Justification for Corollary 4.2

To further simplify the result and achieve the optimal convergence rate in Corollary 4.2, again, we assume $\eta \leq \frac{1}{\tau L}$. We also optimize upon $\eta$ to get the convergence rate. Let $\eta = \frac{1}{\sqrt{d\tau T}}$, we derive that

$$\frac{1}{T}\sum_{t=0}^{T}\mathbb{E}[\|\nabla_{\mathbf{x}} f(\mathbf{x}^t)\|^2] \leq \frac{4\sqrt{d}}{\sqrt{\tau T}}\mathbb{E}[f(\mathbf{x}^0) - f(\mathbf{x}^T)] + \frac{48L d_s\sigma_s^2}{\sqrt{d\tau T}} + 9L^2\lambda^2 d_s^3 + L^2\lambda^2 d_c^3 + \frac{8\sqrt{\tau}L d_c\sigma_c^2}{\sqrt{dT}}$$

Let $d_c = d/\sqrt{\tau}$ and $d_s = d - d/\sqrt{\tau}$, and further let

$$\lambda^2 = \frac{1}{\sqrt{\tau T}d^{5/2}L}$$

Thus, we have

$$\frac{1}{T}\sum_{t=0}^{T}\mathbb{E}[\|\nabla_{\mathbf{x}} f(\mathbf{x}^t)\|^2] \leq \frac{4\sqrt{d}}{\sqrt{\tau T}}\mathbb{E}[f(\mathbf{x}^0) - f(\mathbf{x}^T)] + \frac{48L\sqrt{d}\sigma_s^2}{\sqrt{\tau T}} + \frac{9\sqrt{d}}{\sqrt{\tau T}} + \frac{8L\sigma_c^2}{\sqrt{T}}$$

The convergence rate is seen to be $\mathcal{O}(\frac{\sqrt{d}}{\sqrt{\tau T}})$

## C.2 Important Lemmas

**Lemma C.2** (Bounds on the variance of Zeroth-order Gradient). *Under the same condition as Lemma B.1, and consider the stochastic Zeroth-order Gradient, we can further bound the variance of the stochastic Zeroth-order Gradient by true gradient at the beginning of the local iteration and the local update distance.*

$$\mathbb{E}[\|G_s^{t,i}(x_c^t, x_s^{t,i}; \xi^t) - \nabla_{x_s} f_\lambda^t(x_c^t, x_s^{t,i})\|^2] \leq \frac{4d_s}{P}\mathbb{E}[\|\nabla_{x_s} f(\mathbf{x}^t)\|^2] + \frac{4L^2 d_s}{P}\mathbb{E}[\|x_s^{t,i} - x_s^t\|^2]$$

$$+ \frac{2d_s\sigma_s^2}{P} + \frac{L^2\lambda^2 d_s^3}{2P} - \frac{1}{P}\mathbb{E}[\|\nabla_{x_s} f_\lambda^t(x_c^t, x_s^{t,i})\|^2] \tag{25}$$

*proof:*

We use multi-perturbation to calculate the Zeroth-Order Oracle: $G_s^{t,i}(x_c^t, x_s^{t,i}; \xi^t) = \frac{1}{P}\sum_{p=1}^{P} g_{s,p}^{t,i}(x_c^t, x_s^{t,i}; \xi^t)$, where $g_{s,p}^{t,i}$ is the stochastic Zeroth-Order Oracle for one perturbation. Then, the $\lambda$-smooth function is represented as $\mathbb{E}_{u_p, \xi^t}[g_{s,p}^{t,i}(x_c^t, x_s^{t,i}; \xi^t)] = \nabla_{x_s} f_\lambda^t(x_c^t, x_s^{t,i})$.
By Lemma B.1, we have

$$\mathbb{E}_u[\|g_{s,p}^{t,i}(x_c^t, x_s^{t,i}; \xi^t)\|^2] \le 2d_s \cdot \|\nabla_{x_s} F(x_c^t, x_s^{t,i}; \xi^t)\|^2 + \frac{L^2}{2}\lambda^2 d_s^3.$$

Thus we have

$$\mathbb{E}[\|G_s^{t,i} - \nabla_{x_s} f_\lambda^t(x_c^t, x_s^{t,i})\|^2]$$

$$= \frac{1}{P^2}\sum_{p=1}^{P} \mathbb{E}[\|g_{s,p}^{t,i}(x_c^t, x_s^{t,i}) - \nabla_{x_s} f_\lambda^t(x_c^t, x_s^{t,i})\|^2]$$

$$= \frac{1}{P^2}\sum_{p=1}^{P} \mathbb{E}[\|g_{s,p}^{t,i}(x_c^t, x_s^{t,i})\|^2] - \frac{1}{P}\|\nabla_{x_s} f_\lambda^t(x_c^t, x_s^{t,i})\|^2$$

$$\le \frac{1}{P^2}\sum_{p=1}^{P} \left[ 2d_s\mathbb{E}[\|\nabla_{x_s} F(x_c^t, x_s^{t,i}; \xi^t)\|^2] + \frac{L^2}{2}\lambda^2 d_s^3 \right] - \frac{1}{P}\mathbb{E}[\|\nabla_{x_s} f_\lambda^t(x_c^t, x_s^{t,i})\|^2]$$

$$\le \frac{1}{P^2}\sum_{p=1}^{P} \left[ 2d_s(\mathbb{E}[\|\nabla_{x_s} f(x_c^t, x_s^{t,i})\|^2] + \sigma_s^2) + \frac{L^2}{2}\lambda^2 d_s^3 \right] - \frac{1}{P}\mathbb{E}[\|\nabla_{x_s} f_\lambda^t(x_c^t, x_s^{t,i})\|^2]$$

$$= \frac{1}{P}\left[ 2d_s\mathbb{E}[\|\nabla_{x_s} f(x_c^t, x_s^{t,i})\|^2] + 2d_s\sigma_s^2 + \frac{L^2}{2}\lambda^2 d_s^3 - \mathbb{E}[\|\nabla_{x_s} f_\lambda^t(x_c^t, x_s^{t,i})\|^2] \right] \tag{26}$$

The bound for the squared norm of the variance is:

$$\mathbb{E}[\|\nabla_{x_s} f(x_c^t, x_s^{t,i})\|]^2 = \mathbb{E}[\|\nabla_{x_s} f(x_c^t, x_s^{t,i}) - \nabla_{x_s} f(\mathbf{x}^t) + \nabla_{x_s} f(\mathbf{x}^t)\|]^2$$

$$\le 2\mathbb{E}[\|\nabla_{x_s} f(x_c^t, x_s^{t,i}) - \nabla_{x_s} f(\mathbf{x}^t)\|^2] + 2\mathbb{E}[\|\nabla_{x_s} f(\mathbf{x}^t)\|^2]$$

$$\le 2L^2\mathbb{E}[\|x_s^{t,i} - x_s^t\|^2] + 2\mathbb{E}[\|\nabla_{x_s} f(\mathbf{x}^t)\|^2] \tag{27}$$

Substituting (27) into (26), and we finish the proof.
**Lemma C.3** (Bounds on the norm of the Zeroth-order gradient estimator).

$$\mathbb{E}[\|G_s^{t,i}(x_c^t, x_s^{t,i})\|^2]$$

$$\le \frac{4(d_s + P - 1)}{P}\mathbb{E}[\|\nabla_{x_s} f(\mathbf{x}^t)\|^2] + \frac{4L^2(d_s + P - 1)}{P}\mathbb{E}[\|x_s^{t,i} - x_s^t\|^2] + \frac{2d_s\sigma_s^2}{P} + \frac{L^2\lambda^2 d_s^3}{2} \tag{28}$$

*proof:*

It follows that

$$\mathbb{E}[\|G_s^{t,i}(x_c^t, x_s^{t,i})\|^2] = \mathbb{E}[\|G_s^{t,i}(x_c^t, x_s^{t,i}) - \nabla_{x_s} f_\lambda^t(x_c^t, x_s^{t,i})\|^2] + \mathbb{E}[\|\nabla_{x_s} f_\lambda^t(x_c^t, x_s^{t,i})\|^2] \tag{29}$$

From Lemma B.1 we have

$$\mathbb{E}[\|\nabla_{x_s} f_\lambda^t(x_c^t, x_s^{t,i})\|^2] \le 2\mathbb{E}[\|\nabla_{x_s} f(x_c^t, x_s^{t,i})\|^2] + \frac{L^2}{2}\lambda^2 d_s^3$$

$$\le 2\mathbb{E}[\|\nabla_{x_s} f(x_c^t, x_s^{t,i}) - \nabla_{x_s} f(\mathbf{x}^t) + \nabla_{x_s} f(\mathbf{x}^t)\|^2] + \frac{L^2}{2}\lambda^2 d_s^3$$

$$\le 4\mathbb{E}[\|\nabla_{x_s} f(x_c^t, x_s^{t,i}) - \nabla_{x_s} f(\mathbf{x}^t)\|^2] + 4\mathbb{E}[\|\nabla_{x_s} f(\mathbf{x}^t)\|^2] + \frac{L^2}{2}\lambda^2 d_s^3$$

$$\le 4\mathbb{E}[\|\nabla_{x_s} f(\mathbf{x}^t)\|^2] + 4L^2\mathbb{E}[\|x_s^{t,i} - x_s^t\|^2] + \frac{L^2}{2}\lambda^2 d_s^3 \tag{30}$$

Then we can finish the proof by combining Lemma C.2 and (30).

**Lemma C.4** (Bounds on multiple update steps(Zeroth Order)). *If $\eta_s \leq \frac{\sqrt{P}}{4\tau L\sqrt{(P+d_s/\tau)}}$, we have*

$$\sum_{i=0}^{\tau-1}\mathbb{E}[\|x_s^{t,i} - x_s^t\|^2] \leq \frac{16\eta_s^2\tau^3(P+d_s/\tau)}{P}\mathbb{E}[\|\nabla_{x_s}f(\mathbf{x}^t)\|^2] + \frac{8\eta_s^2\tau^2\sigma_L^2 d_s}{P} + 2\eta_s^2\tau^3 L^2\lambda^2 d_s^3$$

*proof:*

We first apply the update formula:

$$\sum_{i=0}^{\tau-1}\mathbb{E}[\|x_s^{t,i} - x_s^t\|^2] = \sum_{i=0}^{\tau-1}\eta_s^2\mathbb{E}[\|\sum_{j=0}^{i-1}G_s^{t,j}(x_c^t, x_s^{t,j})\|^2]$$

By the property of martingale difference sequence, we have

$$\mathbb{E}[\|\sum_{j=0}^{i-1}G_s^{t,j}(x_c^t, x_s^{t,j})\|^2]$$

$$\leq 2\mathbb{E}[\|\sum_{j=0}^{i-1}\nabla_{x_s}f_\lambda^t(x_c^t, x_s^{t,j})\|^2] + 2\mathbb{E}[\|\sum_{j=0}^{i-1}G_s^{t,j}(x_c^t, x_s^{t,j}) - \nabla_{x_s}f_\lambda^t(x_c^t, x_s^{t,j})\|^2]$$

$$\leq 2i\sum_{j=0}^{i-1}\mathbb{E}[\|\nabla_{x_s}f_\lambda^t(x_c^t, x_s^{t,j})\|^2] + 2\sum_{j=0}^{i-1}\mathbb{E}[\|G_s^{t,j}(x_c^t, x_s^{t,j}) - \nabla_{x_s}f_\lambda^t(x_c^t, x_s^{t,j})\|^2] \qquad (31)$$

We thus have

$$\sum_{i=0}^{\tau-1}\mathbb{E}[\|x_s^{t,i} - x_s^t\|^2]$$

$$\leq 2\sum_{i=0}^{\tau-1}\eta_s^2\left(i\sum_{j=0}^{i-1}\mathbb{E}[\|\nabla_{x_s}f_\lambda^t(x_c^t, x_s^{t,j})\|^2] + \sum_{j=0}^{i-1}\mathbb{E}[\|G_s^{t,j}(x_c^t, x_s^{t,j}) - \nabla_{x_s}f_\lambda^t(x_c^t, x_s^{t,j})\|^2]\right)$$

$$\leq 2\eta_s^2\tau^2\sum_{i=0}^{\tau-1}\mathbb{E}[\|\nabla_{x_s}f_\lambda^t(x_c^t, x_s^{t,i})\|^2] + 2\eta_s^2\tau\sum_{i=0}^{\tau-1}\mathbb{E}[\|G_s^{t,i}(x_c^t, x_s^{t,i}) - \nabla_{x_s}f_\lambda^t(x_c^t, x_s^{t,i})\|^2],$$

where the last inequality is by the following equations:

$$\sum_{i=0}^{\tau-1}\sum_{j=0}^{i-1}iX_j = \sum_{j=0}^{\tau-1}(\sum_{i=m}^{\tau-1}i)X_j \leq \sum_{j=0}^{\tau-1}\tau^2 X_j$$

And

$$\sum_{i=0}^{\tau-1}\sum_{j=0}^{i-1}X_j = \sum_{j=0}^{\tau-1}(\sum_{i=m}^{\tau-1})X_j \leq \sum_{j=0}^{\tau-1}\tau X_j$$

Substituting in (25):

$$\sum_{i=0}^{\tau-1}\mathbb{E}[\|x_s^{t,i} - x_s^t\|^2]$$

$$\leq 2\eta_s^2\frac{P\tau^2 - \tau}{P}\sum_{i=0}^{\tau-1}\mathbb{E}[\|\nabla_{x_s}f_\lambda^t(x_c^t, x_s^{t,i})\|^2]$$

$$+ 2\eta_s^2\tau\sum_{i=0}^{\tau-1}\left(\frac{4d_s}{P}\mathbb{E}[\|\nabla_{x_s}f_\mathbf{x}^t)\|^2] + \frac{4L^2 d_s}{P}\mathbb{E}[\|x_s^{t,i} - x_s^t\|^2] + \frac{2\sigma_L^2 d_s}{P} + \frac{L^2\lambda^2 d_s^3}{2P}\right)$$

Further substitute in (30):

$$\sum_{i=0}^{\tau-1}\mathbb{E}[\|x_s^{t,i}-x_s^t\|^2]$$

$$\leq 2\eta_s^2\frac{P\tau^2-\tau}{P}\sum_{i=0}^{\tau-1}\left(4\mathbb{E}[\|\nabla_{x_s}f(\mathbf{x}^t)\|^2]+4L^2\mathbb{E}[\|x_s^{t,i}-x_s^t\|^2]+\frac{L^2}{2}\lambda^2 d_s^3\right)$$

$$+2\eta_s^2\tau\sum_{i=0}^{\tau-1}\left(\frac{4d_s}{P}\mathbb{E}[\|\nabla_{x_s}f_{\mathbf{x}}^t)\|^2]+\frac{4L^2 d_s}{P}\mathbb{E}[\|x_s^{t,i}-x_s^t\|^2]+\frac{2\sigma_L^2 d_s}{P}+\frac{L^2\lambda^2 d_s^3}{2P}\right)$$

$$\leq\frac{8\eta_s^2\tau^3(P+d_s/\tau)}{P}\mathbb{E}[\|\nabla_{x_s}f_{\mathbf{x}}^t)\|^2]+\frac{8\eta_s^2\tau^2 L^2(P+d_s/\tau)}{P}\sum_{i=0}^{\tau-1}\mathbb{E}[\|x_s^{t,i}-x_s^t\|^2]$$

$$+\frac{4\eta_s^2\tau^2\sigma_L^2 d_s}{P}+\eta_s^2\tau^3 L^2\lambda^2 d_s^3$$

Rearranging the terms, we have

$$(1-\frac{8\eta_s^2\tau^2 L^2(P+d_s/\tau)}{P})\sum_{i=0}^{\tau-1}\mathbb{E}[\|x_s^{t,i}-x_s^t\|^2]$$

$$\leq\frac{8\eta_s^2\tau^3(P+d_s/\tau)}{P}\mathbb{E}[\|\nabla_{x_s}f_{\mathbf{x}}^t)\|^2]+\frac{4\eta_s^2\tau^2\sigma_L^2 d_s}{P}+\eta_s^2\tau^3 L^2\lambda^2 d_s^3$$

where we moved the term $\mathbb{E}[\|x_s^{t,i}-x_s^t\|^2]$ to the left in the last inequality. Let $\eta_s\leq\frac{1}{4L\sqrt{\tau^2+\tau d_s/P}}$, we have the coefficient on the L.H.S larger than $\frac{1}{2}$. Thus, we complete the proof.

# D   Proof for for MU-SplitFed

## D.1   Proof of main theorem

We now prove the main theorem of `MU-SplitFed`, and defer the important lemmas to Appendix D.2. We re-state the theorem below:

**Theorem D.1.** *Under Assumption B.1 to B.3, consider a SFL framework with $M$ clients, and let the server iteration number be $\tau$. If the learning rates on client and server satisfy $\eta_c/\tau=\eta_s=\eta\leq\min\{\frac{1}{\sqrt{120L^2(\tau^2+2\tau d_s)}},\frac{M}{12\tau Ld_c}\}$, the sequence of iterates generated by MU-Split satisfies:*

$$\frac{1}{T}\sum_{t=0}^T\mathbb{E}[\|\nabla_{\mathbf{x}}f(\mathbf{x}^t)\|^2]\leq\frac{4}{T\eta_g\eta\tau}\mathbb{E}[f(\mathbf{x}^0)-f(\mathbf{x}^T)]+24\eta(4\eta\tau L+\eta_g/M)L(\tau+2d_s)\epsilon^2$$

$$+16\eta(2\eta\tau L+\eta_g/M)Ld_s\sigma_s^2+(1/\tau+8\eta^2\tau L^2+2\eta_g\eta/M)\tau L^2\lambda^2 d_s^3$$

$$+\frac{12\eta_g\eta\tau Ld_c\epsilon^2}{M}+\frac{4\eta_g\eta\tau Ld_c\sigma_c^2}{M}+L^2\lambda^2 d_c^3 \qquad (32)$$

Similar to the proof of `MU-Split`, We begin by analyzing the update on client and server side, respectively. By (17), we bound one-round update $\mathcal{K}_1$, $\mathcal{K}_2$ on the server side, and $\mathcal{K}_3$, $\mathcal{K}_4$ on the client side.

### D.1.1   One-Round Update on Server Side

For $\mathcal{K}_1$:

$$\mathbb{E}[\langle\nabla_{x_s}f(\mathbf{x}^t),x_s^{t+1}-x_s^t\rangle]$$

$$=\mathbb{E}[\langle\nabla_{x_s}f(\mathbf{x}^t),-\frac{\eta_g}{M}\sum_{m=1}^M\sum_{i=0}^{\tau-1}\eta_s G_{s,m}^{t,i}(\mathbf{x}_m^{t,i};\xi_m^t)\rangle]$$

$$=\mathbb{E}[\langle \nabla_{x_s} f(\mathbf{x}^t), -\frac{\eta_g}{M} \sum_{m=1}^{M} \sum_{i=0}^{\tau-1} \eta_s \left( \nabla_{x_s} f_{m,\lambda}^{t,i} - \nabla_{x_s} f(\mathbf{x}^t) + \nabla_{x_s} f(\mathbf{x}^t) \right) \rangle]$$

$$=\mathbb{E}[\langle \sqrt{\eta_g \eta_s \tau} \nabla_{x_s} f(\mathbf{x}^t), -\frac{\sqrt{\eta_g \eta_s}}{M\sqrt{\tau}} \sum_{m=1}^{M} \sum_{i=0}^{\tau-1} \left( \nabla_{x_s} f_{m,\lambda}^{t,i} - \nabla_{x_s} f_m^t \right) \rangle] - \eta_g \eta_s \tau \mathbb{E}[\|\nabla_{x_s} f(\mathbf{x}^t)\|^2]$$

$$=\frac{\eta_g \eta_s \tau}{2} \mathbb{E}[\|\nabla_{x_s} f(\mathbf{x}^t)\|^2] + \frac{\eta_g \eta_s}{2M^2\tau} \mathbb{E}\left[ \left\| \sum_{m=1}^{M} \sum_{i=0}^{\tau-1} \left( \nabla_{x_s} f_{m,\lambda}^{t,i} - \nabla_{x_s} f_m^t \right) \right\|^2 \right]$$

$$\quad - \frac{\eta_g \eta_s}{2M^2\tau} \mathbb{E}\left\| \sum_{m=1}^{M} \sum_{i=0}^{\tau-1} \nabla_{x_s} f_{m,\lambda}^{t,i} \right\|^2 - \eta_g \eta_s \tau \mathbb{E}[\|\nabla_{x_s} f(\mathbf{x}^t)\|^2]$$

$$=-\frac{\eta_g \eta_s \tau}{2} \mathbb{E}[\|\nabla_{x_s} f(\mathbf{x}^t)\|^2] + \frac{\eta_g \eta_s}{2M^2\tau} \mathbb{E}\left[ \left\| \sum_{m=1}^{M} \sum_{i=0}^{\tau-1} \left( \nabla_{x_s} f_{m,\lambda}^{t,i} - \nabla_{x_s} f_m^{t,i} + \nabla_{x_s} f_m^{t,i} - \nabla_{x_s} f_m^t \right) \right\|^2 \right]$$

$$\quad - \frac{\eta_g \eta_s}{2M^2\tau} \mathbb{E}\left\| \sum_{m=1}^{M} \sum_{i=0}^{\tau-1} \nabla_{x_s} f_{m,\lambda}^{t,i} \right\|^2$$

$$\leq -\frac{\eta_g \eta_s \tau}{2} \mathbb{E}[\|\nabla_{x_s} f(\mathbf{x}^t)\|^2] + \frac{\eta_g \eta_s}{M^2\tau} \mathbb{E}\left[ \left\| \sum_{m=1}^{M} \sum_{i=0}^{\tau-1} \left( \nabla_{x_s} f_{m,\lambda}^{t,i} - \nabla_{x_s} f_m^{t,i} \right) \right\|^2 \right]$$

$$\quad + \frac{\eta_g \eta_s}{M^2\tau} \mathbb{E}\left[ \left\| \sum_{m=1}^{M} \sum_{i=0}^{\tau-1} \left( \nabla_{x_s} f_m^{t,i} - \nabla_{x_s} f_m^t \right) \right\|^2 \right] - \frac{\eta_g \eta_s}{2M^2\tau} \mathbb{E}\left\| \sum_{m=1}^{M} \sum_{i=0}^{\tau-1} \nabla_{x_s} f_{m,\lambda}^{t,i} \right\|^2$$

$$\leq -\frac{\eta_g \eta_s \tau}{2} \mathbb{E}[\|\nabla_{x_s} f(\mathbf{x}^t)\|^2] + \frac{\eta_g \eta_s}{M} \sum_{m=1}^{M} \sum_{i=0}^{\tau-1} \mathbb{E}\left[ \left\| \left( \nabla_{x_s} f_{m,\lambda}^{t,i} - \nabla_{x_s} f_m^{t,i} \right) \right\|^2 \right]$$

$$\quad + \frac{\eta_g \eta_s}{M} \sum_{m=1}^{M} \sum_{i=0}^{\tau-1} \mathbb{E}\left[ \left\| \left( \nabla_{x_s} f_m^{t,i} - \nabla_{x_s} f_m^t \right) \right\|^2 \right] - \frac{\eta_g \eta_s}{2M^2\tau} \mathbb{E}\left\| \sum_{m=1}^{M} \sum_{i=0}^{\tau-1} \nabla_{x_s} f_{m,\lambda}^{t,i} \right\|^2$$

$$\leq -\frac{\eta_g \eta_s \tau}{2} \mathbb{E}[\|\nabla_{x_s} f(\mathbf{x}^t)\|^2] + \frac{\eta_g \eta_s}{4} \tau L^2 \lambda^2 d_s^3 + \frac{\eta_g \eta_s L^2}{M} \sum_{m=1}^{M} \underbrace{\sum_{i=0}^{\tau-1} \mathbb{E}\left[ \|x_{s,m}^{t,i} - x_{s,m}^t\|^2 \right]}_{\mathcal{A}_1}$$

$$\quad - \frac{\eta_g \eta_s}{2M^2\tau} \mathbb{E}\left\| \sum_{m=1}^{M} \sum_{i=0}^{\tau-1} \nabla_{x_s} f_{m,\lambda}^{t,i} \right\|^2$$

$$\leq -\frac{\eta_g \eta_s \tau}{2} \mathbb{E}[\|\nabla_{x_s} f(\mathbf{x}^t)\|^2] + \frac{\eta_g \eta_s}{4} \tau L^2 \lambda^2 d_s^3 - \frac{\eta_g \eta_s}{2M^2\tau} \mathbb{E}\left\| \sum_{m=1}^{M} \sum_{i=0}^{\tau-1} \nabla_{x_s} f_{m,\lambda}^{t,i} \right\|^2$$

$$\quad + \eta_g \eta_s L^2 \left( 24\eta_s^2(\tau^3 + \tau^2 d_s/P)\mathbb{E}[\|\nabla_{x_s} f(\mathbf{x}^t)\|^2] + 24\eta_s^2(\tau^3 + \tau^2 d_s/P)\epsilon^2 + \frac{8\eta_s^2 \tau^2 d_s \sigma_s^2}{P} + 2\eta_s^2 \tau^3 L^2 \lambda^2 d_s^3 \right)$$

$$=\left( 24\eta_g \eta_s^3 L^2(\tau^3 + \tau^2 d_s/P) - \frac{\eta_g \eta_s \tau}{2} \right) \mathbb{E}[\|\nabla_{x_s} f(\mathbf{x}^t)\|^2] + \frac{\eta_g \eta_s}{4} \tau L^2 \lambda^2 d_s^3 + 24\eta_g \eta_s^3 L^2(\tau^3 + \tau^2 d_s/P)\epsilon^2$$

$$\quad + \frac{8\eta_g \eta_s^3 \tau^2 L^2 d_s \sigma_s^2}{P} + 2\eta_g \eta_s^3 \tau^3 L^4 \lambda^2 d_s^3 - \frac{\eta_g \eta_s}{2M^2\tau} \mathbb{E}\left\| \sum_{m=1}^{M} \sum_{i=0}^{\tau-1} \nabla_{x_s} f_{m,\lambda}^{t,i} \right\|^2,$$

where in the last step we use Lemma D.3 for $\mathcal{A}_1$.

For $\mathcal{K}_2$:

$$\mathbb{E}[\|x_s^{t+1} - x_s^t\|^2] = \frac{\eta_g^2 \eta_s^2}{M^2} \mathbb{E}\left\| \sum_{m=1}^{M} \sum_{i=0}^{\tau-1} G_{s,m}^{t,i}(\mathbf{x}_m^{t,i}; \xi_m^t) \right\|^2$$

$$\leq 2\frac{\eta_g^2\eta_s^2}{M^2}\mathbb{E}\left\|\sum_{m=1}^{M}\sum_{i=0}^{\tau-1}\nabla_{x_s}f_{m,\lambda}^{t,i}\right\|^2 + 2\frac{\eta_g^2\eta_s^2}{M^2}\sum_{m=1}^{M}\sum_{i=0}^{\tau-1}\mathbb{E}\left\|G_{s,m}^{t,i}-\nabla_{x_s}f_{m,\lambda}^{t,i}\right\|^2$$

Substituting (34) and (40) in order, we have

$$\frac{\eta_g^2\eta_s^2}{M^2}\sum_{m=1}^{M}\sum_{i=0}^{\tau-1}\mathbb{E}\left\|G_{s,m}^{t,i}-\nabla_{x_s}f_{m,\lambda}^{t,i}\right\|^2$$

$$\leq\frac{\eta_g^2\eta_s^2}{M^2}\sum_{m=1}^{M}\sum_{i=0}^{\tau-1}\left(\frac{-1}{P}(6\mathbb{E}[\|\nabla_{x_s}f(\mathbf{x}^t)\|^2]+6\epsilon^2+6L^2\mathbb{E}[\|x_{s,m}^{t,i}-x_{s,m}^t\|^2]+\frac{L^2}{2}\lambda^2 d_s^3)\right.$$

$$\left.+\frac{6d_s}{P}\mathbb{E}[\|\nabla_{x_s}f(\mathbf{x}^t)\|^2]+\frac{6L^2d_s}{P}\mathbb{E}[\|x_{s,m}^{t,i}-x_{s,m}^t\|^2]+\frac{6d_s\epsilon^2}{P}+\frac{2d_s\sigma_s^2}{P}+\frac{L^2\lambda^2 d_s^3}{2P}\right)$$

$$\leq\frac{\eta_g^2\eta_s^2}{M^2}\sum_{m=1}^{M}\sum_{i=0}^{\tau-1}\left(\frac{6d_s}{P}\mathbb{E}[\|\nabla_{x_s}f(\mathbf{x}^t)\|^2]+\frac{6L^2d_s}{P}\mathbb{E}[\|x_{s,m}^{t,i}-x_{s,m}^t\|^2]+\frac{6d_s\epsilon^2}{P}+\frac{2d_s\sigma_s^2}{P}\right)$$

$$=\frac{\eta_g^2\eta_s^2}{M}\left(\frac{6d_s}{P}\mathbb{E}[\|\nabla_{x_s}f(\mathbf{x}^t)\|^2]+\frac{6d_s\epsilon^2}{P}+\frac{2d_s\sigma_s^2}{P}\right)$$

$$+\frac{6\eta_g^2\eta_s^2 L^2 d_s}{PM}\sum_{m=1}^{M}\sum_{i=0}^{\tau-1}\mathbb{E}[\|x_{s,m}^{t,i}-x_{s,m}^t\|^2]$$

We then use Lemma D.3, and assume that $\eta_s\leq\frac{\sqrt{P}}{L\sqrt{24\tau d_s}}$. It follows that

$$\frac{\eta_g^2\eta_s^2}{M^2}\sum_{m=1}^{M}\sum_{i=0}^{\tau-1}\mathbb{E}\left\|G_{s,m}^{t,i}-\nabla_{x_s}f_{m,\lambda}^{t,i}\right\|^2$$

$$\leq\frac{\eta_g^2\eta_s^2}{M}\left(\frac{6d_s}{P}\mathbb{E}[\|\nabla_{x_s}f(\mathbf{x}^t)\|^2]+\frac{6d_s\epsilon^2}{P}+\frac{2d_s\sigma_s^2}{P}\right)$$

$$+\frac{\eta_g^2}{4\tau M^2}\sum_{m=1}^{M}\left(24\eta_s^2(\tau^3+\tau^2 d_s/P)\mathbb{E}[\|\nabla_{x_s}f(\mathbf{x}^t)\|^2]+24\eta_s^2(\tau^3+\tau^2 d_s/P)\epsilon^2\right.$$

$$\left.+\frac{8\eta_s^2\tau^2 d_s\sigma_s^2}{P}+2\eta_s^2\tau^3 L^2\lambda^2 d_s^3\right)$$

$$\leq\frac{\eta_g^2\eta_s^2}{M}\left(6(\tau^2+2\tau d_s/P)\mathbb{E}[\|\nabla_{x_s}f(\mathbf{x}^t)\|^2]+6(\tau^2+2\tau d_s/P)\epsilon^2+\frac{\tau^2 L^2\lambda^2 d_s^3}{2}+\frac{4\tau d_s\sigma_s^2}{P}\right),$$

where in the last step we use the fact that $\tau\geq 1$.

### D.1.2 One-Round Update on Client Side

For $\mathcal{K}_3$:

$$\mathbb{E}[\langle\nabla_{x_c}f(\mathbf{x}^t),x_c^{t+1}-x_c^t\rangle]$$

$$=\mathbb{E}[\langle\nabla_{x_c}f(\mathbf{x}^t),-\frac{\eta_g}{M}\sum_{m=1}^{M}\eta_c G_{c,m}^{t,i}(\mathbf{x}_m^t;\xi_m^t)\rangle]$$

$$=\mathbb{E}[\langle\nabla_{x_c}f(\mathbf{x}^t),-\frac{\eta_g}{M}\sum_{m=1}^{M}\eta_c\left(\nabla_{x_c}f_{m,\lambda}^t-\nabla_{x_c}f(\mathbf{x}^t)+\nabla_{x_c}f(\mathbf{x}^t)\right)\rangle]$$

$$=\mathbb{E}[\langle\sqrt{\eta_g\eta_c}\nabla_{x_c}f(\mathbf{x}^t),-\frac{\sqrt{\eta_g\eta_c}}{M}\sum_{m=1}^{M}\left(\nabla_{x_c}f_{m,\lambda}^t-\nabla_{x_c}f_m^t\right)\rangle]-\eta_g\eta_c\mathbb{E}[\|\nabla_{x_c}f(\mathbf{x}^t)\|^2]$$

$$=\frac{\eta_g\eta_c}{2}\mathbb{E}[\|\nabla_{x_c}f(\mathbf{x}^t)\|^2]+\frac{\eta_g\eta_c}{2M^2}\mathbb{E}[\|\sum_{m=1}^{M}\left(\nabla_{x_c}f_{m,\lambda}^t-\nabla_{x_c}f_m^t\right)\|^2]$$

$$- \frac{\eta_g \eta_c}{2M^2} \mathbb{E} \left\| \sum_{m=1}^{M} \nabla_{x_c} f_{m,\lambda}^t \right\|^2 - \eta_g \eta_c \mathbb{E}[\|\nabla_{x_c} f(\mathbf{x}^t)\|^2]$$

$$\leq - \frac{\eta_g \eta_c}{2} \mathbb{E}[\|\nabla_{x_c} f(\mathbf{x}^t)\|^2] + \frac{\eta_g \eta_c}{4} L^2 \lambda^2 d_c^3 - \frac{\eta_g \eta_c}{2M^2} \mathbb{E} \left\| \sum_{m=1}^{M} \nabla_{x_c} f_{m,\lambda}^t \right\|^2$$

For $\mathcal{K}_4$:

$$\mathbb{E}[\|x_c^{t+1} - x_c^t\|^2] = \frac{\eta_g^2 \eta_c^2}{M^2} \mathbb{E} \left\| \sum_{m=1}^{M} G_{c,m}^t(\mathbf{x}_m^t; \xi_m^t) \right\|^2$$

$$= \frac{\eta_g^2 \eta_c^2}{M^2} \mathbb{E} \left\| \sum_{m=1}^{M} \nabla_{x_c} f_{m,\lambda}^t \right\|^2 + \frac{\eta_g^2 \eta_c^2}{M^2} \mathbb{E} \left\| \sum_{m=1}^{M} (G_{c,m}^t - \nabla_{x_c} f_{m,\lambda}^t) \right\|^2$$

$$\leq \frac{\eta_g^2 \eta_c^2}{M^2} \mathbb{E} \left\| \sum_{m=1}^{M} \nabla_{x_c} f_{m,\lambda}^t \right\|^2 + \frac{\eta_g^2 \eta_c^2}{M^2} \sum_{m=1}^{M} \mathbb{E} \left\| G_{c,m}^t - \nabla_{x_c} f_{m,\lambda}^t \right\|^2$$

Substituting (34) and (40) in order, we have

$$\frac{\eta_g^2 \eta_c^2}{M^2} \sum_{m=1}^{M} \mathbb{E} \left\| G_{c,m}^t - \nabla_{x_c} f_{m,\lambda}^t \right\|^2$$

$$\leq \frac{\eta_g^2 \eta_c^2}{M^2} \sum_{m=1}^{M} \left( -\frac{1}{P} \mathbb{E} \left\| \nabla_{x_c} f_{m,\lambda}^t \right\|^2 + \frac{6d_c}{P} \mathbb{E}[\|\nabla_{x_c} f(\mathbf{x}^t)\|^2] + \frac{6d_c \epsilon^2 d_c}{P} + \frac{2d_c \sigma_c^2}{P} + \frac{L^2 \lambda^2 d_c^3}{2P} \right)$$

$$\leq \frac{\eta_g^2 \eta_c^2}{M^2} \sum_{m=1}^{M} \left( -\frac{1}{P} (6\mathbb{E}[\|\nabla_{x_c} f(\mathbf{x}^t)\|^2] + 6\epsilon^2 + \frac{L^2}{2} \lambda^2 d_c^3) \right.$$

$$\left. + \frac{6d_c}{P} \mathbb{E}[\|\nabla_{x_c} f(\mathbf{x}^t)\|^2] + \frac{6d_c \epsilon^2}{P} + \frac{2d_c \sigma_c^2}{P} + \frac{L^2 \lambda^2 d_c^3}{2P} \right)$$

$$\leq \frac{\eta_g^2 \eta_c^2}{M} \left( \frac{6d_c}{P} \mathbb{E}[\|\nabla_{x_c} f(\mathbf{x}^t)\|^2] + \frac{6d_c \epsilon^2}{P} + \frac{2d_c \sigma_c^2}{P} \right)$$

### D.1.3 Server-Client Combination

Putting together:

$$\mathbb{E}[f(\mathbf{x}^{t+1}) - f(\mathbf{x}^t)]$$

$$\leq \left( 24\eta_g \eta_s^3 L^2 (\tau^3 + \tau^2 d_s/P) - \frac{\eta_g \eta_s \tau}{2} \right) \mathbb{E}[\|\nabla_{x_s} f(\mathbf{x}^t)\|^2] + \frac{\eta_g \eta_s}{4} \tau L^2 \lambda^2 d_s^3 + 24\eta_g \eta_s^3 L^2 (\tau^3 + \tau^2 d_s/P)\epsilon^2$$

$$+ \frac{8\eta_g \eta_s^3 \tau^2 L^2 d_s \sigma_s^2}{P} + 2\eta_g \eta_s^3 \tau^3 L^4 \lambda^2 d_s^3 - \frac{\eta_g \eta_s}{2M^2 \tau} \mathbb{E} \left\| \sum_{m=1}^{M} \sum_{i=0}^{\tau-1} \nabla_{x_s} f_{m,\lambda}^{t,i} \right\|^2 + \frac{\eta_g^2 \eta_s^2 L}{M^2} \mathbb{E} \left\| \sum_{m=1}^{M} \sum_{i=0}^{\tau-1} \nabla_{x_s} f_{m,\lambda}^{t,i} \right\|^2$$

$$+ \frac{\eta_g^2 \eta_s^2 L}{M} \left( 6(\tau^2 + 2\tau d_s/P) \mathbb{E}[\|\nabla_{x_s} f(\mathbf{x}^t)\|^2] + 6(\tau^2 + 2\tau d_s/P)\epsilon^2 + \frac{1}{2}\tau^2 L^2 \lambda^2 d_s^3 + \frac{4\tau d_s \sigma_s^2}{P} \right)$$

$$- \frac{\eta_g \eta_c}{2} \mathbb{E}[\|\nabla_{x_c} f(\mathbf{x}^t)\|^2] + \frac{\eta_g \eta_c}{4} L^2 \lambda^2 d_c^3 - \frac{\eta_g \eta_c}{2M^2} \mathbb{E} \left\| \sum_{m=1}^{M} \nabla_{x_c} f_{m,\lambda}^t \right\|^2$$

$$+ \frac{\eta_g^2 \eta_c^2 L}{2M^2} \mathbb{E} \left\| \sum_{m=1}^{M} \nabla_{x_c} f_{m,\lambda}^t \right\|^2 + \frac{\eta_g^2 \eta_c^2 L}{M} \left( \frac{3d_c}{P} \mathbb{E}[\|\nabla_{x_c} f(\mathbf{x}^t)\|^2] + \frac{3d_c \epsilon^2}{P} + \frac{d_c \sigma_c^2}{P} \right)$$

$$\leq \left( 6\eta_g \eta_s^2 (4\eta_s \tau L + \eta_g/M) L(\tau^2 + 2\tau d_s/P) - \frac{\eta_g \eta_s \tau}{2} \right) \mathbb{E}[\|\nabla_{x_s} f(\mathbf{x}^t)\|^2]$$

$$+ 6\eta_g \eta_s^2 (4\eta_s \tau L + \eta_g/M) L(\tau^2 + 2\tau d_s/P)\epsilon^2$$

$$+ \frac{4\eta_g \eta_s^2 (2\eta_s \tau L + \eta_g/M)\tau L d_s \sigma_s^2}{P} + \frac{\eta_g \eta_s (1/\tau + 8\eta_s^2 \tau L^2 + 2\eta_g \eta_s/M)}{4} \tau^2 L^2 \lambda^2 d_s^3$$

$$+ \left(\frac{3\eta_g^2\eta_c^2 Ld_c}{MP} - \frac{\eta_g\eta_c}{2}\right)\mathbb{E}[\|\nabla_{x_c}f(\mathbf{x}^t)\|^2] + \frac{3\eta_g^2\eta_c^2 Ld_c\epsilon^2}{MP} + \frac{\eta_g^2\eta_c^2 Ld_c\sigma_c^2}{MP} + \frac{\eta_g\eta_c L^2\lambda^2 d_c^3}{4}$$

$$\leq -\frac{\eta_g\eta_s\tau}{4}\mathbb{E}[\|\nabla_{x_s}f(\mathbf{x}^t)\|^2] + 6\eta_g\eta_s^2(4\eta_s\tau L + \eta_g/M)L(\tau^2 + 2\tau d_s/P)\epsilon^2$$

$$+ \frac{4\eta_g\eta_s^2(2\eta_s\tau L + \eta_g/M)\tau Ld_s\sigma_s^2}{P} + \frac{\eta_g\eta_s(1/\tau + 8\eta_s^2\tau L^2 + 2\eta_g\eta_s/M)}{4}\tau^2 L^2\lambda^2 d_s^3$$

$$- \frac{\eta_g\eta_c}{4}\mathbb{E}[\|\nabla_{x_c}f(\mathbf{x}^t)\|^2] + \frac{3\eta_g^2\eta_c^2 Ld_c\epsilon^2}{MP} + \frac{\eta_g^2\eta_c^2 Ld_c\sigma_c^2}{MP} + \frac{\eta_g\eta_c L^2\lambda^2 d_c^3}{4},$$

where we assume

$$\eta_s \leq \frac{1}{\sqrt{120L^2(\tau^2 + 2\tau d_s/P)}}, \quad \eta_c \leq \frac{MP}{12Ld_c}$$

and combine the terms.

To combine the squared norm of the server gradient $\mathbb{E}[\|\nabla_{x_s}F\|^2]$ and client gradient $\mathbb{E}[\|\nabla_{x_c}F\|^2]$, we define the universal step size $\eta := \eta_s$, and let $\eta_c = \eta\tau$. Rearranging the terms, we have

$$\frac{\eta_g\eta\tau}{4}\mathbb{E}[\|\nabla_{\mathbf{x}}f(\mathbf{x}^t)\|^2] \leq \mathbb{E}[f(\mathbf{x}^t) - f(\mathbf{x}^{t+1})] + 6\eta_g\eta^2(4\eta\tau L + \eta_g/M)L(\tau^2 + 2\tau d_s/P)\epsilon^2$$

$$+ \frac{4\eta_g\eta^2(2\eta\tau L + \eta_g/M)\tau Ld_s\sigma_s^2}{P} + \frac{\eta_g\eta(1/\tau + 8\eta^2\tau L^2 + 2\eta_g\eta/M)}{4}\tau^2 L^2\lambda^2 d_s^3$$

$$+ \frac{3\eta_g^2\eta^2\tau^2 Ld_c\epsilon^2}{MP} + \frac{\eta_g^2\eta^2\tau^2 Ld_c\sigma_c^2}{MP} + \frac{\eta_g\eta\tau L^2\lambda^2 d_c^3}{4},$$

Taking average from $t = 0$ to $T - 1$ at both sides:

$$\frac{1}{T}\sum_{t=0}^{T}\frac{\eta_g\eta\tau}{4}\mathbb{E}[\|\nabla_{\mathbf{x}}f(\mathbf{x}^t)\|^2] \leq \frac{1}{T}\mathbb{E}[f(\mathbf{x}^0) - f(\mathbf{x}^T)] + 6\eta_g\eta^2(4\eta\tau L + \eta_g/M)L(\tau^2 + 2\tau d_s/P)\epsilon^2$$

$$+ \frac{4\eta_g\eta^2(2\eta\tau L + \eta_g/M)\tau Ld_s\sigma_s^2}{P} + \frac{\eta_g\eta(1/\tau + 8\eta^2\tau L^2 + 2\eta_g\eta/M)}{4}\tau^2 L^2\lambda^2 d_s^3$$

$$+ \frac{3\eta_g^2\eta^2\tau^2 Ld_c\epsilon^2}{MP} + \frac{\eta_g^2\eta^2\tau^2 Ld_c\sigma_c^2}{MP} + \frac{\eta_g\eta\tau L^2\lambda^2 d_c^3}{4}$$

$$\frac{1}{T}\sum_{t=0}^{T}\mathbb{E}[\|\nabla_{\mathbf{x}}f(\mathbf{x}^t)\|^2] \leq \frac{4}{T\eta_g\eta\tau}\mathbb{E}[f(\mathbf{x}^0) - f(\mathbf{x}^T)] + 24\eta(4\eta\tau L + \eta_g/M)L(\tau + 2d_s/P)\epsilon^2$$

$$+ \frac{16\eta(2\eta\tau L + \eta_g/M)Ld_s\sigma_s^2}{P} + (1/\tau + 8\eta^2\tau L^2 + 2\eta_g\eta/M)\tau L^2\lambda^2 d_s^3$$

$$+ \frac{12\eta_g\eta\tau Ld_c\epsilon^2}{MP} + \frac{4\eta_g\eta\tau Ld_c\sigma_c^2}{MP} + L^2\lambda^2 d_c^3 \tag{33}$$

where in the last step we divided both sides by $\frac{\eta\tau}{4}$. Let $P = 1$, and we complete the proof.

### D.1.4 Justification for Corollary 4.4

The optimal convergence rate is achieved by optimizing (33) w.r.t $\eta$ and $\eta_g$, solving which gives $\eta_g = \sqrt{\tau M}$ and $\eta = \frac{1}{\tau L\sqrt{dT}}$. Since $d_s, d_c$ is typically very large, and $\tau$ is relatively small, we can assume that $\tau \leq d_s$. Thus, we have

$$\frac{1}{T}\sum_{t=0}^{T}\mathbb{E}[\|\nabla_{\mathbf{x}}f(\mathbf{x}^t)\|^2] \leq \frac{4L\sqrt{d}}{\sqrt{\tau TM}}\mathbb{E}[f(\mathbf{x}^0) - f(\mathbf{x}^T)] + \frac{24(4/\sqrt{T} + \sqrt{\tau}/\sqrt{M})(d_s/\sqrt{d})\epsilon^2}{\tau\sqrt{T}}$$

$$+ \frac{16(2/\sqrt{T} + \sqrt{\tau}/\sqrt{M})(d_s/\sqrt{d})\sigma_s^2}{\tau\sqrt{T}} + (1 + 8/dT + 2\sqrt{\tau}/L\sqrt{dTM})L^2\lambda^2 d_s^3$$

$$+ \frac{12\sqrt{\tau}(d_c/\sqrt{d})\epsilon^2}{\sqrt{TM}} + \frac{4\sqrt{\tau}(d_c/\sqrt{d})\sigma_c^2}{\sqrt{TM}} + L^2\lambda^2 d_c^3$$

Since $d, T, M, \tau$ are positive integers and $L$ are typically large, we have that

$$(1 + 8/dT + 2\sqrt{\tau}/L\sqrt{dTM})L^2\lambda^2 d_s^3 + L^2\lambda^2 d_c^3 \leq 11L^2\lambda^2 d^3$$

Let $d/d_c^2 = \tau$, so that $d_c = \sqrt{d/\tau}$ and $d_s = d - \sqrt{d/\tau}$, and further let

$$\lambda^2 = \frac{1}{\tau T d^{5/2} L^2}$$

Finally, we have

$$\frac{1}{T}\sum_{t=0}^{T}\mathbb{E}[\|\nabla_{\mathbf{x}} f(\mathbf{x}^t)\|^2] \leq \frac{4L\sqrt{d}}{\sqrt{\tau TM}}\mathbb{E}[f(\mathbf{x}^0) - f(\mathbf{x}^T)] + \frac{8\sqrt{d}(3\epsilon^2 + 2\sigma_s^2)}{\sqrt{\tau TM}} + \frac{32\sqrt{d}(3\epsilon^2 + \sigma_s^2)}{\tau T}$$
$$+ \frac{4(3\epsilon^2 + \sigma_c^2)}{\sqrt{TM}} + \frac{6\sqrt{d}}{\tau T}$$

We can conclude that, the overall convergence rate is $O(\frac{\sqrt{d}}{\sqrt{\tau TM}})$

### D.2 Important Lemmas

**Lemma D.2** (Bounds on the variance of Zeroth-order Gradient). *Under the same condition as Lemma B.1, and consider the stochastic Zeroth-order Gradient, we can further bound the variance of the **local** stochastic Zeroth-order Gradient by **global** gradient at the beginning of the local iteration and the local update distance.*

$$\mathbb{E}[\|G_s^{t,i}(\mathbf{x}_m^{t,i};\xi_m^t) - \nabla_{x_s} f_\lambda^t(\mathbf{x}_m^{t,i})\|^2] \leq \frac{6d_s}{P}\mathbb{E}[\|\nabla_{x_s} f(\mathbf{x}^t)\|^2] + \frac{6L^2 d_s}{P}\mathbb{E}[\|x_{s,m}^{t,i} - x_{s,m}^t\|^2]$$
$$+ \frac{6d_s\epsilon^2}{P} + \frac{2d_s\sigma_s^2}{P} + \frac{L^2\lambda^2 d_s^3}{2P} - \frac{1}{P}\mathbb{E}[\|\nabla_{x_s} f_\lambda^t(\mathbf{x}_m^{t,i})\|^2]$$

$$(34)$$

*proof:*

First notice that $G_s^{t,i}(\mathbf{x}_m^{t,i};\xi_m^t) = \frac{1}{P}\sum_{p=1}^{P} g_{s,p}^{t,i}(\mathbf{x}_m^{t,i};\xi_m^t)$ and $\mathbb{E}_{u_p,\xi_m^t}[g_{s,p}^{t,i}(\mathbf{x}_m^{t,i};\xi_m^t)] = \nabla_{x_s} f_\lambda^t(\mathbf{x}_m^{t,i})$. By Lemma B.1, we have

$$\mathbb{E}_u[\|g_{s,p}^{t,i}(\mathbf{x}_m^{t,i};\xi_m^t)\|^2] \leq 2d_s \cdot \|\nabla_{x_s} F(\mathbf{x}_m^{t,i};\xi_m^t)\|^2 + \frac{L^2}{2}\lambda^2 d_s^3.$$

Thus we have

$$\mathbb{E}[\|G_s^{t,i}(\mathbf{x}_m^{t,i};\xi_m^t) - \nabla_{x_s} f_\lambda^t(\mathbf{x}_m^{t,i})\|^2$$
$$= \frac{1}{P^2}\sum_{p=1}^{P}\mathbb{E}[\|g_{s,p}^{t,i}(\mathbf{x}_m^{t,i}) - \nabla_{x_s} f_\lambda^t(\mathbf{x}_m^{t,i})\|^2]$$
$$= \frac{1}{P^2}\sum_{p=1}^{P}\mathbb{E}[\|g_{s,p}^{t,i}(\mathbf{x}_m^{t,i})\|^2] - \frac{1}{P}\|\nabla_{x_s} f_\lambda^t(\mathbf{x}_m^{t,i})\|^2$$
$$\leq \frac{1}{P}\left[2d_s\mathbb{E}[\|\nabla_{x_s} F(\mathbf{x}_m^{t,i};\xi_m^t)\|^2] + \frac{L^2}{2}\lambda^2 d_s^3\right] - \frac{1}{P}\mathbb{E}[\|\nabla_{x_s} f_\lambda^t(\mathbf{x}_m^{t,i})\|^2]$$
$$\leq \frac{1}{P}\left[2d_s(\mathbb{E}[\|\nabla_{x_s} f_m^{t,i}\|^2] + \sigma_s^2) + \frac{L^2}{2}\lambda^2 d_s^3\right] - \frac{1}{P}\mathbb{E}[\|\nabla_{x_s} f_\lambda^t(\mathbf{x}_m^{t,i})\|^2]$$
$$= \frac{1}{P}\left[2d_s\mathbb{E}[\|\nabla_{x_s} f_m^{t,i}\|^2] + 2d_s\sigma_s^2 + \frac{L^2}{2}\lambda^2 d_s^3 - \mathbb{E}[\|\nabla_{x_s} f_\lambda^t(\mathbf{x}_m^{t,i})\|^2]\right] \qquad (35)$$

Now we bound the squared norm of the variance:

$$\mathbb{E}[\|\nabla_{x_s} f_m^{t,i}\|^2] = \mathbb{E}[\|\nabla_{x_s} f_m^{t,i} - \nabla_{x_s} f_m^t + \nabla_{x_s} f_m^t - \nabla_{x_s} f(\mathbf{x}^t) + \nabla_{x_s} f(\mathbf{x}^t)\|]^2$$
$$\leq 3\mathbb{E}[\|\nabla_{x_s} f_m^{t,i} - \nabla_{x_s} f_m^t\|]^2 + 3\mathbb{E}[\|\nabla_{x_s} f_m^t - \nabla_{x_s} f(\mathbf{x}^t)\|^2] + 3\mathbb{E}[\|\nabla_{x_s} f(\mathbf{x}^t)\|^2]$$
$$\leq 3L^2\mathbb{E}[\|x_{s,m}^{t,i} - x_{s,m}^t\|^2] + 3\mathbb{E}[\|\nabla_{x_s} f(\mathbf{x}^t)\|^2] + 3\epsilon^2 \qquad (36)$$

Substituting (36) into (35), and we finish the proof.

**Lemma D.3** (Bounds on multiple update steps). *If $\eta_s^t \leq \frac{\sqrt{P}}{\tau L \sqrt{24(P+d_s/\tau)}}$, we have*

$$\sum_{i=0}^{\tau-1} \mathbb{E}[\|x_{s,m}^{t,i} - x_{s,m}^t\|^2] \leq 24(\eta_s^t)^2(\tau^3 + \tau^2 d_s/P)\mathbb{E}[\|\nabla_{x_s} f(\mathbf{x}^t)\|^2] + 24(\eta_s^t)^2(\tau^3 + \tau^2 d_s/P)\epsilon^2$$

$$+ \frac{8(\eta_s^t)^2 \tau^2 d_s \sigma_s^2}{P} + 2(\eta_s^t)^2 \tau^3 L^2 \lambda^2 d_s^3$$

*proof:*

We first apply the update formula:

$$\sum_{i=0}^{\tau-1} \mathbb{E}[\|x_{s,m}^{t,i} - x_{s,m}^t\|^2] = \sum_{i=0}^{\tau-1} (\eta_s^t)^2 \mathbb{E}[\|\sum_{j=0}^{i-1} G_{s,m}^{t,j}(x_c^t, x_s^{t,j})\|^2]$$

By the property martingale difference sequence, we have

$$\mathbb{E}[\|\sum_{j=0}^{i-1} G_{s,m}^{t,j}\|^2] \leq 2\mathbb{E}[\|\sum_{j=0}^{i-1} \nabla_{x_s} f_{m,\lambda}^{t,j}\|^2]2 + \sum_{j=0}^{i-1} \mathbb{E}[\|G_{s,m}^{t,j} - \nabla_{x_s} f_{m,\lambda}^{t,j}\|^2]$$

$$\leq 2i \sum_{j=0}^{i-1} \mathbb{E}[\|\nabla_{x_s} f_{m,\lambda}^{t,j}\|^2] + 2 \sum_{j=0}^{i-1} \mathbb{E}[\|G_{s,m}^{t,j} - \nabla_{x_s} f_{m,\lambda}^{t,j}\|^2] \qquad (37)$$

We thus have

$$\sum_{i=0}^{\tau-1} \mathbb{E}[\|x_{s,m}^{t,i} - x_{s,m}^t\|^2] \leq 2 \sum_{i=0}^{\tau-1} (\eta_s^t)^2 \left( i \sum_{j=0}^{i-1} \mathbb{E}[\|\nabla_{x_s} f_{m,\lambda}^{t,j}\|^2] + \sum_{j=0}^{i-1} \mathbb{E}[\|G_{s,m}^{t,j} - \nabla_{x_s} f_{m,\lambda}^{t,j}\|^2] \right)$$

$$\leq 2(\eta_s^t)^2 \tau^2 \sum_{i=0}^{\tau-1} \mathbb{E}[\|\nabla_{x_s} f_{m,\lambda}^{t,i}\|^2] + (\eta_s^t)^2 \tau \sum_{i=0}^{\tau-1} \mathbb{E}[\|G_{s,m}^{t,i} - \nabla_{x_s} f_{m,\lambda}^{t,i}\|^2],$$

$$(38)$$

where the last inequality is by the following equations:

$$\sum_{i=0}^{\tau-1} \sum_{j=0}^{i-1} iX_j = \sum_{j=0}^{\tau-1} (\sum_{i=m}^{\tau-1} i)X_j \leq \sum_{j=0}^{\tau-1} \tau^2 X_j$$

And

$$\sum_{i=0}^{\tau-1} \sum_{j=0}^{i-1} X_j = \sum_{j=0}^{\tau-1} (\sum_{i=m}^{\tau-1}) X_j \leq \sum_{j=0}^{\tau-1} \tau X_j$$

Substituting in (34):

$$\sum_{i=0}^{\tau-1} \mathbb{E}[\|x_{s,m}^{t,i} - x_{s,m}^t\|^2]$$

$$\leq 2(\eta_s^t)^2 \frac{P\tau^2 - \tau}{P} \sum_{i=0}^{\tau-1} \mathbb{E}[\|\nabla_{x_s} f_{m,\lambda}^{t,i}\|^2]$$

$$+ 2(\eta_s^t)^2 \tau \sum_{i=0}^{\tau-1} \left( \frac{6d_s}{P} \mathbb{E}[\|\nabla_{x_s} f(\mathbf{x}^t)\|^2] + \frac{6L^2 d_s}{P} \mathbb{E}[\|x_{s,m}^{t,i} - x_{s,m}^t\|^2] + \frac{6d_s \epsilon^2}{P} + \frac{2d_s \sigma_s^2}{P} + \frac{L^2 \lambda^2 d_s^3}{2P} \right)$$

$$(39)$$

From Lemma B.1 we have

$$\mathbb{E}[\|\nabla_{x_s} f_{m,\lambda}^{t,i}\|^2]$$

$$\leq 2\mathbb{E}[\|\nabla_{x_s} f_m^{t,i}\|^2] + \frac{L^2}{2}\lambda^2 d_s^3$$

$$\leq 2\mathbb{E}[\|\nabla_{x_s} f_m^{t,i} - \nabla_{x_s} f_m^t + \nabla_{x_s} f_m^t - \nabla_{x_s} f(\mathbf{x}^t) + \nabla_{x_s} f(\mathbf{x}^t)\|^2] + \frac{L^2}{2}\lambda^2 d_s^3$$

$$\leq 6\mathbb{E}[\|\nabla_{x_s} f_m^{t,i} - \nabla_{x_s} f_m^t\|^2] + 6\mathbb{E}[\|\nabla_{x_s} f_m^t - \nabla_{x_s} f(\mathbf{x}^t)\|^2] + 6\mathbb{E}[\|\nabla_{x_s} f(\mathbf{x}^t)\|^2] + \frac{L^2}{2}\lambda^2 d_s^3$$

$$\leq 6\mathbb{E}[\|\nabla_{x_s} f(\mathbf{x}^t)\|^2] + 6\epsilon^2 + 6L^2\mathbb{E}[\|x_{s,m}^{t,i} - x_{s,m}^t\|^2] + \frac{L^2}{2}\lambda^2 d_s^3 \tag{40}$$

Substitute into (39):

$$\sum_{i=0}^{\tau-1} \mathbb{E}[\|x_{s,m}^{t,i} - x_{s,m}^t\|^2]$$

$$\leq 2(\eta_s^t)^2 \frac{P\tau^2 - \tau}{P} \sum_{i=0}^{\tau-1} \left( 6\mathbb{E}[\|\nabla_{x_s} f(\mathbf{x}^t)\|^2] + 6L^2\mathbb{E}[\|x_{s,m}^{t,i} - x_{s,m}^t\|^2] + 6\epsilon^2 + \frac{L^2}{2}\lambda^2 d_s^3 \right)$$

$$+ 2(\eta_s^t)^2 \tau \sum_{i=0}^{\tau-1} \left( \frac{6d_s}{P}\mathbb{E}[\|\nabla_{x_s} f(\mathbf{x}^t)\|^2] + \frac{6L^2 d_s}{P}\mathbb{E}[\|x_{s,m}^{t,i} - x_{s,m}^t\|^2] + \frac{6d_s\epsilon^2 d_s}{P} + \frac{2d_s\sigma_s^2}{P} + \frac{L^2\lambda^2 d_s^3}{2P} \right)$$

$$\leq \frac{12(\eta_s^t)^2\tau^3(P + d_s/\tau)}{P}\mathbb{E}[\|\nabla_{x_s} f(\mathbf{x}^t)\|^2] + \frac{12(\eta_s^t)^2\tau^2 L^2(P + d_s/\tau)}{P} \sum_{i=0}^{\tau-1}\mathbb{E}[\|x_{s,m}^{t,i} - x_{s,m}^t\|^2]$$

$$+ \frac{12(\eta_s^t)^2\tau^3(P + d_s/\tau)\epsilon^2}{P} + \frac{4(\eta_s^t)^2\tau^2 d_s\sigma_s^2}{P} + (\eta_s^t)^2\tau^3 L^2\lambda^2 d_s^3$$

Rearranging the terms, we have

$$(1 - \frac{12(\eta_s^t)^2\tau^2 L^2(P + d_s/\tau)}{P}) \sum_{i=0}^{\tau-1}\mathbb{E}[\|x_{s,m}^{t,i} - x_{s,m}^t\|^2]$$

$$\leq 12(\eta_s^t)^2(\tau^3 + \tau^2 d_s/P)\mathbb{E}[\|\nabla_{x_s} f(\mathbf{x}^t)\|^2] + 12(\eta_s^t)^2 L^2(\tau^3 + \tau^2 d_s/P)\epsilon^2$$

$$+ \frac{4(\eta_s^t)^2\tau^2 d_s\sigma_s^2}{P} + (\eta_s^t)^2\tau^3 L^2\lambda^2 d_s^3$$

where we moved the term $\mathbb{E}[\|x_{s,m}^{t,i} - x_{s,m}^t\|^2]$ to the left in the last inequality. Let $\eta_s^t \leq \frac{\sqrt{P}}{\tau L\sqrt{24(P + d_s/\tau)}}$, we have the coefficient on the L.H.S larger than $\frac{1}{2}$. Thus, we complete the proof.

# E    Additional Experiments

To investigate the interplay between splitting strategy and unbalanced update frequency $\tau$, we conduct an ablation study examining various combinations of $\tau$ values and cutting layers using OPT-1.3B on the SST-2 dataset. To isolate the effects of our core mechanism from confounding factors inherent in federated settings, such as data heterogeneity and client variability, we employ a simplified `MU-Split` configuration with a single client.

Table 4 shows the total communication round required to attain $85\%$ accuracy across different cut layers and values of $\tau$. For a fixed cut layer (e.g. $L_c = 2$), setting $\tau = 4$ reduces communication rounds by more than half compared to the baseline without unbalanced updates. Crucially, our results reveal a clear trade-off between $\tau$ and $L_c$. When $L_c$ is fixed, increasing $\tau$ initially improves convergence, but excessive server updates eventually lead to diminishing or adverse effects. Conversely, when fixing $\tau$ and tuning the cut layer, convergence consistently improves as $L_c$ decreases, indicating a deeper server-side model is beneficial for model performance. Moreover, the optimal value of $\tau$ shifts higher as $L_c$ moves earlier in the model. These trends confirm our theoretical insight in Section 4: to fully exploit server-side acceleration, the model partition must scale with the number of server iterations.

Table 5 presents the final accuracy after 1,500 training steps under different combinations of split layers and $\tau$. Consistent with the observations in Table 4, for a fixed split layer, increasing $\tau$ initially improves the final accuracy but eventually leads to a decline. However, unlike Table 4, when varying the split layer, the highest accuracy is consistently achieved at $\tau = 2$ or $\tau = 3$. This pattern aligns with our theoretical analysis in Section 4: although a larger $\tau$ can accelerate convergence, it does not necessarily yield smaller loss value, which is strongly connected to better final accuracy. In practice, selecting appropriate values for $\tau$ and the split layer requires balancing multiple factors, including desired training time, target accuracy, and device memory constraints.

Table 4: Ablation study of influence of $\tau$ and cutting layer on communication rounds

| Split Layer | $\tau = 1$ | $\tau = 2$ | $\tau = 3$ | $\tau = 4$ | $\tau = 5$ | $\tau = 6$ |
|---|---|---|---|---|---|---|
| 2 | 38 | 17 | 19 | **16** | 18 | 18 |
| 4 | - | 18 | **16** | 22 | 20 | 33 |
| 8 | - | 23 | **22** | 26 | 22 | 32 |
| 12 | - | **22** | 32 | 25 | 29 | 32 |
| 16 | - | **21** | 29 | 28 | 40 | 36 |

Table 5: Ablation study of influence of $\tau$ and cutting layer on final accuracy

| Split Layer | $\tau = 1$ | $\tau = 2$ | $\tau = 3$ | $\tau = 4$ | $\tau = 5$ | $\tau = 6$ |
|---|---|---|---|---|---|---|
| 2 | 88.75 | 88.97 | **90.90** | 87.95 | 87.05 | 88.52 |
| 4 | - | 89.09 | **89.89** | 87.05 | 86.93 | 89.04 |
| 8 | - | **90.34** | 90.11 | 89.50 | 89.54 | 88.30 |
| 12 | - | 89.20 | **89.43** | 88.41 | 88.41 | 88.43 |
| 16 | - | **88.98** | 88.75 | 87.95 | 88.41 | 87.99 |

# F  Choice of Hyperparameters

Table 6: Hyperparameters

| PARAMETER | VALUE | EXPLANATION |
|---|---|---|
| $\eta_g$ | 0.3 | *Global aggregation learning rate* |
| $\eta_s$ | 0.01 | *Server learning rate* |
| $\eta_c$ | 0.005 | *Client learning rate* |
| $\lambda$ | 0.005 | *Scale of perturbation for ZOO* |
| $B$ | 32 | *Batch size* |

