# OpenReview forum: "Towards Straggler-Resilient Split Federated Learning: An Unbalanced Update Approach"
_NeurIPS.cc/2025/Conference — NeurIPS 2025 poster_

### Official Review · Reviewer_QcJK · 2025-06-20

**Clarity:** 3
**Significance:** 3
**Originality:** 3
**Rating:** 5
**Confidence:** 4

**Summary:**

With the growing attention to the importance of federated learning (FL), a hybrid approach called Split Federated Learning (SFL) has been proposed to overcome the limitations of traditional FL and Split Learning (SL). However, even in SFL, overcoming the heterogeneity among clients and enabling efficient training remains a challenging task.
This paper proposes a method to enhance the efficiency of SFL by allowing the server to compensate for the slower clients' performance. Specifically, the server supports clients with limited computational resources, thus addressing the imbalance among heterogeneous clients.
Additionally, by incorporating ZO optimization, this paper presents a memory-efficient design suitable for SFL frameworks.

**Questions:**

There seems to be a reason for using perturbed parameters for gradient estimation on the server side and perturbed embeddings on the client side. Is it not possible to unify the gradient estimation method to perturbed parameters or perturbed embeddings for both the server and the clients? Alternatively, could they be reversed?

In the sentence,
where delta_{...} denotes the loss difference obtained from the perturbed embeddings.
appears to include a typo. It seems to be perturbed parameters.

**Ethical Concerns:**

["NO or VERY MINOR ethics concerns only"]

**Final Justification:**

This reviewer believes this paper has dealt with an important topic related to Split Federated Learning and has addressed it appropriately. The authors have responded to this reviewer’s questions, which has clarified my understanding. The concerns and opinions of the other reviewers have also been taken into consideration. This reviewer would like to maintain the positive rating.

**Limitations:**

Yes

**Paper Formatting Concerns:**

That’s fine with me.

**Quality:**

3

**Strengths And Weaknesses:**

To address the straggler problem among clients, this paper proposes a method that effectively utilizes the server. It addresses memory constraints on the client side through ZO optimization. The advantages of the proposed method are validated through comparisons with recent works, and its convergence is supported by theoretical analysis. From this reviewer's perspective, the proposed method presents a novel contribution.

---

> ### Author Rebuttal · Authors · 2025-07-31
>
> Thank you so much for your encouragement and recognition of our theoretical contribution! We're happy that our research has raised your interest. We will further refine the detail in our work according to suggestions.
>
> **Clarification on "perturbed parameters and embeddings".**
>
> The "perturbed embedding" $h^{t+,-}$ refers to "parameter-perturbed embedding", which is obtained by perturbing the **client-side parameters** during the forward pass.
>
> For server-side updates, we take $h^t$ as input when perturbing the **server-side parameters** to obtain ZO gradient estimates. Meanwhile, $h^{t+,-}$ is used on server-side to calculate $\delta^t$(Eq. 6), which will be sent back for client-side gradient estimation and model updates. That being said, in our algorithm design, both client and server follow the same unified ZO update protocol.
>
> This also helps explain our statement that “$\delta^t$ obtained from the perturbed embeddings”, since $\delta^t$ is obtained by using $h^{t+,-}$ as input. This information is used for client-side updates and does not require perturbing the server-side parameters.
>
> Hope our explanation addresses your question. Thank you again for your careful review!

---

> > ### Comment · Reviewer_QcJK · 2025-08-04
> >
> > Thank you to the authors for the additional explanation regarding my question. My concern has been addressed.
> > I will give further consideration to the other reviewers' comments.

---

### Official Review · Reviewer_PKLf · 2025-06-24

**Clarity:** 3
**Significance:** 3
**Originality:** 2
**Rating:** 3
**Confidence:** 4

**Summary:**

This work proposed an algorithm to solve the straggler issue in split federated learning. Meanwhile, a zeroth-order based model update approach was proposed to reduce the memory consumption and load at the client side. The authors provided theoretical analysis on the convergence of the proposed algorithm and evaluate its performance using experiments.

**Questions:**

Please see above weakness.

**Ethical Concerns:**

["NO or VERY MINOR ethics concerns only"]

**Final Justification:**

The motivation and main idea of this work appear to be promising. Despite this, there might be some limitations regarding the theoretical analysis on how straggler affects the convergence. Also, although the ZOO is a promising approach for reducing the memory of clients, some prior work have incorporated it into federated learning, so the main contribution of this work should be related to analyzing the impact of the stragglers on gradient estimation in ZOO. Thus, I would choose to maintain the score.

**Limitations:**

Yes

**Paper Formatting Concerns:**

No.

**Quality:**

2

**Strengths And Weaknesses:**

Strength:
- This paper is well-written.
- The incorporation of zeroth-order optimization is valid and interesting.
- The theoretical analysis is solid.
- The experimental results validate that the proposed approach can achieve a better performance than baselines.

Weakness:
- Although I understand the proposed algorithm, it is hard to see how the zeroth-order optimization addresses the straggler issue. Is it related to only when and how many rounds the updates are performance? If this is the case, the contribution on the algorithm design might be minor, as conventional zeroth-order optimization was used.
- Zeroth-order optimization usually imposes multiple rounds of perturbations and hence increase the duration of each round. How is this problem solved in this work?
- Is non-iid issue considered in this work? Straggler is an issue only when non-iid data is considered; otherwise, asynchronous aggregation does not impose any model accuracy degradation. As suggested in the unbalanced server update, different models may experience different number updates. Will the non-iid data be a problem?
- Zeroth-order optimization is essentially an approximation. It is surprised to see the algorithm experience accuracy improvement. An ablation study may be needed to see if the zeroth-order optimization leads to any degradation, i.e., if the performance improvement comes from the schedule design (i.e., multiple updates) or the zeroth-order optimization. If it is from the schedule design, I am not sure about the necessity and significance of introducing the zeroth-order optimization.

---

> ### Author Rebuttal · Authors · 2025-07-31
>
> Thank you so much for your thoughtful feedback and for recognizing the strengths of our theoretical work. Below, we address your main concern regarding **the necessity of incorporating ZOO with our unbalanced update strategy**.
>
> **On the Role of ZOO in Addressing Straggler Issues:**
>
> 1. **Theoretical dependency of ZOO in solving straggler issue.** Our straggler resilience result is derived from the sublinear speed-up convergence rate presented in Corollaries 4.4 and 4.8. Achieving this sublinear speed-up w.r.t the unbalanced update factor $\tau$ requires a coordinated cutting strategy, where the dimension split satisfies $d_c = \sqrt{\frac{d}{\tau}}$ and $d_s = d - \sqrt{\frac{d}{\tau}}$. And this conclusion is only validated under ZOO, **since only the convergence rate of ZOO is dependent on dimension** $d$. This implies that the mitigation of straggler issue is **not merely due to multiple updates**, **but is tied to the incorporation of ZOO.**
> 2. **Practical suitability of ZOO for straggler settings.** ZOO also offers practical advantages in straggler scenarios, particularly for memory-constrained edge devices. Compared to FOO, **ZOO can significantly reduce memory usage**[17, 18]. This aligns well with the realities of resource-constrained edge devices, where memory and computation overhead can introduce time delay that further exacerbates the straggler problem[a,b,c].
>
> **More interesting findings uncovered by incorporating ZOO with Unbalanced Update:**
>
> 1. **ZOO enables theoretical interpretation between cutting strategy and unbalanced update.**  A key novelty of our work lies in providing a theoretical explanation for how the choice of splitting layer coordinates with unbalanced updates $\tau$ affects convergence rate. Compared to FOO, the convergence rate of ZOO is related to the model dimensionality $d$, which helps us link the convergence performance to the cutting layer position. Specifically, adjusting the splitting layer allows us to optimize convergence behavior, which provides a principled guideline for choosing the cutting layer in practice. In contrast, such a theoretical connection between the splitting layer and convergence behavior is difficult to uncover in the FOO setting without the dimension dependency of ZOO convergence rate.
> 2. **Dimension-Free ZOO achieved by Unbalanced Updates.**
> Our work is able to eliminate the dimension dependence in ZOO by employing unbalanced updates. Based on Corollaries 4.4 and 4.8, by appropriately scaling the unbalanced update factor $\tau$ to the same order as the model dimension dimension $d$, the convergence rate becomes independent of $d$. This directly addresses a key limitation of traditional ZOO methods, where convergence rates typically deteriorate with increasing dimension.
>
> **Regarding your question about whether our algorithm design is just simply combines ZOO with Unbalanced updates.**
>
> Firstly, we would like to clarify that our algorithm does not merely apply conventional ZOO within SFL, but rather introduces a tailored design that explicitly addresses the straggler issue by reducing synchronization delays through a novel **Eager forward** transmission scheme.
>
> In traditional ZOO, updates often require waiting for all perturbations to be computed before proceeding. We refer to it as the “Lazy” forward. **In contrast, our “Eager” forward method follows a sequential update process that effectively reduces waiting time.** Specifically, during each forward pass on the client, parameter-perturbed embeddings are immediately transmitted to the server as they are computed, without waiting for the full set of parameter-perturbed embeddings ($h, h^+, h^-$) to complete. For instance, the unperturbed embedding $h$ is first computed and sent to the server, which can immediately begin performing $\tau$ local updates with $h$ as input, while the clients continue computing.
>
> To have a more direct observation about our eager methods in minimizing waiting time, we conduct experiment using OPT1.3B on SST-2, and calculate the required time using different forwarding manner. As shown in Table 4, where we set $\tau=10$, our eager scheme **reduces the overall forward time by nearly half.** This confirms the effectiveness of our eager update manner in minimizing synchronization delays by utilizing the overlapping computation time.
>
> Table 4 Required Time for Different Forward Manner
> | Dataset | Client Forward Time (ms) | Server Forward Time (ms) | Eager Forward(ms) | Lazy Forward(ms) |
> | --- | --- | --- | --- | --- |
> | SST-2 | 61.18 | 5.41 | 673 | 1214 |
>
> > Zeroth-order optimization usually imposes multiple rounds of perturbations and hence increase the duration of each round. How is this problem solved in this work?
> >
>
> Towards this question, we believe that our **Eager Forward** design directly addresses this issue by performing **sequential transmission of parameter-perturbed embeddings**. This eliminates the need to wait for all perturbations to complete before the server can begin updates, thus significantly reducing time delay caused by multiple perturbations.
>
> Moreover, from a theoretical perspective, multiple perturbations lead to **faster convergence rates** w.r.t  $p$ [18]. Although applying multiple perturbations can introduce time delay for each local round, it can accelerate global convergence, thereby helping to mitigate the straggler issue to some extent.
>
> **Regarding non-iid issue:**
>
> **In this paper, we do not focus on the non-IID issue**. Non-IID data is often used to model data heterogeneity, which is indeed an important challenge in federated learning. **However, our focus is on the straggler problem, which stems from system heterogeneity** [8-14]. This refers to disparities in computational capabilities and network latency among clients, as we stated in the first paragraph of our paper.
>
> We introduce variable time delays across all the clients which is a widely adopted approach to simulate the impact of stragglers in distributed learning system. This method is well-supported in prior literature [8–14]. To establish a more realistic model that reflects the randomness of clients' time delay, we consider an exponential distribution model, which has been widely used to capture the computation delay for distributed clusters [d,e,f].
>
> As we acknowledge the importance of data heterogeneity in federated learning. To evaluate the robustness of our method under **non-IID data**, we incorporate a high data heterogeneity setup with shard=2.
>
> As shown in Table 5, although all methods experience some performance degradation under non-IID settings, our proposed algorithm continuously outperforms the baseline GAS[8], which is specifically designed for non-IID scenarios. This suggests that our method is not only effective in mitigating straggler effects but also exhibits robustness to data heterogeneity.
>
> Table 5 Impact of data heterogeneity on SFL performance
> | Task | GAS | MU-SplitFed |
> | --- | --- | --- |
> | CIFAR-10(homogeneous) | 75.28 | 77.86 |
> | CIFAR-10(shard=2) | 66.23 | 66.83 |
> | Fashion-MNIST(homogeneous) | 83.70 | 85.45 |
> | Fashion-MNIST(shard=2) | 75.10 | 76.27 |
>
> **Questions about performance boost:**
>
> To avoid any misunderstanding, we would like to first clarify that, as stated in Line 276, **we reimplemented all baselines using ZOO** to ensure a fair comparison across methods.
>
> To address your concern, we conducted an ablation study on the SST-2 dataset using OPT-1.3B. Specifically, we implemented MU-Split in both its FO and ZO versions. This simplified setup includes only one client and one server, isolating the impact of the unbalanced update strategy. As shown in Table 6,7, we observe consistent performance improvement in both FO and ZO versions, indicating that the **accuracy gain primarily stems from the unbalanced server update schedule**. This further confirms the effectiveness of our proposed unbalanced update strategy in enhancing convergence and accuracy.
>
> Table 6  Best accuracy after 100 rounds of MU-Split(OPT-1.3B, Zeroth Order)
> | Split Layer | $τ=1$ | $τ=2$ | $τ=3$ | $τ=4$ | $τ=5$ | $τ=6$ |
> | --- | --- | --- | --- | --- | --- | --- |
> | 2 | 88.75 | 88.97 | **90.90** | 87.95 | 87.05 | 88.52 |
> | 4 | — | 89.09  | **89.89**  | 87.05  | 86.93  | 89.04 |
> | 8 | — | **90.34**  | 90.11  | 89.50 | 89.54 | 88.30 |
> | 12 | — | 89.20   | **89.43**  | 88.41  | 88.41 | 88.34 |
> | 16 | — | **88.98** | 88.75  | 87.95  | 88.41 | 87.99 |
>
> Table 7  Best accuracy after 100 rounds of MU-Split(OPT-1.3B, First Order)
> | Split Layer | $τ=1$ | $τ=2$ | $τ=3$ | $τ=4$ | $τ=5$ | $τ=6$ |
> | --- | --- | --- | --- | --- | --- | --- |
> | 2 | 88.06 | **91.82** | 88.75 | 88.07 | 87.73 | 88.30 |
> | 4 | — | 89.43 | 86.7 | 89.89 | **90.23** | 86.59 |
> | 8 | — | **89.89** | 87.61 | 84.20 | **89.89** | 88.86 |
> | 12 | — | 87.95 | 86.82 | 87.84 | **89.66** | 86.70 |
> | 16 | — | 87.84 | 88.64 | 86.70 | **89.77** | 88.86 |
>
> a. “Thinking Forward: Memory-Efficient Federated Finetuning of Language Models”
>
> b. “Breaking the Memory Wall for Heterogeneous Federated Learning via Progressive Training”
>
> c. “Breaking the Memory Wall for Heterogeneous Federated Learning via Model Splitting”
>
> d. “Straggler-resilient federated learning: Leveraging the interplay between statistical accuracy and system heterogeneity”
>
> e. "Speeding up distributed machine learning using codes."
>
> f. "Coded computation over heterogeneous clusters.”

---

> > ### Comment · Reviewer_PKLf · 2025-08-03
> >
> > Thank you for your detailed information. Most of my comments have been addressed.
> >
> > I have questions on the response to the non-IID issue. Based on my understanding, if IID data is considered, then federated learning and split federated learning become a distributed version of centralized learning, under which the straggler is no longer an issue. That is, consider an extreme example where everyone has the same datasets. In this case, no matter which client submits the trained model, it makes no difference to the training result. Also, no optimization regarding the aggregation frequency and communication round is needed.  Thus, if IID data is considered, then how and why does the straggler issue affect the convergence (e.g., in the convergence analysis)? It would be great if the authors could clarify this point.

---

> > > ### Author Response · Authors · 2025-08-05
> > >
> > > Thanks for clarifying your concern to us. A more accurate description towards your question is: **the system-level heterogeneity is our main addressing issue**. Specifically, the difference in **computational capabilities** or **network conditions** among all the clients. For instance, a smartwatch as an edge device could require significantly longer computing time than an iPhone, making the smartwatch a straggler regardless of data distribution.
> > >
> > > Regarding your question, we'd like to clarify several important points:
> > >
> > > 1. **Straggler remains an problem regardless of IID or non-IID data in distributed learning.**  As long as synchronization is required in the distributed system, stragglers remain a critical problem. While non-IID data can **exacerbate** this straggler issue, the fundamental problem persists even with IID data. To illustrate this, consider the extreme case raised by the reviewer where all clients share the same dataset. Even in this scenario, synchronization decisions at the server still involve trade-offs. For example, if there are 10 clients but only 2 have returned their results so far, the server must still decide whether to aggregate the global model immediately (abandon stragglers) or wait for more clients. While this example assumes shared data across clients, the number of participating clients per round still affects convergence and must be taken into account.
> > > 2. **Federated Learning with IID data is not equivalent to “Distributed version of centralized learning”.** In Federated Learning setting, each client holds a separate **private dataset**. Even when the data is IID (i.e. drawn from the same distribution), the actual data samples differ across clients. In such cases, both the number of participating clients per round and the frequency with which each client contributes are critical. In the illustrated case, if the same 2 out of 10 clients are always stragglers and are consistently dropped, the effective optimization is still over the remaining 8 clients’ data. Therefore, even under IID settings, this can introduce bias or degrade convergence performance.
> > > 3. **Our algorithm includes the non-IID case.** Theoretically, we use bounded heterogeneity (Assumption A.3) to represent data heterogeneity. Our theoretical results include error terms related to data heterogeneity that reflect the impact of data heterogeneity(see Corollary 4.8), where  $\epsilon^2$ denotes the bounded data heterogeneity.
> > >
> > > **Summary:** The straggler issue fundamentally arises from system heterogeneity (asynchronous computation and communication delays) and persists in distributed systems regardless of whether data is IID or non-IID. However, non-IID data can amplify the negative impact of stragglers by affecting convergence speed and accuracy. This is supported by both our theoretical analysis (showing additional error terms for non-IID data) and experimental results comparing IID and non-IID settings. In conclusion, non-IID issue is not the primary factor in the purpose of our algorithm design, but this direction is very important as well and can be a great future direction.

---

> > > > ### Comment · Reviewer_PKLf · 2025-08-07
> > > >
> > > > Thank you very much for the detailed response. Based on point 1, I would assume the convergence depends on the data quantity (as in author's example, the data of those clients with the same/similar dataset) and aggregation frequency (as in the example, whether to aggregate after two clients or four clients). Based on point 3, since heterogeneity is accounted, there are some degree of data heterogeneity considered in the bound. Combining these two points, I would guess how straggler affects the data heterogeneity was not characterized, which appears to be an issue regarding the theoretical analysis. Despite this, I like the idea of ZOO and split learning, which can help to reduce the memory and computation load of the clients.
> > > >
> > > > Thank the authors again for their time and the detailed response.

---

> > > > > ### Author Response · Authors · 2025-08-07
> > > > >
> > > > > Thank you very much for your insightful comments and for recognizing our idea of ZOO and split learning to reduce client-side resource demands.
> > > > >
> > > > > You raise an important point regarding the effect of stragglers on data heterogeneity. As you correctly noted, our current analysis, which, like much of the existing literature in split learning and federated learning, provides theoretical bounds under static data heterogeneity assumptions, but does not explicitly characterize how the presence of stragglers interacts with data heterogeneity. This is indeed a challenging and largely open problem: in dynamic systems with varying client participation (e.g., due to stragglers), it becomes difficult to formally capture the dynamic of data heterogeneity due to stragglers unless strong assumptions are made about the stragglers. To the best of our knowledge, existing theoretical works do not provide convergence guarantees that account for potential correlations between straggling behavior and data heterogeneity. We agree that addressing this interaction would be both meaningful and technically demanding, likely requiring new analytical tools. We consider this an important direction for future work.
> > > > >
> > > > > We sincerely appreciate your constructive feedback and your positive reception of our core ideas. In the meantime, please do not hesitate to let us know if you have any further comments or questions.

---

### Official Review · Reviewer_WuCT · 2025-06-29

**Clarity:** 3
**Significance:** 3
**Originality:** 2
**Rating:** 4
**Confidence:** 3

**Summary:**

This paper propose a straggler-resilient SFL algorithm by incorporating zeroth-order optimization.

**Questions:**

**Questions**:

I have checked the proofs of MU-Split (Appendix B). Some questions and suggestions are as follows:

1. Lines 110-111.

   > Unlike FL, SFL requires the Split Server to receive and process distinct activation vectors from each client, which are concatenated rather than aggregated [8].

   This is inconsistent with the description in Figure 1 and Algorithm 1, where only aggregation (no concatenation) is performed.

2. Equation (17). Should the last inequality be equality?

3. The forth inequality of the equation between Lines 561 and 562. Are $f_\lambda$ and $f_\lambda^{t,i}$ equivalent?

4. The third equality of Equation (22). Should $f_\lambda^t (x_c^t, x_c^t)$ be $f_\lambda^t(x_c^t,x_s^t)$?

5. The first inequality of Equation (32). Does $\mathbb{E}[X^2] = (\mathbb{E}X)^2 + \text{Var}(X)$ used here? I think it should use Jensen's inequality. Please see Appendix C.2 in the following paper.

    Convergence Analysis of Sequential Federated Learning on Heterogeneous Data, NeurIPS, 2023.

**Suggestions**:

1. Assumption 4.1. In MU-SplitFed, there are multiple loss functions. It would be better to specify which function is assumed to be $L$-smooth clearly.

2. Lines 175-176.

   > It is also worth noting that our convergence analysis does not require convexity of the loss function, adding to the complexity of the proof.

   In my opinion, this note is not necessary (even a little misleading). If the convexity assumption is given, we can often get tighter bounds under other milder assumptions. Besides, the convexity assumption enables the use of more advanced techniques. So, it is not appropriate to say that the analysis of the non-convex settings is more complex than the convex settings without further discussion.

3. Theorem 4.7. It seems that Assumptions 4.3-4.5 are not explicitly provided.

4. Figures. It seems that the figures are not in PDF format? I recommend using the PDF format, which will help the paper be more professional.

5. References. There are some papers which are cited as preprint, but have been published. For example, [2] has been published in NeurIPS, [29] has been published in ICLR.

6. It would be better to see equations as one part of the sentences. For example, for Lines 573-574, please add one comma after Equation (23) and make "Where" in lower case. It would be better to revise the other sentences accordingly, e.g., Lines 615, 634, 673. The omitted punctuations should be added after the equations.

7. Some equations are too long.

8. Some sentences are unclear.

   > Line 567: Similar to the proof in B.4, we substitute in (26) and (31) in order:
   >
   > Line 617: Substituting in (26):
   >
   > Line 618: Further substitute in (31):

For the formatting problems, I think that this page ( https://www.jmlr.org/format/formatting-errors.html ) of JMLR will be helpful.

If the problems I raised above are wrong, please free free to point them out.

**Ethical Concerns:**

["NO or VERY MINOR ethics concerns only"]

**Final Justification:**

This paper proposes MU-SplitFed, which is equipped with the unbalanced updates and zeroth order optimization.

The main concern is focused on the tightness of the bounds (i.e., $O(\eta^2)$ terms).

This is mainly because the original SplitFedv1 (or SFL_v1) (in Thapa et al, (2022)) is equivalent to FedAvg in the absence of model partitioning. So, in my opinion, the bounds of SplitFed may be quite similar to those of FedAvg. However, this does not seem to be the case, based on the existing works (Han et al., 2024) and this paper.

This inconsistency makes me not really convinced of the tightness of the bounds provided in this paper.

However, it is also unfair for me to reject this paper based on this concern given my unfamiliarity with the existing works in split learning and zeroth order optimization.

I hope the authors will reflect on the reasons behind the inconsistency and discuss them in detail in the revised version.

As a consequence, I decide to increase the rating to 4, and decrease the confidence to 3.

**Limitations:**

yes

**Paper Formatting Concerns:**

No.

**Quality:**

3

**Strengths And Weaknesses:**

**Strengths**:

1. This paper combines zeroth-order optimization and SFL.
2. The proposed algorithm allows unbalanced server-client updates. This alleviates the impact of stragglers in SFL.
3. The convergence analysis of the proposed algorithm is provided.
4. This paper is well-written and well-organized.

**Weaknesses**:

1. Convergence analysis of SFL. There has been one paper [2] that has established the convergence guarantees of SFL (including both SFLv1 and SFLv2) for strongly convex, convex and non-convex settings. Thus, for SFL, I recommend comparing the convergence guarantees of this paper with [2]. Furthermore, given [2], the claim that "existing theoretical results in SFL either assume convexity or bounded gradients (Lines 51-52)" need to be revised.

    [2] Convergence Analysis of Split Federated Learning on Heterogeneous Data, NeurIPS, 2024.

2. Convergence analysis of zeroth-order optimization in FL. It would also be beneficial to compare the convergence guarantees of this paper for SFL with ZO methods for FL (e.g., [17,18] mentioned in Line 47).
3. Increased communication. The communication may be increased in the following two aspects: (i) In Mu-SplitFed, the clients send $H_m^t = \{h_m^t, h_m^{t+}, h_{m}^{t-}\}$ to the server, while in vanilla SplitFed (SFLv1), the clients only send $h_m^t$. This increases the communication. (ii)  In vanilla SplitFed, the clients and the server can perform multiple local updates before the aggregation of the local models. However, in Mu-SplitFed, the clients perform only one local update and the server can perform multiple local updates before the aggregation of the local models. This may increase the communication.

---

> ### Author Rebuttal · Authors · 2025-07-31
>
> We really appreciate the time and effort you dedicated to reviewing our work and providing such valuable insights! To facilitate a direct comparison, we have added **Table 3**, which includes the convergence rates and  communication complexity of our proposed MU-SplitFed alongside the SplitFed algorithms in [2] and other FedZO methods.
>
> Table3 Convergence/Communication Complexity Comparison Result
>
> | **Method** | **Convergence Rate** | **Communication Complexity** |
> | --- | --- | --- |
> | FedAvg(FO)[29] | $O(\frac{F+1}{\sqrt{MKT}})$ | $O(\frac{F^2}{MK\epsilon^2})$ |
> | SFL-V1(FO)[2] | $O(\frac{F}{\sqrt{T}})$ | $O(\frac{F^2}{\epsilon^2})$ |
> | **Ours: MU-SplitFed(ZO)** | $O(\frac{\sqrt{d}(F+1)}{\sqrt{MT\tau}})$ | $O(\frac{d F^2}{M\tau\epsilon^2})$ |
> | FedZO[a] | $O(\frac{\sqrt{d}F}{\sqrt{MKT}})$ | $O(\frac{dF^2}{MK\epsilon^2})$ |
> | DecomFL(ZO)[18] | $O(\frac{\sqrt{d}(F+1)}{\sqrt{MKT}})$ | $O(\frac{dF^2}{MK\epsilon^2})$ |
>
> **Weakness 1:**
>
> We initially avoided a direct rate comparison with [2] because their analysis is grounded in **first-order optimization (FOO)**, while our work is based on **zeroth-order optimization (ZOO)**. Given the fundamentally different gradient estimation paradigms, we considered the comparisons less direct. However, we agree that contrasting them will offer more valuable insights, and Table 3 now addresses this.
>
> While [2] provides an important theoretical foundation for SFL, we note two key limitations in their analysis:
> 1. Conflict in the Strongly Convex Case: The results in [2] for the strongly convex setting rely on the assumption of bounded gradients. However, this assumption contradicts strongly convex, making the conclusions theoretically questionable.
> 2. Loose Bounds and Strong Assumption in the Non-Convex Case: Although [2] does present convergence results for the non-convex case, it relies on **stronger assumptions** (e.g., bounded gradients), whereas we use **bounded variance**. Despite using milder assumptions, our analysis achieves a sublinear speed-up w.r.t $\tau$, attributed to our use of more advanced techniques such as properties of martingale difference sequence and tighter equality-based estimations. Interestingly, [2] allows for multiple local updates in each round that should intuitively accelerate convergence. However, their theoretical bounds do not reflect this benefit, suggesting room for tightening.
>
> In contrast, our analysis reveals that both increasing $\tau$ and involving more clients contribute to convergence rate. This reflects a more refined understanding of the interplay between local computation and global coordination in SFL.
>
> Lastly, thank you for catching our inaccurate phrasing in Lines 51–52. We will revise that section to more precisely reflect the limitations of [2] in the updated version of our paper.
>
> **Weakness 2:**
>
> We have now included a detailed comparison of both convergence rates and communication complexities between our proposed method and existing ZOO approaches in federated learning, including one of the earliest works combining ZO with FL [a], and DecomFL [18]. The results are also presented in Table 3.
>
> From this comparison, we observe that our algorithm achieves a convergence rate comparable to those of existing FedZO methods. The key distinction lies in how acceleration is achieved:
>
> While FedZO and DecomFL rely on increasing the number of local update steps $K$ to accelerate convergence, our method introduces a novel acceleration mechanism via our **unbalanced update strategy** parameterized by $\tau$.
>
> To isolate the effect of $\tau$, we consider the setting where $K=1$ across all methods. In this scenario, our algorithm exhibits a **sublinear speed-up w.r.t $\tau$**, highlighting the role of unbalanced updates in improving convergence. Importantly, our method does **not impose any restrictive assumptions** that would hinder generalization. This allows our framework to naturally extend to $K > 1$, enabling multiple local updates and potentially further enhancing convergence.
>
> In summary, this comparison highlights the advantage of our framework in exploiting the unbalanced update ratio $\tau$ to improve convergence, while remaining compatible with and extensible to broader settings explored in the FedZO literature.
>
> **Weakness 3:**
>
> **Regarding your increased communication concern (i):**
>
> We would like to clarify two key facts:
>
> 1. **Per-round backward communication is reduced in MU-SplitFed.** In our proposed algorithm, during the forward pass, the client transmits $H_m^t=h^t, h^{t+},h^{t-}$, which involves three vectors. However, in the backward pass, the client only transmits a **scalar** value $\delta_{c,m}^t$ as defined in Eq. (6). In contrast, in SplitFed-V1 [2], the backward pass requires transmitting the the partial gradient w.r.t. $h^t$. As such, although our forward pass involves slightly more communication, the backward communication is **significantly lighter**, making the total communication per round **comparable**.
> 2. **Our total communication complexity is theoretically lower compared to [2].** As shown in Table1, our method exhibits a **lower overall communication complexity** despite slightly higher per-round communication. This is due to our **unbalanced update strategy**, which plays an important role in reducing the number of global communication rounds. In comparison, the method in [2] incurs a bad communication complexity even fail to reveal the accelerating of $K$ and $M$. Thus, our approach achieves a **reduction in total communication cost** over the entire training process.
>
> **Regarding your question(ii) about client local update steps:**
>
> 1. Simplifying the analysis with $K=1$. To make our unbalanced update strategy easier to understand and analyze, we set $K=1$ in our current framework. However, our method is **not limited to this setting**. It can be naturally extended to support $K>1$. In such cases, during each local $K=j$ round, the client and server would still apply our proposed unbalanced update strategy. After completing $n$ local rounds, the client and server would perform global aggregation via FedServer. This generalization preserves the structure of our method while achieving accelerated convergence.
> 2. No increase in communication complexity. Even with $K=1$, our method **does not suffer from increased communication cost**. As shown in Table 3, the overall communication complexity remains comparable to other FedZO methods. This is because our server-side update parameter **$\tau$** plays a similar role to $K$ in reducing the frequency of global communication. Furthermore, extending our method to support $K>1$ is expected to **further improve both convergence rate and communication efficiency**.
> 3. Lower communication cost for model aggregation. We understand that your concern is because global communication costs are comparably higher than local communication costs. However, our algorithm is designed to **substantially reduce the communication cost of model aggregation**. In particular, it allows us to allocate only a **small portion of the model to the client** (typically we choose $L_c=2$). This is supported by our theoretical analysis: by tuning $\tau$, we can effectively minimize the client-side model size. Additionally, experimental results indicates that our unbalanced update strategy also lead to performance boost(see Supplementary Table 6, 7). In other words, our approach enables the use of a smaller client-side model while still maintaining competitive accuracy, thereby reducing the overall cost of global aggregation. In contrast, Han et al. [2] suggest that a larger portion of the model should be put on client to achieve decent accuracy, which inherently requires higher global communication cost.
>
> **Response to Questions:**
>
> Q1: Thank you for pointing out this inconsistency, and we apologize for the misleading description. The reason we originally referred to concatenation is that certain SFL designs [8,b] use concatenation of activation vectors instead of aggregation. However, we acknowledge that our current implementation follows the aggregation-based approach. To accurately represent our design, we will revise the sentence as follows in the updated version:
> “The straggler issue only delays the global aggregation in traditional FL. In contrast, in SFL, it also impacts local updates due to SFL's relay-based update mechanism. This motivates us to revisit straggler solutions tailored for SFL.”
>
> Q2: Thank you for your comment. To achieve a tighter bound, we apply an equality in the derivation. However, since the final result is expressed as “less than or equal to,” using the inequality is **technically not incorrect**.
>
> Q3: $f_\lambda^t$ and $f_\lambda^{t, i}$ are not equivalent. Specifically, $f_\lambda^t$ denotes the loss function at the t-th global training round, while $f_\lambda^{t, i}$ denotes the loss function at the t-th global training round and i-th server iteration. We will clarify this notation more explicitly in the revised version.
>
> Q4: Thank you for catching this typo, we appreciate your careful reading. We will correct it in the final version.
>
> Q5: Thank you for your comment. We would like to clarify that we did not apply the equation you mentioned in deriving the first inequality of Equation (32). Instead, our analysis follows the **martingale-based approach** under conditional expectation, consistent with the second situation in ***Lemma 4*** of **SCAFFOLD** paper[c]. However, we acknowledge that a scaling factor of 2 was omitted in front of the inequality, and we will correct this in the revised version.
>
> We will follow your suggestions to refine our work in our future version.
>
> Supplement references:
>
> a. “Communication-efficient stochastic zeroth-order optimization for federated learning”
>
> b. “SCALA: Split Federated Learning with Concatenated Activations and Logit Adjustments”
>
> c. “SCAFFOLD: Stochastic Controlled Averaging for Federated Learning”

---

> > ### Comment · Reviewer_WuCT · 2025-08-02
> > **Further comments**
> >
> > Thanks for your detailed rebuttal, which have addressed most of my concerns. Further comments are as follows.
> >
> > I don't read the reviews of other reviewers due to the heavy workload of NeurIPS in such a short time. If you have answered some of them, just pointed them out (no necessary to restate them).
> >
> > According to the rules in this year, the final rating (made in Reviewer-AC Discussions) will not be disclosed to the authors.
> >
> > **Detailed bounds including other factors**:
> >
> > 1. Table 3 (rebuttal): Could the authors kindly provide a detailed bounds including other factors (e.g., intra-client heterogeneity, inter-client heterogeneity and $L$-smoothness) in Table 3? For example, [Ra]'s Tables 1 and 2. The detailed bounds will help me know the difference better.
> >
> >    [Ra] Momentum Benefits Non-Iid Federated Learning Simply and Provably, ICLR, 2024.
> >
> > 2. Table 3 (rebuttal): I am curious to know why a numerical constant $1$ appears in the bounds, such as $O(\frac{F+1}{\sqrt{MKT}})$. This is mainly for my personal learning purposes. I also met them in other papers.
> >
> > **Where the improved bounds benefits from? (i) zeroth-order optimization, (ii) unbalanced updates, or (iii) both.**
> >
> > 1. My weakness 1: Thank you very much for bringing the limitations of [Rb] (i.e., [2] in the original paper) to my attention: (i) they use bounded gradients (one strong assumption); (ii) their bounds do not reflect the benefit of local updates; (iii) [Rb] suggests that a larger portion of the model should be put on client, which is undesirable in practice.
> >
> >    Intuitively, I think (i) the local updates and (ii) the unbalanced updates (proposed in this paper) will also benefit SFL (for both versions 1 and 2) with first-order optimization. Can the authors kindly establish tighter bounds than [Rb] for SFL (with first-order optimization)? This is mainly due to my concern about where the improved bounds benefits from. (i) zeroth-order optimization, (ii) unbalanced updates, or (iii) both. I know this question is beyond the scope of this paper. So I will not discount the value of the paper because of a "no", but a "yes" would increase my appreciation of it.
> >
> >    [Rb] Convergence Analysis of Split Federated Learning on Heterogeneous Data, NeurIPS, 2024.
> >
> > **Further suggestions or questions**:
> >
> > 1. My weaknesses 2 and 3: It seems that the bounds in this paper can be extended for multiple local updates ($K>1$)? I recommend using multiple local updates ($K>1$) if possible.
> > 2. My question 3: Since $f_\lambda^t$ and $f_\lambda^{t,i}$ are not equivalent, back to the fifth inequality of the equation between Lines 561 and 562, it seems that we cannot use $L$-smoothness for $\\|\nabla_{x_s} f_\lambda (x^t) - \nabla_{x_s}f_\lambda^{t,i}(x_c^t,x_s^{t,i}) \\|$.
> >
> > 5. My weakness 3: (i) SplitFed: In forward pass, the client sends 1 vector; in the backward pass, the server sends 1 vectors. (ii) MU-SplitFed: In forward pass, the client sends 3 vectors; in the backward pass, the server sends 0 vectors (one scalar). (iii) MU-SplitFed suffers from higher per-round communication. Note that the vectors in the forward pass and backward pass are the same size. Are they right?

---

> > > ### Author Response · Authors · 2025-08-05
> > >
> > > We have updated a new **Table1**  according to [Ra], covering both SL and SFL settings for ZO and FO methods.
> > >
> > >  Table1 Detailed convergence bound
> > >
> > > |**Method**|**Convergence Rate**|**Communication Complexity**|
> > > |---|---|---|
> > > |MU-Split(ZO)|$O(\frac{\sqrt{d}(F+\sigma^2)}{\sqrt{T\tau}})$|$O(\frac{d(F^2+\sigma^4)}{\tau\epsilon^2})$|
> > > |MU-Split(FO)|$O(\frac{F+\sigma^2}{\sqrt{T\tau}}+\frac{\sigma^2}{T})$|$O(\frac{F^2+\sigma^4}{\tau\epsilon^2}+\frac{\sigma^2}{\epsilon})$|
> > > |MU-SplitFed(ZO)|$O(\frac{\sqrt{d}(F+\sigma^2+\zeta^2)}{\sqrt{MT\tau}})$|$O(\frac{d(F^2+\sigma^4+\zeta^4)}{M\tau\epsilon^2})$|
> > > |MU-SplitFed(FO)|$O(\frac{F+\sigma^2}{\sqrt{MT\tau}}+\frac{\sigma^2+\tau\zeta^2}{T\tau})$|$O(\frac{F^2+\sigma^4}{M\tau\epsilon^2}+\frac{\sigma^2+\tau\zeta^2}{\tau\epsilon})$|
> > > |SFL-V1(FO)[Rb]|$O(\frac{F+\sigma^2+\zeta^2}{\sqrt{T}})$|$O(\frac{F^2+\sigma^4+\zeta^4}{\epsilon^2})$|
> > > |FedZO[a], DecomFL(ZO)[18]|$O(\frac{\sqrt{d}(F+\sigma^2+\zeta^2)}{\sqrt{MTK}})$|$O(\frac{d(F^2+\sigma^4+\zeta^4)}{MK\epsilon^2})$|
> > >
> > > # Systematic analysis approach to identify the source of convergence improvements
> > >
> > > Since our algorithm design involves multiple factors(e.g. separate model, unbalanced update, multiple clients), we isolate the impact of each element in a step-by-step analysis. We begin by proving the convergence of MU-Split (first-order) where there is only one client, isolating the effect of unbalanced updates. Then we extend our case to the ZO version, where the incorporation of ZOO introduces additional error terms related to ZOO. This gives us our first conclusion in MU-Split as stated in Corollary 4.4: **the improved bound comes from unbalanced updates**.
> > >
> > > Based on that, we gradually extend our cases to the SFL setting where multiple clients are involved. We repeat the same approach by first analyzing under FO to ensure tight bounds, then expanding to the more complicated ZO setting. **As indicated in the Table 1, under the FO setting, our proposed algorithm indicate a tighter bound compared to [Rb], indicating a speed up aligned with our unbalanced updates $\tau$ as well as partitioned clients.**
> > >
> > > As mentioned, our unbalanced update strategy can be extended to include multiple local updates. We have drafted a sketch of the required steps and are happy to provide a formal proof in revision.
> > >
> > > # Sketch Proof for K>1
> > >
> > > For K>1, the client update follows
> > > $
> > > x_c^{t+1}-x_c^t =-\frac{\eta_g}{M}\sum_{m=1}^{M}\sum_{k=0}^{K-1}\eta_c G_{c,m}(x_m^{t,k};\xi_{m}^{t,k}),
> > > $
> > > where $k$ denotes the local update steps. The server updates is similar to this.
> > >
> > > Then, we analyze $\mathcal{K}_1,\mathcal{K}_2,\mathcal{K}_3,\mathcal{K}_4$ in (17) .
> > >
> > > For $\mathcal{K}_1,\mathcal{K}_3$, we use similar techniques as we did in the analysis of $\mathcal{K}_1$ in SFL. For  $\mathcal{K}_2,\mathcal{K}_4$, we use similar techniques for $\mathcal{K}_2$ in SFL.
> > >
> > > Both requires that we bound $A=\sum_{k=0}^{K-1}\sum_{i=0}^{\tau-1}E\|x_{s,m}^{t,k,i}-x_{s,m}^t\|^2$ and $B=\sum_{k=0}^{K-1}E\|x_{c,m}^{t,k}-x_{c,m}^t\|^2$, i.e., we need a different version of Lemma C.3. Here we start from bounding $B$.
> > >
> > > Applying the update formula and the property of martingale difference sequence:
> > > $$
> > > B\le2(\eta_c^t)^2K^2\sum_{k=0}^{K-1}E\|\nabla_{x_c} F_{m,\lambda}^{t,k}\|^2+2(\eta_c^t)^2K\sum_{k=0}^{K-1}E\| G_{c,m}^{t,k}-\nabla_{x_c} F_{m,\lambda}^{t,k}\|^2)(\*)
> > > $$
> > > By similar technique of Lemma C.2, it follows
> > > $$
> > > \begin{align}\sum_{k=0}^{K-1}E\| G_{c,m}^{t,k}-\nabla_{x_c} F_{m,\lambda}^{t,k}\|^2)\le&\sum_{k=0}^{K-1} \left(\frac{6d_c}{P}E\|\nabla_{x_c} f(x^t)\|^2+\frac{6L^2d_c}{P}E\|x_{c,m}^{t,i}-x_{c,m}^{t}\|^2\right.\\\\+&\frac{6L^2d_c}{P}E\|x_{s,m}^{t,i}-x_{s,m}^{t}\|^2\left.+\frac{6 d_c\epsilon^2}{P}+\frac{2 d_c\sigma_{c}^2}{P}+\frac{L^2\lambda^2 d_c^3}{2P}\right)\end{align}
> > > $$
> > > By similar technique of (40), it follows:
> > > $$
> > > E\|\nabla_{x_c} F_{m,\lambda}^{t,k}\|^2\le6E\|\nabla_{x_s}f(x^t)\|^2+6\epsilon^2+\frac{L^2}{2}\lambda^2 d_s^3+6L^2E\|x_{s,m}^{t,k}-x_{s,m}^{t}\|^2+6L^2E\|x_{c,m}^{t,k}-x_{c,m}^{t}\|^2
> > > $$
> > > Substituting the above two inequalities back into (*), we first get a bound of $B$ including $A$. Similarly, we can get a bound of $A$ including $B$. By combining them we can derive their independent bounds.
> > >
> > > # Other Questions
> > >
> > > 1. “Numerical constant 1”: The constant 1 in the bound represents a statistical error term arising from the variance bound $\sigma$ and related factors, while the term $F$ corresponds to the optimization initial error. This separation offers a clearer interpretation of the convergence bound.
> > > 2. Question 3: $f^t_\lambda,f^{t,i}_\lambda$ are actually the same function: the smoothed version of global loss function f(x), where f(x) is defined in (2) and the smoothed version is defined in Lemma A.4. We abused the notation and use the superscript to denote values of model weights at different training rounds $x^{t}$ and $x^{t,i}$.
> > > 3. Weakness 3: “Yes”.
> > >
> > > We want to express our appreciation again towards your dedicated review of our work!

---

> ### Comment · Reviewer_WuCT · 2025-08-06
> **Further questions**
>
> 1. Which algorithm does MU-SplitFed correspond to in SplitFed? SplitFedv1 or SplitFedv2?
>
> 2. Does this paper include the first order methods with unbalanced updates in SplitFed for now? If it does, this is an important contribution for me.
>
> 3. I am not convinced that there is no $O(\eta^2)$ terms in the bound in Theorem 4.7, which is common in FedAvg. See Yang et al. (2021); Karimireddy et al (2020). I only see the $O(\eta)$ terms in the bound.
>
>    Yang et al., Achieving Linear Speedup with Partial Worker Participation in Non-IID Federated Learning, ICLR, 2021.
>
>    Karimireddy et al., SCAFFOLD: Stochastic Controlled Averaging for Federated Learning, ICML, 2020.
>
> 4. It seems that it uses $\eta_g = \frac{1}{\sqrt{M}}$ in Corollary 4.8. For comparison, if the function of $\eta_g$ is the same for all algorithms, I recommend using the same $\eta_g$ for all the algorithms. This is because $\eta_g$ makes a big difference to the bounds. As shown in Cheng et al. (2024)'s Table 1 (the reproduced bound of Yang et al. (2021)), when $\eta_g \to \infty$, FedAvg is even immune to the heterogeneity.
>
> 5. The claims in the rebuttal for $f_\lambda^t$ and $f_\lambda^{t,i}$ are not uniform. Note that function $f$ is related to the model parameters and data points; when the data or parameters change, the function also changes.
>
> Note that the questions with respect to the bounds are mostly from Table 1. I do not met the bounds (that I expected) with the form like Cheng et al. (2024). These bounds seem unfamiliar to me, which makes me a little puzzled.
>
> Cheng et al., Momentum Benefits Non-Iid Federated Learning Simply and Provably, ICLR, 2024.
>
> At last, some notations are not displayed well in openreview, which can be adjusted by adding `\`.

---

> > ### Author Response · Authors · 2025-08-08
> >
> > RL1:
> > MU-SplitFed corresponds to SFL_v1.
> >
> > RL2:
> > Our paper does not include First-Order methods in the SplitFed setting. In this paper, we focus on a resource-constrained regime, where both memory and computational limitations on the client side can lead to straggler issues and slow down the entire training process. In such scenarios, our goal is to reduce the client workload, and ZOO offers a compelling fit for this purpose.
> >
> > While extending FO methods to our setting is indeed interesting, it is beyond the current scope of our paper, as also acknowledged by the reviewer. Meanwhile, we are happy to discuss such an extension and have proved the convergence rates in the table provided in previous response in the discussion with the reviewer. While we believe the current version of the paper is self-contained, we are open to including a discussion of this extension in the revision if the reviewer feels it would strengthen the paper.
> >
> > RL3:
> > We believe the absence of the $O(\eta^2)$ term in our Theorem 4.7 is primarily due to differences in proof techniques. It is indeed common in FL literature[3, 4] for $O(\eta^2)$ terms to arise, typically as a result of bounding the statistical error introduced by local updates. Similarly, our analysis does include $O(\eta^2)$ terms at intermediate steps (see Line 643) to capture the same types of errors.
> >
> > However, in the final convergence result (Theorem 4.7), we apply a scaling and simplification strategy: we reduce all higher-order terms such as $O(\eta^3)$ and $O(\eta^2)$ to $O(\eta)$ to allow similar terms to be merged. This approach helps us address technical challenges specific to the split learning setup, where bounding and recombining the split model components introduces additional complexities compared to conventional FL. The merging of similar terms not only streamlines the analysis but also enables clearer control over convergence behavior and simplifies the choice of step size.
> >
> > We would also like to note that the $\eta^2$ terms don't necessarily appear in final convergence rates in the literature on zeroth-order optimization and split learning. For example, Theorem 1 of [1] and Eq. (155) of [2] do not include $O(\eta^2)$ terms in their final convergence rates due to different proof techniques.
> >
> > RL4: Thanks for your suggestion. We will make it clear in the revision.
> >
> > RL5:
> > The definitions are as follows: $\nabla_{x_s}f_\lambda(x^t)$ $:= \nabla_{x_s} f_\lambda(x_c^t,x_s^{t})$, $\nabla_{x_s}f_\lambda^{t,i}(x_c^t,x_s^{t,i})$$:= \nabla_{x_s}f_\lambda(x_c^t,x_s^{t,i})$. The subscript of the $t, i$ indicates local update steps, and does not represent a different function. We aim to highlight the distinction between local models and the global reference model, following notations used in prior works, e.g., in the last inequality of the chain following equation (33) in [1], and step (a3) in equation (2) in [3]. We will clarify this notation more explicitly in the revised version of the paper.
> >
> > > I do not met the bounds with the form like Cheng et al. (2024). These bounds seem unfamiliar to me
> >
> > Our response:
> >
> > We believe there are two reasons the convergence bounds in our paper may appear unfamiliar.
> >
> > First, our setting focuses on *split learning* and *zeroth-order optimization*, which differ significantly from the FO-FL methods considered in Cheng et al. (2024). As a result, a direct comparison is not straightforward.
> >
> > Second, the rates reported in our tables are taken directly from the corresponding reference papers, using their best-known learning rates. These rates typically focus on key complexity measures such as communication rounds, local update steps, and the number of clients—consistent with the conventions in optimization literature[1,3,4].
> >
> > While Cheng et al. (2024) provides helpful summary tables, the reported bounds are sometimes expressed using different parameter dependencies (e.g. explicit constants involving gradient variance or smoothness parameters). As an example, consider the rate listed for the first baseline (Yu et al., 2019b) in Cheng et al. (2024): this differs from the original convergence rate stated in Corollary 1 of Yu et al. (2019b). A similar mismatch can also be seen by comparing the rates in Cheng et al. (2024) with those in Table 2 of the SCAFFOLD paper.
> >
> > That said, it is easy to verify that the convergence rates in our table, those in Cheng et al. (2024), and the original baseline papers are consistent in terms of dominant order. The differences primarily lie in constant factors and the presentation of problem-dependent parameters such as the variance bound $\sigma$ and related terms.
> >
> > 1. “Achieving dimension-free communication in federated learning via zeroth-order optimization”
> > 2. “Convergence Analysis of Split Federated Learning on Heterogeneous Data”
> > 3. “Achieving linear speedup with partial worker participation in non-iid federated learning”
> > 4. “SCAFFOLD: Stochastic Controlled Averaging for Federated Learning”

---

> ### Comment · Reviewer_WuCT · 2025-08-09
>
> The responses seem reasonable. I realize that I may not be as familiar with the existing works in split learning as I thought I was.
>
> The concern about the tightness of the bounds (i.e., $O(\eta^2)$ terms) still exists. This is mainly because the original SplitFedv1 (or SFL_v1) (in Thapa et al, (2022)) is equivalent to FedAvg in the absence of model partitioning. So, in my opinion, the bounds of SplitFed may be quite similar to those of FedAvg. However, this does not seem to be the case, based on the existing works (Han et al., 2024) and this paper.
>
> As a consequence, I decide to increase the rating to 4, and decrease the confidence to 3, to avoid the rejection of this paper due to my unfamiliarity with the existing works in split learning.
>
> Some suggestions are given:
>
> 1. I hope the authors can consider the concern more deeply in the revised version. Some intuitive discussion is still required.
> 2. I also hope the authors consider the form of Koloskova et al. (2020); Cheng et al. (2024) in the revised version. As you have said, they consider the other factors (beyond the training rounds, local update steps, and the number of clients), which is more professional and convincing for me.
> 3. In addition, try to avoid using "communication round" in split learning, as communication also exists in the stage of local updates. See Thapa et al, (2022); no "communication round" is used. One alternative can be "training round".
> 4. At last, I hope the authors revise the paper as promised.
>
> Thapa et al., SplitFed When Federated Learning Meets Split Learning, AAAI, 2022.
>
> Han et al., Convergence Analysis of Split Federated Learning on Heterogeneous Data, NeurIPS, 2024.
>
> Koloskova et al., A Unified Theory of Decentralized SGD with Changing Topology and Local Updates, ICML, 2020.
>
> Cheng et al., Momentum Benefits Non-Iid Federated Learning Simply and Provably, ICLR, 2024.

---

> > ### Author Response · Authors · 2025-08-09
> >
> > We sincerely thank the reviewer for the recognition of our work and for the constructive suggestions that help improve our paper quality.

---

### Official Review · Reviewer_yz22 · 2025-07-02

**Clarity:** 3
**Significance:** 2
**Originality:** 2
**Rating:** 4
**Confidence:** 4

**Summary:**

This paper proposes an MU-SplitFed method to address the straggler problem in Split Federated Learning. Unlike existing methods, the MU-SplitFed approach reduces communication rounds between the server and clients to mitigate the impact of stragglers. Furthermore, a Zeroth-Order optimization technique is introduced on the client side to enable training without back-propagation. Theoretical analysis demonstrates that MU-SplitFed achieves linear speedup. Experimental results validate the effectiveness of the proposed method.

**Questions:**

1.More experiments should be added. This paper focuses on the straggler problem, but the experiments and analysis under straggler scenario are limited. Only SplitFed and GAS are used as the baseline methods, other SOTA methods for straggler problem in related works should be compared. The experiments also should be conducted on more complex dataset and LLMs.

2.The novelty of the proposed method is limited. Only an unbalanced update ratio /tau is introduced on the server-side, which is trivial.

**Ethical Concerns:**

["NO or VERY MINOR ethics concerns only"]

**Final Justification:**

The authors have done a good job in rebuttal. I raised my rating.

**Limitations:**

Yes

**Quality:**

2

**Strengths And Weaknesses:**

Strengths:
1.This paper has clearly described the proposed method.
2.The analysis is sufficient.
Weaknesses:
1.The experiments should be improved;
2. The novelty of this method is limited.

---

> ### Author Rebuttal · Authors · 2025-07-31
>
> **Response to Reviewer's Novelty Concerns:**
>
> We believe that our work identifies and addresses fundamental challenges in Split Federated Learning (SFL) that have not been properly analyzed in prior work.
>
> **The Critical Challenge in SFL:**
>
> The goal of Split Federated Learning (SFL) is to reduce memory usage on resource-constrained edge devices by dividing the model between the client and the server. However, this model partitioning introduces additional communication overhead during each local update round. As a result, SFL faces a fundamentally different straggler problem: unlike traditional Federated Learning (FL), where stragglers only delay global aggregation, stragglers in SFL block **every local iteration,** significantly hindering the training progress.
>
> Existing work on addressing straggler issue in SFL remains limited[8,9,16]. Most approaches borrow straggler mitigation strategies from FL, primarily focusing on minimizing server waiting time. However, under the relay-based update mechanism in SFL, this strategy proves less effective, yielding only marginal improvements. These limitations highlight the need to revisit the straggler problem in SFL.
>
> **Our Novel Solution - MU-SplitFed:**
>
> Acknowledging the limitations of current work, we propose MU-SplitFed that combines unbalanced updates with ZOO to address straggler issue from a fundamentally different perspective-by effectively reducing the number of global communication rounds. For further improvement, we introduce an eager forward method that minimizes server waiting time in each local round. The effectiveness of this method in reducing time delay has been empirically validated in Table 4.
>
> **Contribution I: Solving Straggler Issues in SFL**
>
> 1. **Improved Convergence with Sublinear Speedup:** Our MU-SplitFed provides sublinear speed-up regarding unbalanced update steps, parameterized by $τ.$ Specifically, we improve the convergence rate from $O(\frac{\sqrt{d}}{\sqrt{T}})$ to $O(\frac{\sqrt{d}}{\sqrt{\tau T}})$, without relying on strong assumptions such as bounded gradients.
> 2. **Theoretical Solution to Straggler Problem:** We provide direct theoretical insight into how our unbalanced strategy effectively solves the straggler issue under Corollary 4.8, addressing the fundamental communication bottleneck in SFL.
> 3. **Practical Effectiveness:** Our extensive experimental results demonstrate the advantage of our proposed framework in both small models and LLMs. As indicated in supplementary Table 2, our proposed unbalanced update can reduce more than half of the communication rounds, demonstrating the effectiveness of our algorithm in practical application.
>
> **Contribution II: Additional Theoretical Insights**
>
> 1. **Principled Model Splitting Strategy:** Our work reveals important theoretical interpretation regarding the choice of cutting position, which coordinates with our unbalanced update strategy. Our theory indicates that for a larger choice of $τ$, more layers should be placed on the server side. The dependency between the convergence rate of ZOO and parameter dimension $d$ allows us to build this connection, enabling our theory to provide guidance for hyperparameter selection and making our algorithm more explainable.
> 2. **Dimension-Free ZOO:** Our unbalanced update strategy can achieve dimension-free ZOO without further assumptions. Based on Corollaries 4.4 and 4.8, by appropriately scaling the unbalanced update factor $τ$ to the same order as the model dimension $d$, the convergence rate becomes independent of $d$. This directly addresses a key limitation of traditional ZOO methods, where convergence rates typically deteriorate with increasing dimension.
>
> In summary, MU-SplitFed introduces a **fundamentally new perspective on handling stragglers in SFL**. We believe our contributions, both theoretical and empirical, establish a strong case for novelty and impact in this underexplored space.
>
> **Regarding your doubt of “straggler scenario are limited”**
>
> We need to clarify that introducing time delay is a widely adopted approach to simulate the impact of stragglers in distributed learning. This method is well-supported in prior literature [8–14]. To establish a more realistic model that reflects the randomness of clients' time delay, we consider an exponential distribution model, which has been widely used to capture the computation delay for distributed clusters
>
> Here are more works that use this methodologies:
>
> a. “Straggler-resilient federated learning: Leveraging the interplay between statistical accuracy and system heterogeneity”
>
> b. "Speeding up distributed machine learning using codes."
>
> c. "Coded computation over heterogeneous clusters.”
>
> **Regarding your doubt about lack of comparison with other SOTA methods**
>
> Thank you for your suggestion. However, we must clarify that related work addressing straggler issues specifically in the Split Federated Learning setting is extremely limited. We have identified only three recent works tackling straggler issues in SFL [8,9,16] and we provide detailed justification for our comparison choices below.
> [8]: is compared in our manuscript
> [16]: This paper proposes a ring-structured SFL framework, which fundamentally differs from the standard SFL architecture. We believe that comparing algorithms under different ML system structures would not provide meaningful insights.
>
> [9]: While this work addresses straggler issues in SFL, it has significant limitations that make it less effective than our approach. The paper proposes an adaptive splitting strategy that dynamically adjusts layer allocation between clients and server. Specifically, the model is split into parts A, B, and C, where B consists of m+n layers. The client computes A+m layers (while still need to contain A+B), the server computes n+C layers (while still need to contain B+C), and m and n are dynamically adjusted based on client computational capacity.
>
> However, we identify two fundamental limitations of this method:
>
> 1. Additional computational overhead: The dynamic layer adjustment requires continuous monitoring and reallocation, introducing extra computation time on the server side.
> 2. Limited improvement potential: The effectiveness is constrained by the fixed size of the overlap region B. Since B's size is fixed, the potential improvement is inherently bounded and comparatively small.
>
> In contrast, our MU-SplitFed addresses these limitations through a fundamentally different approach: we directly reducing communication rounds rather than merely minimizing client computation time. **Our key insight is that the computation disparity between client and server is considerably large in SFL.** Instead of optimizing the already limited client-side computation time, our framework intelligently utilizes the server's idle time by applying multiple updates strategy. As demonstrated in Table 2, **our method achieves more than 50% reduction in communication rounds,** which is significantly more effective for overall training time reduction.
>
> If you are aware of other SOTA methods for straggler issue in SFL, please feel free to recommend them to us. We would be happy to conduct additional experiments for further comparison.
>
> **Regarding your request about “The experiments also should be conducted on more complex dataset and LLMs.”**
>
> Thank you for your suggestion. In response, we have conducted and included additional experiments across a variety of datasets and model architectures, as detailed below:
>
> 1. **LLM Benchmarks with OPT-125M (Table 1).** We evaluated our proposed MU-SplitFed against SplitFed_V1 [2] using the OPT-125M model on multiple LLM benchmarks, including **SST-2, CB, WSC, WIC, RTE, and BoolQ**. With $L_c=2$ and $\tau=2$, MU-SplitFed consistently outperforms SplitFed_V1 across all tasks, demonstrating its effectiveness under various language understanding challenges.
> 2. **Ablation Study with OPT-1.3B (Table 2).** We conducted an ablation study on the SST-2 dataset using OPT-1.3B to assess the impact of $\tau$ and $L_c$ in reducing the communication rounds. According to the result, for a large choice of $\tau$, less portion of the model should be assigned to client side. This aligns with our theoretical analysis on the interplay between unbalanced update steps and splitting strategy.
> 3. **Image Classification under Data Heterogeneity (Table 5).** To evaluate the performance of our proposed algorithm under data heterogeneity, we added experiments on CIFAR-10 and FashionMNIST under non-IID setting with high heterogeneity (shard=2). The result shows that MU-SplitFed outperforms the baseline, indicating the robustness of our designed algorithm under data heterogeneity.
> 4. **Ablation Study on Update Strategy (OPT-1.3B, SST-2).** To investigate the reason of  performance boost, we compared both FO and ZO versions of MU-Split on the SST-2 dataset. Specifically, we implemented MU-Split in both its FO and ZO versions. As shown in Table 6,7, we observe consistent performance improvement in both FO and ZO versions, indicating that the **accuracy gain primarily stems from the unbalanced server update schedule**. This further confirms the effectiveness of our proposed unbalanced update strategy in enhancing both the convergence and accuracy.
>
> Table 1 Best accuracy after 1500 rounds (OPT125M, Zeroth Order)
> | Dataset | SplitFed_V1 (ZO version) | MU-SplitFed ($\tau=2$) |
> | --- | --- | --- |
> | SST-2 | 83.99 | 84.70 |
> | CB | 72.49 | 74.67 |
> | WSC | 55.18 | 58.35 |
> | WIC | 53.25 | 54.01 |
> | RTE | 52.91 | 55.78 |
> | BoolQ | 61.64 | 62.46 |
>
>  Table 2 Required rounds to reach 85% accuracy of MU-Split (OPT-1.3B, Zeroth Order)
> | Split Layer | $\tau=1$ | $\tau=2$ | $\tau=3$ | $\tau=4$ | $\tau=5$ | $\tau=6$ |
> | --- | --- | --- | --- | --- | --- | --- |
> | 2 | 38 | 17 | 19 | **16** | 18 | 18 |
> | 4 | — | 18 | **16** | 22 | 20 | 33 |
> | 8 | — | 23 | **22** | 26 | 22 | 32 |
> | 12 | — | **22** | 32 | 25 | 29 | 32 |
> | 16 | — | **21** | 29 | 28 | 40 | 36 |

---

### Decision · Program_Chairs · 2025-09-17

**Decision:**

Accept (poster)

**Comment:**

This paper proposes a straggler-resistant split FL method using an unbalanced update mechanism. Zeroth-order optimization enables efficient training in SFL settings. A theoretical proof for linear speedup is also provided. On the other hand, questions remain regarding how the presence of stragglers interacts with data heterogeneity. Overall, the merits outweigh the shortcomings.